# Phosphorylation of Jhd2 by the Ras-cAMP-PKA(Tpk2) pathway regulates histone modifications and autophagy

Qi Yu [1,3], Xuanyunjing Gong[1,3], Yue Tong[1], Min Wang[2], Kai Duan[1], Xinyu Zhang[1], Feng Ge [2], Xilan Yu [1] ✉ & Shanshan Li [1] ✉

Cells need to coordinate gene expression with their metabolic states to maintain cell homeostasis and growth. How cells transduce nutrient availability to appropriate gene expression remains poorly understood. Here we show that glycolysis regulates histone modifications and gene expression by activating protein kinase A (PKA) via the Ras-cyclic AMP pathway. The catalytic subunit of PKA, Tpk2 antagonizes Jhd2-catalyzed H3K4 demethylation by phosphorylating Jhd2 at Ser321 and Ser340 in response to glucose availability. Tpk2-catalyzed Jhd2 phosphorylation impairs its nuclear localization, reduces its binding to chromatin, and promotes its polyubiquitination and degradation by the proteasome. Tpk2-catalyzed Jhd2 phosphorylation also maintains H3K14 acetylation by preventing the binding of histone deacetylase Rpd3 to chromatin. By phosphorylating Jhd2, Tpk2 regulates gene expression, maintains normal chronological life span and promotes autophagy. These results provide a direct connection between metabolism and histone modifications and shed lights on how cells rewire their biological responses to nutrient signals.

Cell metabolism and gene expression are two important cellular processes that are reciprocally regulated in all living organisms[1,2]. To optimize cell growth and maintain cell survival, cells need to coordinate their transcription and metabolism to adapt to environmental nutrition changes. This adaptation can be achieved by the cooperation of multiple signal molecules as sensors of nutrition availability. The extensively studied nutrient responsive molecules are AMP-activated protein kinase (AMPK), the mechanistic target of rapamycin (mTOR), and cyclic AMP (cAMP)-dependent protein kinase A (PKA)[3]. While AMPK is activated by low energy or lack of nutrients to inhibit cell growth, mTOR is activated by nutrient availability to promote cell growth[4,5]. PKA is a tetrameric holoenzyme, which is composed of a regulatory subunit Bcy1 dimer and two catalytic subunits encoded by *TPK1*, *TPK2* and *TPK3*. The glycolytic metabolite, fructose-1,6-biphosphate (FBP) activates small

GTP-binding proteins, Ras1 and Ras2 through the guanine exchange factor, Cdc25[6]. Ras stimulates adenylate cyclase (Cyr1) to synthesize cAMP, which then activates PKA by binding to Bcy1 regulatory subunit to release the catalytic subunits[7]. Activated PKA then exerts its cellular functions by phosphorylating its target proteins[7], which constitutes the Ras/cAMP/PKA pathway that regulates vegetative growth, carbohydrate metabolism, morphogenesis, stress resistance, cell cycle progression and meiosis[7–10]. In addition to the Ras/cAMP/PKA pathway, Cyr1 can also be stimulated by glucose-sensing G-protein-coupled receptor (GPCR) system, which is composed of G-protein-coupled receptor Gpr1, G protein α subunit Gpa2, and its GTPase activating protein Rgs2, so called the Gpr1/Gpa2 branch[3]. Nonetheless, the precise mechanism by which these signaling pathways regulate the biological responses of cells to nutrient availability remains to be resolved.

[1]State Key Laboratory of Biocatalysis and Enzyme Engineering, School of Life Sciences, Hubei University, Wuhan, Hubei 430062, China. [2]Key Laboratory of Algal Biology, Institute of Hydrobiology, Chinese Academy of Sciences, Wuhan, Hubei 430072, China. [3]These authors contributed equally: Qi Yu, Xuanyunjing Gong. ✉e-mail: yuxilan@hubu.edu.cn; shl@hubu.edu.cn

Gene expression can be dynamically regulated by epigenetic modifications in response to environmental changes. The well-known epigenetic modifications are histone post-translational modifications, including acetylation, methylation, phosphorylation, ubiquitination, which play important roles in regulating chromatin structure and gene expression[11–13]. As most histone modifying enzymes require metabolites as cofactors and/or substrates, histone modifications are also subject to metabolic regulation[12,13]. For example, glucose can be converted to acetyl-CoA, which is then fueled to histone acetyltransferases (HATs) to increase the overall histone acetylation and alter the global chromatin structure[14,15]. We have previously reported that glycolysis is required for pyruvate kinase-mediated histone H3 phosphorylation at threonine 11 (H3pT11)[16], which confers cell resistance to oxidative stress and maintains normal telomere silencing[17,18]. Pyruvate kinase-containing SESAME complex interacts with histone acetyltransferase SAS complex to promote histone H4K16ac and maintain transcription silencing at subtelomere regions[19].

H3 lysine 4 trimethylation (H3K4me3) is a conserved histone modification that plays critical roles in gene transcription, DNA replication and repair, and class-switch recombination[20]. The mutation of H3K4 to methionine (H3K4M) has been reported to occur in several cancer types and H3K4 methylation defects are closely correlated with diverse pathologies such as acute myeloid leukemia (AML)[21–23], emphasizing the importance of identifying H3K4me3 regulatory pathways. In budding yeast, H3K4me3 is catalyzed by the SET domain-containing histone methyltransferase Set1 and demethylated by the JmjC domain-containing demethylase Jhd2[24,25], which serve as a good model to study H3K4 methylation[26]. Jhd2 has been reported to demethylate H3K4 to regulate the transcription of postmeiotic genes that are important for the production of healthy meiotic progeny[27]. Although H3K4 is demethylated by Jhd2, loss of Jhd2 only mildly increases H3K4me3 when cells were grown under standard 2% glucose conditions[28,29]. Loss of Jhd2 strongly increases H3K4me3 when cells were grown in glucose-depletion media with acetate as the sole carbon source (YPA)[29], implying that Jhd2 may be regulated by an unknown mechanism in response to glucose availability.

In the current study, we identify a direct connection between nutrient responsive pathway and histone modifications by demonstrating that Ras-cAMP activated PKA(Tpk2) phosphorylates Jhd2 to promote H3K4me3. Further study uncovers the mechanism by which Tpk2 inhibits Jhd2 activity and reduces Jhd2 protein stability in response to glucose, which maintains normal chronological life span. In addition, Tpk2-catalyzed Jhd2 phosphorylation promotes H3K14ac by antagonizing histone deacetylase Rpd3. By inhibiting the activity of Jhd2 and Rpd3, Tpk2-catalyzed Jhd2 phosphorylation facilitates the transcription of autophagy genes and promotes the autophagy pathway. Our study uncovers a signaling pathway that integrates glycolysis and histone modifications to regulate gene transcription, chronological life span and autophagy.

## Results

### Glucose induces H3K4me3 in a manner dependent on the Ras-cAMP-PKA(Tpk2) pathway

To examine the effect of glucose on histone modifications, we grew yeast cells in 2% glucose-containing medium (YPD) until log phase. Cells were collected and then grown in the medium containing different concentrations of glucose for 0.5 hr. By Western blot analysis of known histone modifications, we found that glucose induced H3K4me3 in a dose-dependent manner but had no effect on H3K4me1 and H3K4me2 (Fig. 1a; Supplementary Fig. 1a). Blocking glycolysis by adding 2-deoxy-glucose (2-DG) significantly reduced H3K4me3 (Fig. 1b).

To identify the signaling pathway that mediates glucose-induced H3K4me3, we first examined the effect of glucose on histone modifications that regulate Set1-catalyzed H3K4me3, including H2BK123 monoubiquitination (H2Bub) and H3R2 asymmetric dimethylation

(H3R2me2a)[30–32]. Glucose induced H2Bub but had no significant effect on H3R2me2a (Supplementary Fig. 1b). We then determine whether glucose increases H3K4me3 by inducing H2Bub. As there is no H3K4me3 in the deletion mutants of RAD6 and BRE1, which catalyze H2Bub[30,31], it is impossible to determine the effect of glucose on H3K4me3 in rad6Δ and bre1Δ mutants. We thus examined the effect of glucose on H3K4me3 in the deletion mutant of UBP8 and UBP10, which encode H2B ubiquitin proteases[33,34]. Although H2Bub remained relatively high and constant in ubp8Δ ubp10Δ mutant, H3K4me3 was still increased by glucose (Supplementary Fig. 1c), suggesting there is a H2Bub-independent signaling pathway for glucose to induce H3K4me3. We also examined the effect of glucose on the activity of Set1-containing complex (COMPASS). COMPASS (Spp1-CBP) was purified from cells grown in 0.05% glucose and 2% glucose, respectively and the in vitro histone methyltransferase (HMT) assay showed that glucose did not affect the HMT activity of COMPASS (Supplementary Fig. 1d).

We then examined H3K4me3 in the mutants of potential nutrient sensitive signaling molecules, including AMPK (Snf1), mTOR, Sch9 and PKA (Fig. 1c)[9,35]. Loss of Snf1, mTOR (Tor1, Tco89) and Sch9 had no significant effect on H3K4me3 (Fig. 1d; Supplementary Fig. 1e). We then examined the role of PKA in glucose-induced H3K4me3 in deletion mutants of TPK1, TPK2, and TPK3, which encode the catalytic subunits of PKA. These three Tpks have distinct substrates and different functions in cellular processes[36]. For example, Tpk1, but not Tpk2 and Tpk3 regulates non-homologous end joining double-stranded break repair by phosphorylating Nej1[37]. While Tpk2 activates pseudohyphal growth, Tpk3 inhibits filamentation and Tpk1 has no effect[38]. Loss of Tpk2 but not Tpk3 significantly reduced H3K4me3 (Fig. 1d, e; Supplementary Fig. 1f). Loss of Tpk1 slightly reduced H3K4me3 in WT but not in tpk2Δ mutant (Fig. 1e). Moreover, increasing glucose concentrations induced H3K4me3 in WT but not tpk2Δ mutant (Fig. 1f), indicating that Tpk2 is required for glucose to induce H3K4me3.

PKA is activated by cyclic AMP (cAMP), which binds to its inhibitory subunit, Bcy1 and causes its dissociation (Fig. 1g). We thus examined H3K4me3 in deletion mutants of cAMP phosphodiesterases PDE1 and PDE2, which encode enzymes to hydrolyze cAMP and inactivate PKA[39,40]. In accordance with increased cAMP levels in pde1Δ and pde2Δ mutants[39,40], H3K4me3 was significantly elevated (Fig. 1h). cAMP is synthesized by adenylate cyclase (Cyr1), which is stimulated by the small GTP-binding proteins Ras1 and Ras2 to form the Ras-cAMP-PKA signaling cascade (Fig. 1g). To test whether the Ras-cAMP-PKA pathway mediates glucose-induced H3K4me3, we examined H3K4me3 in ras1Δ and ras2Δ mutants. Loss of Ras1 and Ras2 significantly reduced H3K4me3 (Supplementary Fig. 1g). Ras1/2 can be activated by glycolysis-derived fructose-1,6-biphosphate (FBP) via Cdc25[6] (Fig. 1g). FBP but not pyruvate treatment increased H3K4me3 in a dose-dependent manner (Supplementary Fig. 1h). In line with these results, blocking FBP biosynthesis by deleting genes encoding hexokinase (HXK1, HXK2) and 6-phosphofructokinase (PFK1, PFK2) significantly reduced H3K4me3 (Supplementary Fig. 1i, j).

The activity of Cyr1 can also be stimulated by the Gpr1/Gpa2 branch that consist of G-protein-coupled receptor Gpr1, G protein α subunit Gpa2, and its GTPase activating protein Rgs2[3] (Fig. 1g). We thus examined whether the Gpr1/Gpa2 branch regulates glucose-induced H3K4me3. Loss of Gpr1 and Gpa2 had no significant effect on H3K4me3 (Fig. 1i; Supplementary Fig. 1k). Collectively, these results indicate that the Ras-cAMP-PKA(Tpk2) pathway but not Gpr1/Gpa2 branch is required for glucose to induce H3K4me3.

### Tpk2 phosphorylates Jhd2 at ser321 (S321) and ser340 (S340) in response to glucose

To understand how Tpk2 regulates H3K4me3, we first examined the effect of Tpk2 on H2Bub and H3R2me2a that regulate Set1-catalyzed H3K4me3. Loss of Tpk2 did not affect these two modifications (Supplementary Fig. 2a). Tpk2 had no significant effect on the intracellular

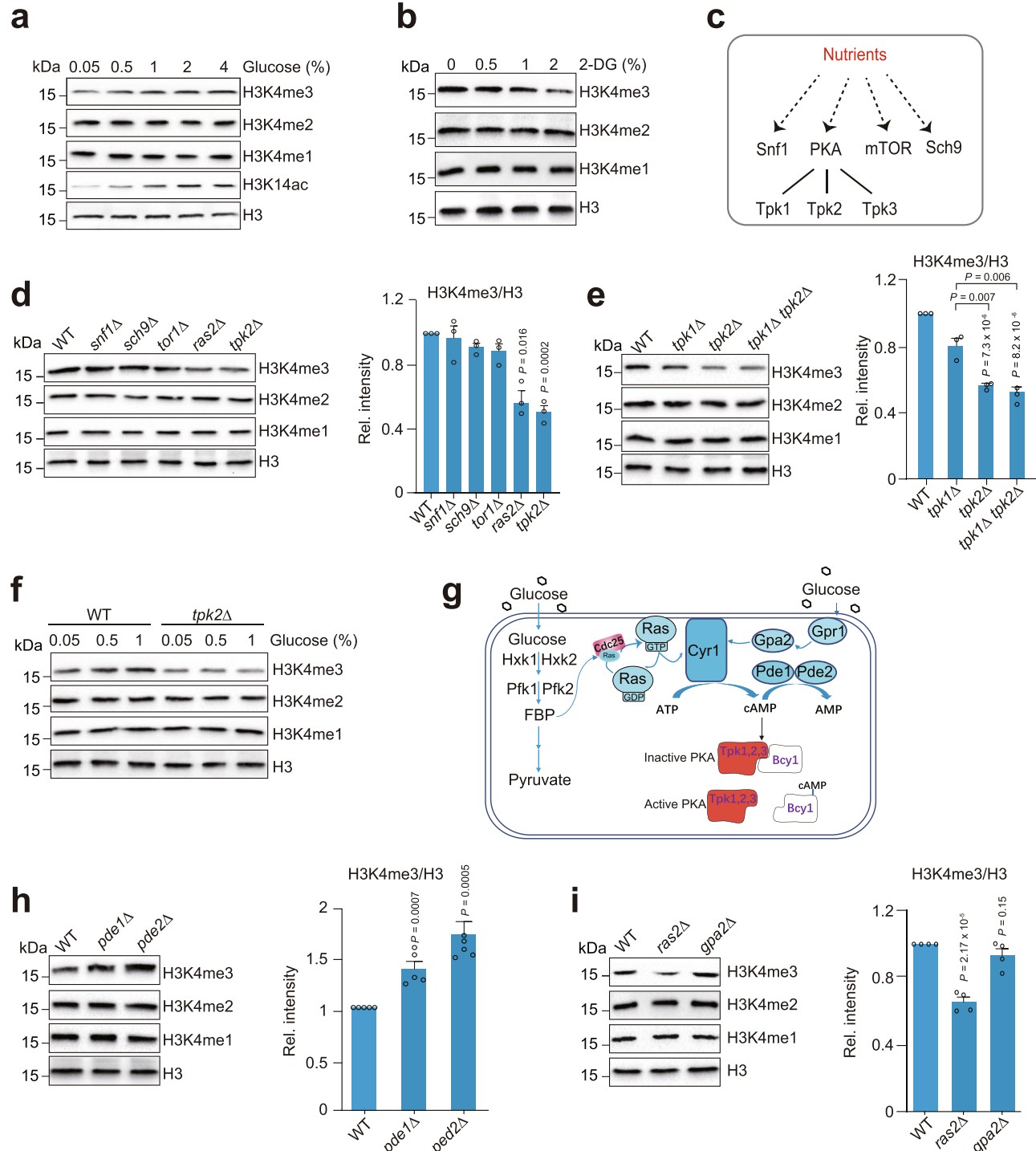

**Fig. 1 | Glucose induces H3K4me3 via the Ras-cAMP-PKA pathway. a** Effect of glucose on histone modifications. Cells were grown in YPD medium until $OD_{600}$ of 0.7–1.0 followed by treatment with YP medium supplemented with different concentrations of glucose for 0.5 hr. Cells were harvested and the extracted histones were analyzed by Western blots with indicated antibodies. **b** Effect of 2-Deoxy-D-glucose (2-DG) on H3K4 methylation. **c** Diagram showing nutrient sensors in budding yeast. **d** Effect of nutrient sensors on H3K4 methylation. **e** Western blot analysis of the effect of Tpk1 and Tpk2 on H3K4 methylation. **f** Tpk2 is required for glucose to induce H3K4me3. **g** Diagram showing the connection between glycolysis and the Ras-cAMP-PKA pathway. **h** The global H3K4me3 was increased in *pde1Δ* and *pde2Δ* mutants. **i** Western blot analysis of H3K4 methylation in WT, *ras2Δ*, and *gpa2Δ* mutants. For **d** and **e**, data represent the mean ± SEM of three biological independent experiments. For **h**, data represent the mean ± SEM of five biological independent experiments. For **i**, data represent the mean ± SEM of four biological independent experiments. Two-sided t-tests were used for statistical analysis. Source data are provided as a Source data file.

S-adenosylmethionine (SAM) levels, which serves as a methyl donor for H3K4 methylation (Supplementary Fig. 2b). The expression of SAM synthetase Sam1 were also unchanged in *tpk2Δ* mutant (Supplementary Fig. 2c), suggesting that Tpk2 regulates H3K4me3 independent of affecting SAM levels. We then examined the effect of Tpk2 on the expression and activity of COMPASS. Loss of Tpk2 had no significant effect on the protein levels of Set1 and COMPASS subunits (Supplementary Fig. 2c, d). We also purified COMPASS from WT and the *tpk2Δ* mutant. Loss of Tpk2 did not affect the subunit composition of COMPASS or its histone methyltransferase activity (Supplementary Fig. 2e, f).

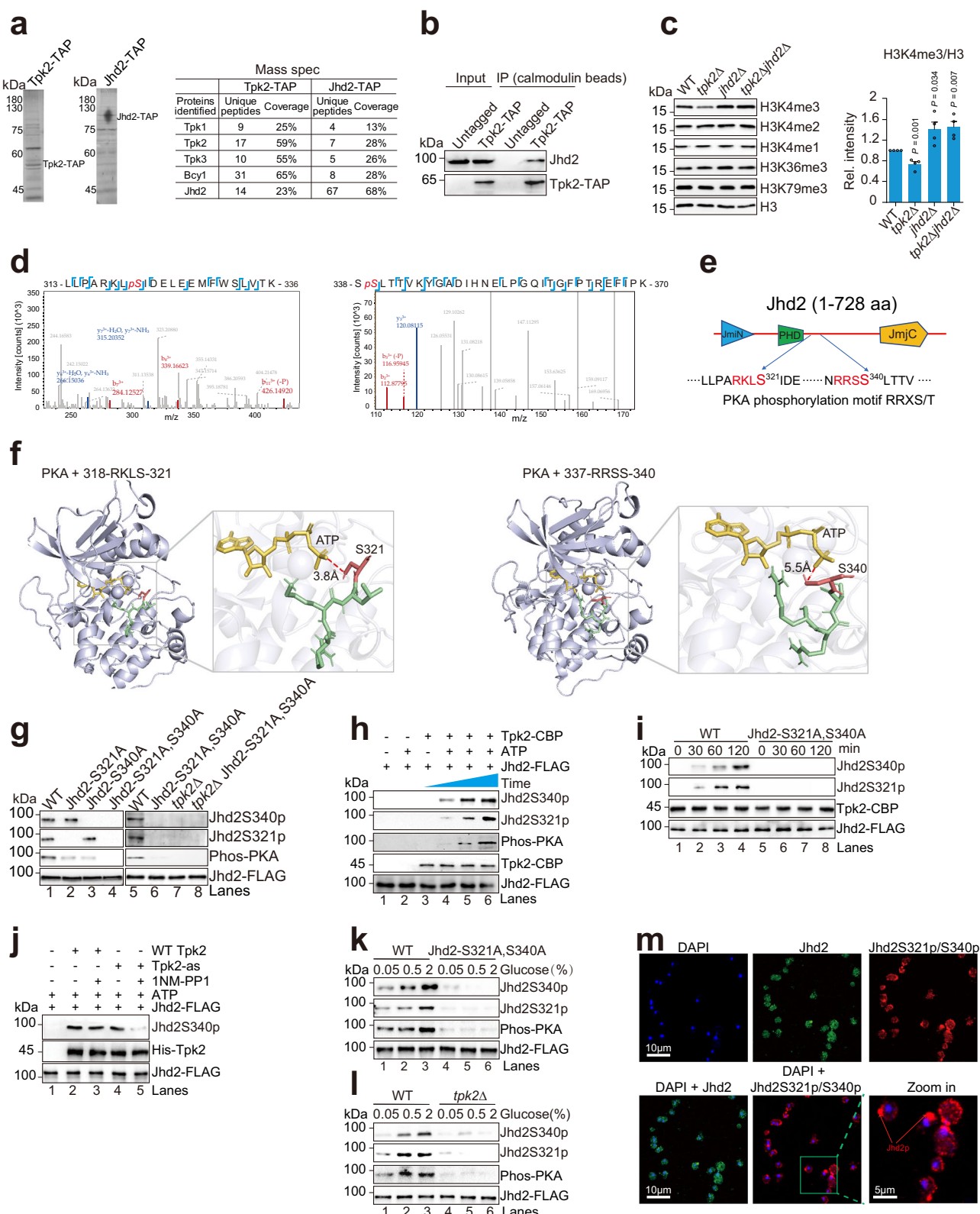

We then immunoprecipitated Tpk2 and performed liquid chromatography-mass spectrometry (LC-MS) analysis, which revealed that H3K4 demethylase Jhd2 was a Tpk2-interacting protein (Fig. 2a). LC-MS analysis of immunoprecipitated Jhd2 also revealed the presence of peptides corresponding to Tpk2 (Fig. 2a). To confirm the interaction between Tpk2 and Jhd2, we immunoprecipitated Tpk2 from Tpk2-TAP cells and found that Jhd2 co-immunoprecipitated with Tpk2 (Fig. 2b). FLAG-purified Jhd2 also directly interacted with TAP-purified Tpk2 (Supplementary Fig. 3a). We then examined the effect of Tpk2 on Jhd2-mediated H3K4 demethylation. Deletion of *JHD2* in *tpk2Δ* mutant rescued the reduced H3K4me3 in *tpk2Δ* mutant (Fig. 2c), suggesting that Tpk2 could antagonize the demethylase activity of Jhd2 to maintain H3K4me3.

**Fig. 2 | Tpk2 phosphorylates Jhd2 at S321 and S340 in response to glucose. a** LC-MS analysis of proteins co-purified with Tpk2 and Jhd2. The identified unique peptides and sequence coverage were listed. **b** Jhd2 interacts with Tpk2 as determined by Co-IP assay. **c** Western blot analysis of histone modifications in WT, *tpk2Δ*, *jhd2Δ*, and *tpk2Δ jhd2Δ* mutants. **d** Jhd2 was phosphorylated at serine 321 and serine 340 as determined by mass spectrometry. **e** Diagram showing Jhd2 protein domains and two phosphorylation sites. **f** Structural view of PKA subunit with Jhd2 peptides (318-RKLS-321, 337-RRSS-340) generated from molecular dynamic simulation. The proteins were shown as cartoons and colored cyan. The ATP and peptides were shown in sticks. Jhd2 S321 and S340 are highlighted in red, respectively. **g** Jhd2 was phosphorylated at S321 and S340 in vivo. Jhd2 was immunoprecipitated from WT Jhd2, Jhd2-S321A, Jhd2-S340A, Jhd2-S321A, S340A, *tpk2Δ* and *tpk2Δ* Jhd2-S321A, S340A mutants. The phosphorylation level of Jhd2 was determined by Western blots with anti-Phos-PKA, anti-Jhd2S321p and anti-Jhd2S340p antibodies. **h-i** Jhd2 was phosphorylated at S321 and S340 by purified Tpk2 in vitro. **j** Jhd2 was phosphorylated by WT Tpk2 but not Tpk2-as mutant when treated with 25 μM 1NM-PP1 as determined by in vitro kinase assay. **k-l** Analysis of the effect of glucose on Jhd2 phosphorylation in WT, Jhd2-S321A, S340A, and *tpk2Δ* mutants. **m** The phosphorylated Jhd2 was localized in the cytoplasm as determined by immunofluorescence microscopy. For **c**, data represent the mean ± SEM of four biological independent experiments. Two-sided t-tests were used for statistical analysis. Source data are provided as a Source data file.

LC-MS/MS analysis of Jhd2 revealed that serine 321 (S321) and serine 340 (S340) of Jhd2 were phosphorylated (Fig. 2d), which are located within the consensus PKA phosphorylation motif in Jhd2 between the PHD and JmjC domains (Fig. 2e)[41]. The molecular docking assay showed that both Jhd2 S321 and S340 residues were in close proximity to ATP within the catalytic domain of PKA (Fig. 2f), while the phosphorylated S321 and S340 residues were far away from ATP (Supplementary Fig. 3b). We thus used the phospho-PKA antibody that recognizes phosphorylated serine/threonine within the consensus PKA motif to measure Jhd2 phosphorylation levels. The phosphorylation signal was detected in Jhd2 immunoprecipitated from WT but not from the *tpk2Δ* mutant (Fig. 2g, lane 5 vs lane 7). We then individually mutated S321 and S340 in Jhd2 to nonphosphorylatable alanine to construct the Jhd2-S321A and Jhd2-S340A mutants. Compared with WT Jhd2, the phosphorylation level was reduced in Jhd2-S321A and Jhd2-S340A mutants (Fig. 2g, lane 1–3). Double mutation of Jhd2 S321A and S340A (Jhd2-S321A, S340A) resulted in no detectable phosphorylation signal (Fig. 2g, lane 1 vs lane 4). To further show that Tpk2 phosphorylates Jhd2 at S321 and S340, we developed antibodies against phospho-specific forms of these two sites, Jhd2S321p and Jhd2S340p with high specificity (Supplementary Fig. 3c). Western blots and immunofluorescence analysis with these two antibodies detected phosphorylated S321 and S340 in WT but not Jhd2-S321A, S340A mutant (Supplementary Fig. 3d, e). Loss of Tpk2 also abolished phosphorylation of Jhd2 at S321 and S340 (Fig. 2g). In contrast, loss of Tpk1 and Tpk3 had no effect on Jhd2 phosphorylation (Supplementary Fig. 3f). An in vitro kinase assay with purified Jhd2 and Tpk2 showed that Tpk2 phosphorylated WT Jhd2 at S321 and S340 (Fig. 2h, lanes 4–6). When Tpk2 was incubated with Jhd2-S321A, S340A mutant, no phosphorylation was detected (Fig. 2i, lanes 5–8; Supplementary Fig. 3g). To further verify that Tpk2 phosphorylates Jhd2, we constructed a Shokat allele of Tpk2 (Tpk2-as) by mutating M147A within its kinase active site, whose kinase activity can be inhibited by adding the bulky ATP analog, 1NM-PP1[42,43]. The intracellular Jhd2 phosphorylation was significantly reduced in Tpk2-as mutant but not WT cells upon treatment with 1NM-PP1 (Supplementary Fig. 3h). The in vitro kinase assay also showed that 1NM-PP1 directly inhibited Tpk2-catalyzed Jhd2 phosphorylation (Fig. 2j, lanes 4–5). All these data indicate that Tpk2 directly phosphorylates Jhd2 at S321 and S340.

As PKA activity is upregulated by increasing glucose concentrations, we thus examined the status of Jhd2 phosphorylation when cells were grown in medium containing 0.05%, 0.5%, and 2% glucose. Jhd2 phosphorylation was elevated when the concentration of glucose was increased from 0.05 to 2% in WT but not in Jhd2-S321A, S340A and *tpk2Δ* mutants (Fig. 2k, l), indicating that Tpk2 phosphorylates Jhd2 at S321 and S340 in response to glucose availability.

We also examined the subcellular localization of phosphorylated Jhd2 by immunofluorescence with anti-Jhd2S321p and anti-Jhd2S340p antibodies. Although Jhd2 distributed throughout cells, the strong Jhd2 phosphorylation signals were observed in the cytoplasm (Fig. 2m), indicating that the phosphorylated Jhd2 is primarily localized in the cytoplasm. Jhd2 has a bipartite nucleus localization signal (NLS) within the sequence ₃₁₃RLLPARKLSIDELEEMFWSLVTKNRRSS₃₄₀. As

S321 and S340 are located within this NLS sequence, we wondered whether phosphorylation of Jhd2 may affect the entry of Jhd2 into the nucleus. By subcellular fractionation of WT and Jhd2-S321A, S340A mutant, we clearly observed that more Jhd2 was localized in the nucleus in Jhd2-S321A, S340A mutant (Supplementary Fig. 4a). Consistently, more Jhd2 was found in the nucleus with granular distribution in Jhd2-S321A, S340A and *tpk2Δ* mutants by immunofluorescence (Supplementary Fig. 4b). Together, these results indicate that Tpk2 can directly phosphorylate Jhd2, which impairs the nucleus localization of Jhd2.

## Tpk2 phosphorylates Jhd2 to inhibit its H3K4 demethylase activity

To determine whether Tpk2 promotes H3K4me3 by phosphorylating Jhd2, we examined the effect of Jhd2 phosphorylation on histone demethylase activity of Jhd2. We first examined global H3K4me3 in WT, Jhd2-S321A, Jhd2-S340A and Jhd2-S321A, S340A mutants by Western blots. The levels of H3K4me3 were marginally reduced in Jhd2-S321A and Jhd2-S340A mutants but were dramatically reduced in Jhd2-S321A, S340A double mutant (Fig. 3a). We also constructed Jhd2 phosphomimic mutants, Jhd2-S321D, Jhd2-S340D, Jhd2-S321D, S340D and found that the global levels of H3K4me3 were increased in these Jhd2 phosphomimic mutants (Fig. 3a). The intracellular H3K4me3 was significantly reduced by 1NM-PP1 in Tpk2-as mutant but not Tpk1-as and Tpk3-as mutants (Supplementary Fig. 4c), suggesting that the kinase activity of Tpk2 is required to promote H3K4me3. Loss of Tpk2 significantly reduced the global levels of H3K4me3; deletion of *TPK2* in Jhd2-S321A, S340A mutant did not further reduce H3K4me3 and deletion of *TPK2* in Jhd2-S321D, S340D mutant did not further increase H3K4me3 (Fig. 3b; Supplementary Fig. 4d), suggesting that Tpk2 regulates H3K4me3 by phosphorylating Jhd2 at S321 and S340.

To directly examine the effect of Tpk2-mediated Jhd2 phosphorylation on Jhd2 activity, we performed in vitro demethylase assay with purified WT Jhd2 and Jhd2-S321A, S340A using the synthesized H3 (1–23) peptide that contains H3K4me3 as the substrate. The in vitro demethylase assay showed that Jhd2-S321A, S340A reduced H3K4me3 to a lower level than WT Jhd2 (Fig. 3c, lane 3 vs lane 2), indicating that Jhd2-S321A, S340A has a higher demethylase activity than WT Jhd2. We also purified unphosphorylated Jhd2 from *tpk2Δ* mutant (Jhd2 *tpk2Δ*) and performed in vitro demethylase assay. Compared with Jhd2 purified from WT cells (WT Jhd2), Jhd2 *tpk2Δ* displayed higher H3K4 demethylase activity (Fig. 3d, lane 3 vs lane 2). We also treated WT Jhd2 with λ phosphatase to remove its phosphorylation and then used this dephosphorylated Jhd2 for in vitro demethylase assay. The dephosphorylated Jhd2 reduced H3K4me3 to a lower level than untreated Jhd2 (Fig. 3e, lane 4 vs lane 3), indicating that phosphorylation of Jhd2 inhibits its demethylase activity. Furthermore, we purified Jhd2 from cells grown in increasing glucose concentrations (0.05%, 0.5%, 2%) and performed in vitro demethylase assay. Jhd2 purified from 2% glucose-containing medium had higher phosphorylation level but lower demethylase activity when compared with Jhd2 purified from cells grown in 0.05% glucose-containing medium (Fig. 3f, lanes 2–4).

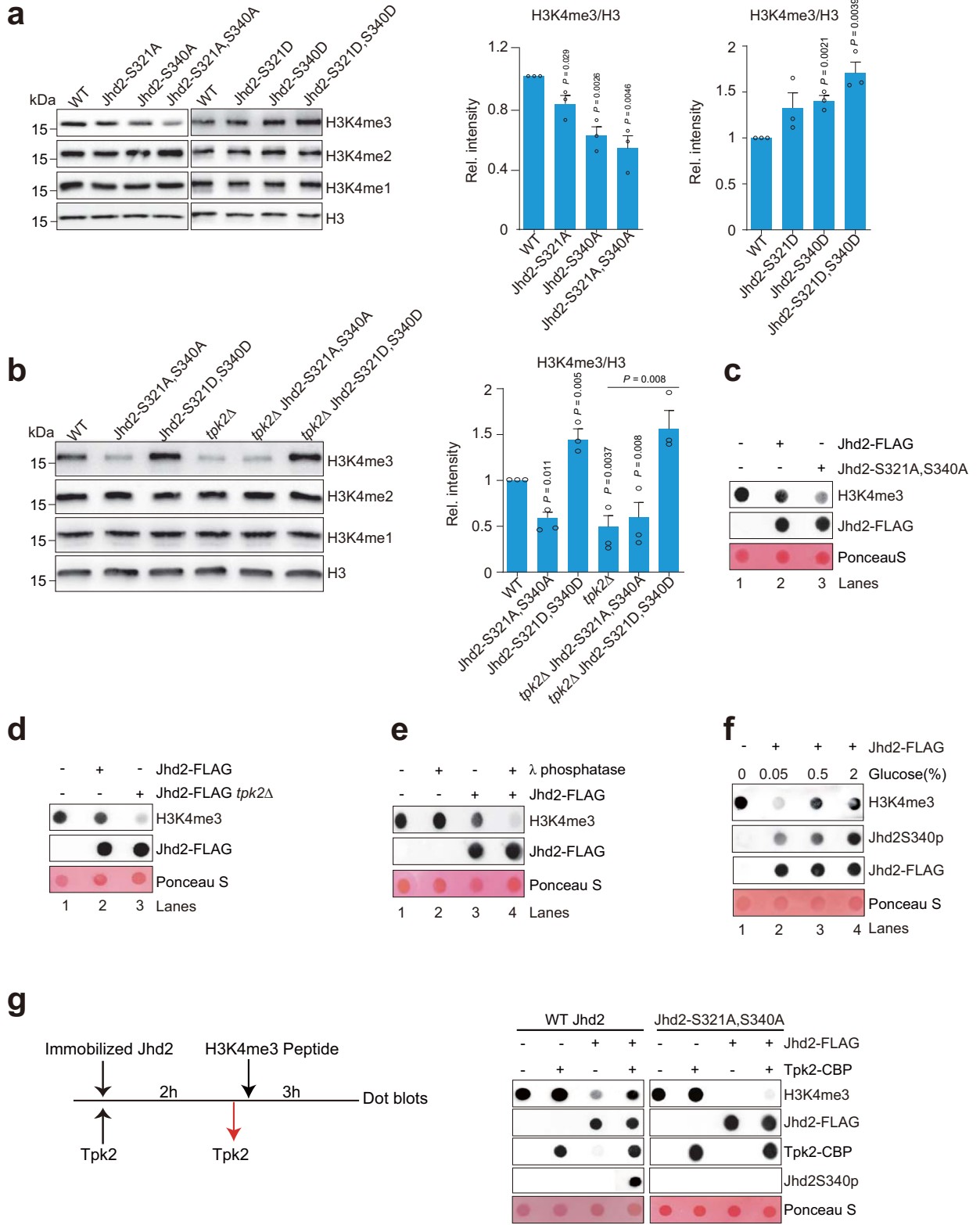

**Fig. 3 | Tpk2-catalyzed Jhd2 phosphorylation inhibits the H3K4 demethylase activity of Jhd2. a** Western blot analysis of H3K4 methylation in WT, Jhd2-S321A, Jhd2-S340A, Jhd2-S321A, S340A, Jhd2-S321D, Jhd2-S340D, and Jhd2-S321D, S340D mutants. **b** Western blot analysis of H3K4 methylation in WT, Jhd2-S321A, S340A, Jhd2-S321D, S340D, *tpk2Δ*, *tpk2Δ* Jhd2-S321A, S340A, and *tpk2Δ* Jhd2-S321D, S340D mutants. **c, d** Analysis the activity of Jhd2 purified from WT (Jhd2-FLAG), Jhd2-S321A, S340A (Jhd2-S321A, S340A-FLAG), and *tpk2Δ* (Jhd2-FLAG *tpk2Δ*) on H3K4me3 (1–23) peptide by in vitro histone demethylase assay.

**e** Dephosphorylation of Jhd2 by λ phosphatase enhanced its histone demethylase activity. **f** The histone demethylase activity of Jhd2 was reduced when cells grown in YP medium containing increasing glucose concentrations. **g** In vitro phosphorylation of Jhd2 by purified Tpk2 reduced the histone demethylase activity of WT Jhd2 but not Jhd2-S321A, S340A. For **a, b**, data represent the mean ± SEM of three biological independent experiments. Two-sided t-tests were used for statistical analysis. Source data are provided as a Source data file.

To directly show that Tpk2 inhibits Jhd2 activity by phosphorylating Jhd2, we immobilized the purified dephosphorylated Jhd2-FLAG on anti-FLAG agarose beads, and then incubated it with purified Tpk2 to phosphorylate Jhd2. After phosphorylation, Tpk2 was then removed and the demethylase activity of the bound Jhd2 was determined (Fig. 3g). Compared with unphosphorylated Jhd2, Tpk2-phosphorylated Jhd2 only slightly reduced H3K4me3 (Fig. 3g, lane 3 vs lane 4), indicating Tpk2 phosphorylates Jhd2 and impairs its demethylase activity. We also performed similar experiment with Jhd2-S321A, S340A mutant. Compared to WT Jhd2, Jhd2-S321A, S340A displayed higher activity towards H3K4me3 even when treated with Tpk2 (Fig. 3g), indicating that Tpk2 phosphorylates Jhd2 at S321 and S340 to inhibit its demethylase activity.

## Tpk2 phosphorylates Jhd2 to reduce its binding to chromatin

To understand how Tpk2-catalyzed Jhd2 phosphorylation inhibits its demethylase activity, we investigated whether phosphorylation of Jhd2 affects its binding to histones. We performed the peptide pulldown assay by incubating purified WT Jhd2 and Jhd2-S321A, S340A with the biotinylated unmodified H3 (1–23), or H3K4 trimethylated (H3K4me3) (1–23) peptides. Compared with WT Jhd2, more Jhd2-S321A, S340A was pulled down by both unmodified H3 (1–23) and H3K4me3 (1–23) peptides (Fig. 4a; Supplementary Fig. 5a, lane 4 vs lane 5), suggesting that phosphorylation of Jhd2 interferes with its binding to histone substrate. By using isothermal titration calorimetry (ITC) with purified Jhd2 (WT Jhd2, Jhd2-S321A, S340A) and H3 (1–23) peptide, we confirmed that Jhd2-S321A, S340A has a higher binding affinity to H3 (1–23) peptide compared to WT Jhd2 (Fig. 4b). H3K14 acetylation (H3K14ac) has been reported to prevent the binding of Jhd2 to chromatin[44]. H3K14ac indeed reduced the binding of WT Jhd2 to H3 (1–23) peptide (Supplementary Fig. 5b, lane 3 vs lane 4); however, H3K14ac had little effect on the binding of Jhd2-S321A, S340A to H3 (1–23) peptide (Supplementary Fig. 5b, lane 5 vs lane 6), suggesting that loss of Jhd2 phosphorylation derepresses the inhibitory effect of H3K14ac on Jhd2 binding to chromatin.

We then examined the effect of Jhd2 phosphorylation on its interaction with nucleosomes by Co-IP assay. Jhd2 was immunoprecipitated from the cell extracts of WT Jhd2 and Jhd2-S321A, S340A mutant, whose chromatin was digested by Micrococcal Nuclease (MNase) to nucleosomes. The bound nucleosomes were detected by Western blots with anti-H3 and anti-H4 antibodies. Our data showed that the Co-IPed nucleosomes were increased in Jhd2-S321A, S340A mutant (Fig. 4c). We also performed in vitro Co-IP by incubating purified WT Jhd2 and Jhd2-S321A, S340A with in vitro assembled recombinant nucleosomes. Compared with WT Jhd2, more nucleosomes were co-IPed with Jhd2-S321A, S340A (Fig. 4d). Subcellular fractionation also showed more Jhd2-S321A, S340A in the chromatin-bound fractions than WT Jhd2 (Fig. 4e, lane 5 vs lane 6), further confirming that Jhd2 phosphorylation inhibits its binding to chromatin.

Next, we performed ChIP-seq (chromatin immunoprecipitation combined with high-throughput sequencing) to examine the effect of Jhd2 phosphorylation on its genome-wide occupancy. Analysis of ChIP-seq data for WT Jhd2 and Jhd2-S321A, S340A showed that the overall binding of Jhd2 at chromatin was significantly higher in Jhd2-S321A, S340A mutant than that of WT Jhd2 (Fig. 4f–h; Supplementary Fig. 5c, d). By comparing the ChIP-seq data for WT Jhd2 and Jhd2-S321A, S340A, we identified 495 new binding genes for Jhd2-S321A, S340A in addition to 614 genes co-occupied by WT Jhd2 and Jhd2-S321A, S340A (Fig. 4i). Further analysis showed that Jhd2-S321A, S340A occupancy was increased at 614 co-binding genes and 495 Jhd2-S321A, S340A specific binding genes (Fig. 4g, j). Kyoto Encyclopedia of Genes and Genomes (KEGG) analysis revealed that these 1109 genes were enriched in meiosis, phagosome, longevity regulating pathway, proteasome and autophagy pathways (Supplementary Fig. 5e). We also performed ChIP-seq to examine the genome-wide occupancy of

H3K4me3 in WT Jhd2 and Jhd2-S321A, S340A. Consistent with increased Jhd2 binding, the overall H3K4me3 was significantly reduced in Jhd2-S321A, S340A mutant (Fig. 4j–m; Supplementary Fig. 5f, g). H3K4me3 was reduced at 614 co-binding genes and 495 Jhd2-S321A, S340A specific binding genes (Fig. 4j, m). The H3K4me3 enrichment at chromatin anti-correlates with Jhd2-S321A, S340A occupancy (Fig. 4n).

To confirm the above ChIP-seq data, we performed ChIP-qPCR to examine the occupancy of Jhd2 at 3 target genes, *MPO1*, *PMA1* and *YEF3* in WT Jhd2, Jhd2-S321A, S340A and *tpk2Δ* mutants. The Jhd2 occupancy at these three genes was significantly higher in Jhd2-S321A, S340A and *tpk2Δ* mutants than WT Jhd2 (Fig. 4o). Accordingly, H3K4me3 enrichment at these three genes was significantly lower in Jhd2-S321A, S340A and *tpk2Δ* mutants than WT Jhd2 (Fig. 4p). Collectively, these data indicate that Tpk2-mediated phosphorylation of Jhd2 reduces its binding to chromatin and enhances H3K4me3 enrichment at chromatin.

## Tpk2-catalyzed Jhd2 phosphorylation promotes the degradation of Jhd2 by the proteasome

In Fig. 4e, we noticed that the total amount of Jhd2 was significantly higher in Jhd2-S321A, S340A than WT Jhd2 (Fig. 4e, lane 1 vs lane 2). We thus examined whether Tpk2-catalyzed Jhd2 phosphorylation affects the expression of Jhd2. The protein level of Jhd2 was significantly higher in Jhd2-S321A, S340A and *tpk2Δ* mutants than WT Jhd2 and the steady state mRNA levels of *JHD2* were similar in WT, Jhd2-S321A, S340A and *tpk2Δ* mutants (Fig. 5a; Supplementary Fig. 6a). To examine whether Jhd2 phosphorylation affects Jhd2 protein stability, we pre-treated WT Jhd2 and Jhd2-S321A, S340A mutant with cycloheximide (CHX) to block protein synthesis. WT Jhd2 had a shorter half-life than Jhd2-S321A, S340A (Fig. 5b). Consistent with these findings, loss of Tpk2 (*tpk2Δ*) significantly increased the half-life of Jhd2 (Fig. 5b). These results indicate that Tpk2-catalyzed Jhd2 phosphorylation reduces Jhd2 protein stability. As Jhd2 phosphorylation sites (S321, S340) are located between PHD and JmjC domains (Fig. 2e), our data is hence consistent with the study by Huang and colleagues that deletion of the region between PHD and JmjC domains leads to Jhd2 protein instability[45].

To determine how Tpk2-mediated Jhd2 phosphorylation regulates Jhd2 stability, we treated WT Jhd2 and Jhd2-S321A, S340A cells with MG132, the proteasome inhibitor. MG132 treatment significantly increased WT Jhd2 protein levels but had no significant effect on Jhd2-S321A, S340A protein level (Fig. 5c). We also examined Jhd2 in a temperature-sensitive proteasome-deficient mutant (*cim3-1*), which is defective in ATPase activity of 26 S proteasome. Inactivation of Cim3 in *cim3-1* mutant at 30 °C but not 26 °C significantly increased Jhd2 protein level (Supplementary Fig. 6b), which is consistent with the report that Jhd2 is ubiquitinated and degraded by the proteasome pathway[28]. Moreover, inactivation of Cim3 in *cim3-1* mutant increased Jhd2 to a level similar to that in Jhd2-S321A, S340A mutant and inactivation of Cim3 in *cim3-1* Jhd2-S321A, S340A mutant did not further increase Jhd2 protein level (Fig. 5d), indicating that Jhd2 phosphorylation promotes Jhd2 degradation by the proteasome pathway.

## Tpk2-catalyzed Jhd2 phosphorylation promotes Not4-mediated Jhd2 ubiquitination and its turnover by proteasome

We then examined the relationship between Jhd2 phosphorylation and Jhd2 ubiquitination. We first examined the ubiquitination status of Jhd2 in WT, Jhd2-S321A, S340A and *tpk2Δ* mutants. Jhd2 was immunoprecipitated from WT, Jhd2-S321A, S340A and *tpk2Δ* mutants followed by Western blots with anti-ubiquitin (FK2) antibody. Consistent with changes of Jhd2 protein levels, the ubiquitination level of WT Jhd2 was higher than that of Jhd2 immunoprecipitated from Jhd2-S321A, S340A and *tpk2Δ* mutants (Fig. 5e, lane 1 vs lane 2, lane 5 vs lane 6), suggesting that Jhd2 is degraded by the polyubiquitination-dependent proteasome pathway.

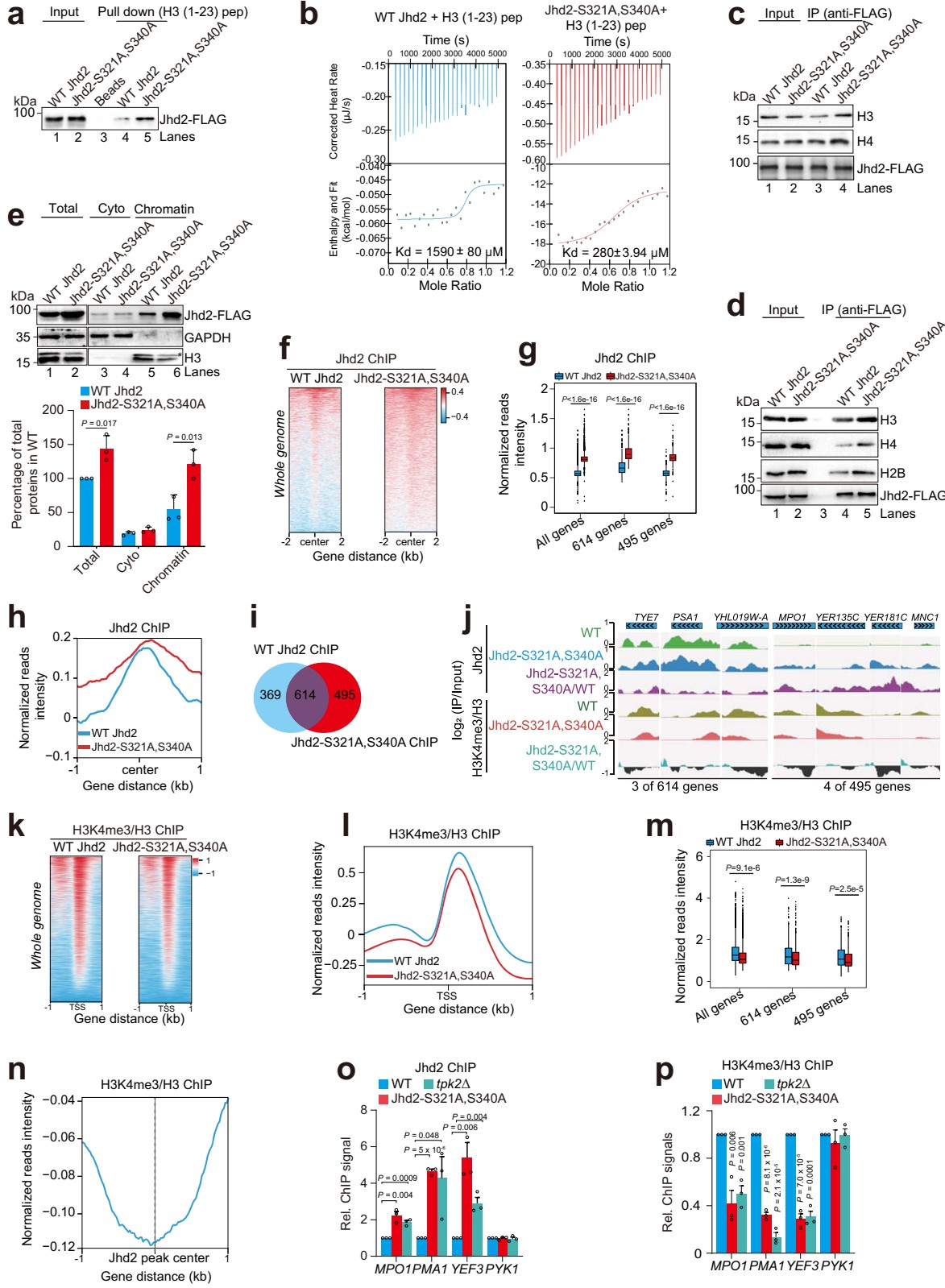

Not4 has been reported to function as an E3 ubiquitin ligase to polyubiquitinate Jhd2 and promote Jhd2 degradation by the proteasome[28]. Our Co-IP assay showed that Not4 interacted with Jhd2 (Supplementary Fig. 6c). Loss of Not4 reduced Jhd2 polyubiquitination and increased Jhd2 protein level to those in Jhd2-S321A, S340A and *tpk2Δ* mutants (Fig. 5e, lanes 1–3, lanes 5–7). Deletion of *NOT4* in Jhd2-S321A, S340A and *tpk2Δ* mutants did not

further reduce ubiquitination of Jhd2 (Fig. 5e). Consistent with increased Jhd2, H3K4me3 was significantly reduced upon loss of Not4 (Fig. 5f, lane 1 vs lane 4). Deletion of *NOT4* did not further reduce H3K4me3 in Jhd2-S321A, S340A and *tpk2Δ* mutants (Fig. 5f, lanes 4–6), suggesting that phosphorylation of Jhd2 facilitates Not4-catalyzed Jhd2 polyubiquitination and proteasome-mediated degradation.

**Fig. 4 | Tpk2-catalyzed Jhd2 phosphorylation reduces its binding at chromatin.**
**a** Peptide pull-down assay showing Jhd2-S321A, S340A had a higher binding affinity to H3 (1–23) peptide than WT Jhd2. **b** ITC assay showing Jhd2-S321A, S340A had a higher binding affinity towards H3 (1–23) peptide than WT Jhd2. The dissociation constant (Kd) for WT Jhd2 and Jhd2-S321A, S340A are 1,590 ± 80 μM and 280 ± 3.94 μM, respectively. **c**, **d** Co-IP assay showing Jhd2-S321A, S340A binds more nucleosomes than WT Jhd2. **e** Effects of Jhd2-S321A, S340A mutation on total Jhd2, cytoplasmic Jhd2 and chromatin-bound Jhd2. **f** Heatmap showing the genome-wide occupancy of Jhd2 in WT Jhd2 and Jhd2-S321A, S340A mutant. **g** Boxplots showing the genome-wide occupancy of Jhd2 in WT Jhd2 and Jhd2-S321A, S340A. **h** Averaged metagene profiles of Jhd2 binding in WT Jhd2 and Jhd2-S321A, S340A. Center was defined as Jhd2 binding peaks. **i** Venn diagram showing the overlap binding sites of WT Jhd2 and Jhd2-S321A, S340A. **j** ChIP-seq tracks of the enrichment of Jhd2 and H3K4me3/H3 at representative genes in WT and Jhd2-S321A, S340A mutant.

**k** Heatmap showing the genome-wide occupancy of H3K4me3/H3 in WT Jhd2 and Jhd2-S321A, S340A mutant. **l** Averaged metagene profiles of H3K4me3/H3 in WT Jhd2 and Jhd2-S321A, S340A. **m** Boxplots showing Jhd2-S321A, S340A mutation reduced the genome-wide enrichment of H3K4me3/H3. **n** Averaged distribution of H3K4me3/H3 around the peaks of Jhd2-S321A, S340A. Log₂ ratios of H3K4me3 versus H3 at all windows are plotted. **o**, **p** ChIP-qPCR analysis of the relative occupancy of Jhd2 (o) and H3K4me3/H3 (p) at representative genes in WT, Jhd2-S321A, S340A, and *tpk2Δ* mutants. *PMA1* and *YEF3* were chosen from 614 co-binding genes identified. *MPO1* was reported to be regulated by Jhd2[29]. The primers were designed for the ORF regions of indicated genes. For **e**, **o**–**p**, data represent the mean ± SE of three biological independent experiments. For **g**, **m**, centre lines denote medians; box limits denote 25th and 75th percentiles; whiskers denote maxima and minima. Two-sided Wilcoxon test in R (package ggpval) was used for statistical analysis. Source data are provided as a Source data file.

To gain a mechanistic insight into Jhd2 phosphorylation-dependent degradation, we examined the effect of Jhd2 phosphorylation on the interaction between Not4 and Jhd2. We purified Not4 by tandem affinity purification (TAP) and incubated it with FLAG-purified WT Jhd2 and Jhd2-S321A, S340A. The in vitro Co-IP showed that the interaction between Not4 and Jhd2-S321A, S340A was remarkably reduced when compared with the interaction between Not4 and WT Jhd2 (Fig. 5g, lane 4 vs lane 3). To directly determine whether Tpk2-catalyzed Jhd2 phosphorylation enhanced the interaction between Jhd2 and Not4, we first immobilized dephosphorylated Jhd2 on anti-FLAG beads and then incubated it with purified Tpk2 to phosphorylate Jhd2. After removing Tpk2, the immobilized phosphorylated Jhd2 was then incubated with purified Not4. As a control, immobilized Jhd2 with no Tpk2 was used. The in vitro Co-IP assay showed that Tpk2-catalyzed Jhd2 phosphorylation enhanced the interaction between Not4 and Jhd2 (Fig. 5h, lane 1 vs lane 2). We also performed the same experiment with Jhd2-S321A, S340A and found that Tpk2 had no effect on the interaction between Jhd2-S321A, S340A and Not4 (Fig. 5h, lane 3 vs lane 4). These data indicate that Tpk2-catalyzed Jhd2 phosphorylation promotes its interaction with Not4, which increases Jhd2 poly-ubiquitination and subsequent turnover by the proteasome (Fig. 5i).

## Tpk2 promotes H3K14ac by phosphorylating Jhd2 and reducing its interaction with Rpd3

In Fig. 1a, we noticed that glucose induced H3K14ac in addition to H3K4me3 (Fig. 1a). We therefore examined the effect of Ras-cAMP-PKA(Tpk2) pathway on H3K14ac. H3K14ac was significantly reduced in *ras2Δ* and *tpk2Δ* mutants and increased in *pde1Δ* and *pde2Δ* mutants (Fig. 6a; Supplementary Fig. 7a, b). Loss of Tpk1 also slightly reduced H3K14ac in WT but not *tpk2Δ* mutant (Supplementary Fig. 7b). Moreover, loss of Jhd2 restored the H3K14ac in *tpk2Δ* mutant (Supplementary Fig. 7c), suggesting that Tpk2 maintains H3K14ac partly if not all by antagonizing Jhd2. Consistent with H3K4me3 and Jhd2 changes, H3K14ac was significantly reduced in Jhd2-S321A, S340A and loss of Tpk2 in Jhd2-S321A, S340A did not further reduce H3K14ac (Fig. 6b; Supplementary Fig. 7d). Moreover, loss of *NOT4* reduced H3K14ac (Fig. 6c), suggesting that Tpk2-catalyzed Jhd2 phosphorylation regulates H3K14ac in addition to H3K4me3.

To explore the mechanism underlying Jhd2 phosphorylation-regulated H3K14ac, we first examined the transcription of HATs (histone acetyltransferase) and HDAC (histone deacetylase) for H3K14ac, including Gcn5, Sas3 and Rpd3. The transcription of these three genes was not significantly changed in Jhd2-S321A, S340A mutant (Supplementary Fig. 7e). We then immunoprecipitated WT Jhd2 and Jhd2-S321A, S340A and LC-MS analysis revealed that Rpd3 was a Jhd2-interacting protein (Fig. 6d). Co-IP assays showed that more Rpd3 interacted with Jhd2-S321A, S340A than WT Jhd2 (Fig. 6e), suggesting that Jhd2-S321A, S340A has a preference to bind Rpd3. Meanwhile, more Rpd3 was co-IPed with Jhd2 in *tpk2Δ* mutant (Fig. 6f). To verify our hypothesis that Tpk2-catalyzed Jhd2 phosphorylation reduces the interaction between Jhd2 and Rpd3, we first phosphorylated purified Jhd2 on anti-FLAG beads with purified Tpk2 and then incubated it with Rpd3. The in vitro Co-IP assay showed that phosphorylation of Jhd2 by Tpk2 decreased the interaction between Rpd3 and Jhd2 (Fig. 6g, lane 1 vs lane 3), whereas the interaction between Jhd2-S321A, S340A and Rpd3 was unaffected by Tpk2 (Fig. 6g, lane 2 vs lane 4). Although Jhd2 is subject to degradation by Not4-mediated proteasome pathway, Rpd3 remained unchanged in WT and *not4Δ* mutant (Fig. 5f).

Given the interaction between Jhd2 and Rpd3, we purified Jhd2 from WT Jhd2 and Jhd2-S321A, S340A mutant and performed in vitro deacetylase assay on H3K14ac peptide. The purified WT Jhd2 showed little histone deacetylase activity, while the purified Jhd2-S321A, S340A had higher deacetylase activity, consistent with more Rpd3 co-IPed with Jhd2 (Fig. 6h). Similar results were observed for Jhd2 purified from *tpk2Δ* (Jhd2 *tpk2Δ*) mutant (Fig. 6h). These data indicate that Tpk2 phosphorylates Jhd2 to interfere with its interaction with Rpd3.

We also examined the amount of Rpd3 within the cytoplasm and chromatin-bound fractions in WT Jhd2 and Jhd2-S321A, S340A mutant by subcellular fractionation. Jhd2-S321A, S340A had more Rpd3 in the chromatin-bound fractions than WT Jhd2 (Fig. 6i). ChIP-seq also revealed that the overall binding of Rpd3 at chromatin was significantly higher in Jhd2-S321A, S340A mutant than that in WT Jhd2 (Fig. 6j, k). In addition, we found that similar to Jhd2 ChIP-seq data in Jhd2-S321A, S340A mutant, the Rpd3 occupancy in Jhd2-S321A, S340A mutant showed a broad and higher binding pattern (Fig. 6j, k; Supplementary Fig. 7f, g). Both Jhd2 and Rpd3 from Jhd2-S321A, S340A mutant co-occupied at 200 genes (Fig. 6l). Among these 200 genes, 55 genes are unique for Jhd2 in WT cells, 16 genes are unique for Rpd3 in WT cells, 100 genes are co-occupied by Jhd2 and Rpd3 in WT cells. As expected, the Rpd3 occupancy at 200 co-occupied genes was significantly higher in Jhd2-S321A, S340A than WT (Fig. 6m, n). KEGG analysis revealed that these 200 co-occupied genes are enriched in metabolic pathways, ribosome, longevity regulating pathway, oxidative phosphorylation, peroxisome and so on (Supplementary Fig. 7h). By ChIP-qPCR, we confirmed the increased Rpd3 occupancy at 2 representative genes (*NCS2*, *EGR28*) in Jhd2-S321A, S340A and *tpk2Δ* mutants (Fig. 6o), suggesting that Tpk2-mediated phosphorylation of Jhd2 attenuates its interaction with Rpd3 as well as the binding of Rpd3 at chromatin. Rpd3 has been reported to be recruited by Pho23 and Rxt1 to repress the expression of *PHO5* under high phosphate conditions[46] and repress the transcription of galactose-repressed genes, such as *REI1*, *TEA1* and *RRN11*[47]. However, loss of Pho23 and Rxt1 had no significant effect on Rpd3 occupancy at *NCS2* and *EGR28* (Supplementary Fig. 7i), suggesting that Rpd3 is recruited to different subsets of genes by distinct mechanisms.

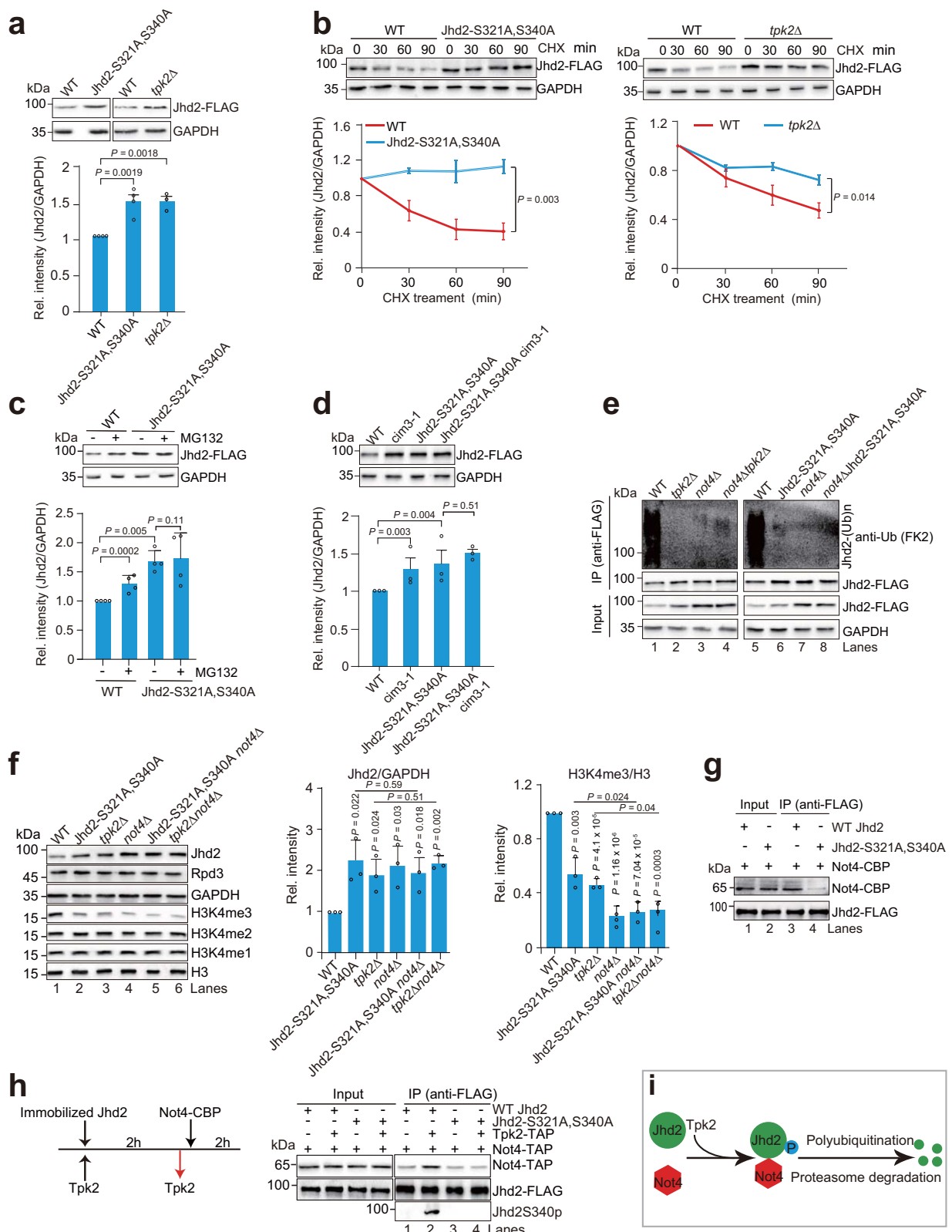

To show that the reduced H3K14ac in Jhd2-S321A, S340A and *tpk2Δ* mutants is caused by Rpd3, we examined H3K14ac in Jhd2-S321A, S340A *rpd3Δ* and *tpk2Δ rpd3Δ* mutants. Loss of Rpd3 rescued the reduced H3K14ac in both Jhd2-S321A, S340A and *tpk2Δ* mutants (Fig. 6p). Moreover, H3K14ac was increased in response to glucose in WT but not in Jhd2-S321A, S340A and *tpk2Δ* mutants (Fig. 6q). H3K14ac remained relatively high and constant in Jhd2-

S321A, S340A *rpd3Δ* and *tpk2Δ rpd3Δ* mutants even in low concentration of glucose (0.05%) (Fig. 6q), suggesting that Jhd2 phosphorylation maintains H3K14ac in part by antagonizing Rpd3 to in response to glucose. Together, these data indicate that Tpk2-catalyzed Jhd2 phosphorylation interferes with the binding of both Jhd2 and Rpd3 at chromatin to maintain normal levels of H3K4me3 and H3K14ac.

**Fig. 5 | Tpk2-catalyzed Jhd2 phosphorylation enhances Not4-mediated Jhd2 ubiquitination and its turnover by the proteasome pathway. a** Western blot analysis of Jhd2 in WT, Jhd2-S321A, S340A, and *tpk2Δ* mutants. **b** Analysis of Jhd2 stability in WT, Jhd2-S321A, S340A, and *tpk2Δ* mutants. **c**, **d** Analysis of the effect of MG132 or inactivation of Cim3 on Jhd2 protein levels. **e** Analysis of the ubiquitination of Jhd2 in WT, Jhd2-S321A, S340A, *not4Δ*, *tpk2Δ*, *not4Δ* Jhd2-S321A, S340A, and *not4Δ tpk2Δ* mutants. **f** Analysis of Jhd2, Rpd3 and H3K4 methylation in WT, Jhd2-S321A, S340A, *tpk2Δ*, *not4Δ*, Jhd2-S321A, S340A *not4Δ*,

and *tpk2Δ not4Δ* mutants. **g** Co-IP assay showing Jhd2-S321A, S340A mutation reduced the interaction between Jhd2 and Not4. **h** Tpk2-catalyzed Jhd2 phosphorylation increased the interaction between Not4 and Jhd2. **i** Proposed model showing that Tpk2-catalyzed Jhd2 phosphorylation promotes Not4-mediated ubiquitination and subsequent proteasome degradation of Jhd2. For **a**–**d**, **f**, data represent the mean ± SEM of at least three biological independent experiments. Two-sided t-tests were used for statistical analysis. Source data are provided as a Source data file.

## Tpk2-catalyzed Jhd2 phosphorylation regulates gene expression and chronological life span

As it has been established that Tpk2-catalyzed phosphorylation of Jhd2 regulates Jhd2 stability and demethylase activity, we next examined their effects on gene expression by performing RNA-seq for WT and Jhd2-S321A, S340A mutant. 343 genes were downregulated and 137 genes were significantly upregulated in Jhd2-S321A, S340A mutant (fold-change ≥1.5, fold-change ≤0.66; *P* < 0.05) (Supplementary Fig. 8a). KEGG analysis revealed these genes were enriched in fatty acid degradation, peroxisome, meiosis, cell cycle, longevity regulating pathway and autophagy pathways (Supplementary Fig. 8b, c). By comparing our RNA-seq data to Jhd2 and H3K4me3 ChIP-seq data, we found that for 480 differentially expressed genes by Jhd2-S321A, S340A, the occupancy of Jhd2 was significantly increased in Jhd2-S321A, S340A when compared with WT cells (Supplementary Fig. 8d). Consistently, H3K4me3 enrichment at these 480 genes was significantly reduced in Jhd2-S321A, S340A mutant (Supplementary Fig. 8d), suggesting that phosphorylation of Jhd2 could regulate the transcription of these genes by inhibiting Jhd2-mediated H3K4 demethylation.

By comparing the transcriptome data for Jhd2-S321A, S340A and *tpk2Δ*, we found that among 343 Jhd2-S321A, S340A-repressed genes, 100 (29.2%) genes were downregulated in *tpk2Δ* mutant (Fig. 7a). Among 137 Jhd2-S321A, S340A-induced genes, 26 (19.0%) genes were upregulated in *tpk2Δ* mutant (Fig. 7a). KEGG analysis revealed that these genes were enriched in non-homologous end-joining, peroxisome, fatty acid degradation, longevity regulating pathway, MAPK signaling pathway, ubiquitination-mediated proteolysis pathways (Supplementary Fig. 8e). Gene set enrichment analysis (GSEA) revealed that Tpk2-catalyzed Jhd2 phosphorylation significantly regulates the transcription of genes involved in longevity regulation pathway (Fig. 7b). In Jhd2-S321A, S340A mutant, the occupancy of Jhd2 at longevity regulating genes was significantly increased and H3K4me3 was significantly reduced (Supplementary Fig. 8f). We thus examined the effect of Tpk2-catalyzed Jhd2 phosphorylation on chronological life span. Consistent with downregulation of most longevity regulating genes (Fig. 7b), the chronological life span of *tpk2Δ* and Jhd2-S321A, S340A mutants was significantly shorter than WT (Fig. 7c). Together, these data show that Tpk2-catalyzed Jhd2 phosphorylation regulates gene expression and maintains normal chronological life span.

## Tpk2-catalyzed Jhd2 phosphorylation promotes autophagy in part by inhibiting Rpd3 binding at autophagy genes

By comparing the transcriptome data for Jhd2-S321A, S340A and *rpd3Δ*, we found that 125 out of 343 (36.4%) Jhd2-S321A, S340A-repressed genes were upregulated and 66 out of 137 (48.2%) Jhd2-S321A, S340A-activated genes were downregulated in *rpd3Δ* mutant (Fig. 7d). KEGG analysis revealed these genes were enriched in meiosis, cell cycle, non-homologous end-joining, autophagy and peroxisome pathways (Supplementary Fig. 9a). GSEA analysis showed that Tpk2-catalyzed Jhd2 phosphorylation and Rpd3 oppositely regulate genes involved in the autophagy pathway (Fig. 7e). ChIP-seq analysis showed that loss of Jhd2 phosphorylation led to increased Jhd2 and Rpd3 occupancy at autophagy (*ATG*) genes in Jhd2-S321A, S340A mutant (Fig. 7f; Supplementary Fig. 9b, c). Using

ChIP-qPCR, we confirmed increased occupancy of Jhd2 and Rpd3 at autophagy genes including *ATG9*, *ATG32* and *ATG39* in Jhd2-S321A, S340A and *tpk2Δ* mutants (Fig. 7g). Loss of Pho23 and Rxt1 had no significant effect on Rpd3 occupancy at *ATG* genes (Supplementary Fig. 9d). These data suggest that Tpk2-mediated Jhd2 phosphorylation inhibits the binding of Jhd2 and Rpd3 at autophagy genes. RNA-seq data showed the transcription of *ATG* genes was reduced in Jhd2-S321A, S340A and *tpk2Δ* mutants (Fig. 7f). Using RT-qPCR, we confirmed that the transcription of *ATG* genes was significantly reduced in Jhd2-S321A, S340A and *tpk2Δ* mutants (Fig. 7h). In contrast, loss of Tpk1 and Tpk3 had no significant effect on the transcription of *ATG* genes (Supplementary Fig. 9e).

RNA-seq data also showed the transcription of *ATG* genes was increased in *rpd3Δ* mutant (Fig. 7f). RT-qPCR also confirmed the increased transcription of autophagy genes in *rpd3Δ* mutant (Fig. 7h), which is consistent with the reports that Rpd3 inhibits autophagy by repressing the expression of autophagy genes[48,49]. Loss of Pho23 had no significant effect on the transcription of *ATG* genes (Supplementary Fig. 9f). Given the opposite effect of Jhd2 phosphorylation and Rpd3 on autophagy gene expression, we hypothesized that Jhd2 phosphorylation may antagonize the binding of Rpd3 at autophagy genes to promote their transcription. We thus examined the transcription of *ATG* genes in WT, *rpd3Δ*, Jhd2-S321A, S340A and Jhd2-S321A, S340A *rpd3Δ* mutants. The *ATG* genes were significantly reduced in Jhd2-S321A, S340A and *tpk2Δ* but was restored in *rpd3Δ* Jhd2-S321A, S340A and *rpd3Δ tpk2Δ* mutants (Fig. 7h).

We then employed a GFP (green fluorescent protein) liberation assay to directly investigate the effect of Tpk2 and Jhd2 phosphorylation on autophagy activity, which detects free GFP that is liberated upon the delivery of endogenous promoter-driven Atg8 with an N-terminal GFP tag (GFP-Atg8) to the vacuole and subsequent degradation of the Atg8 portion of the fusion protein[50]. The GFP liberation assay demonstrated significantly reduced GFP-Atg8 proteolysis in Jhd2-S321A, S340A and *tpk2Δ* mutants compared to their WT counterpart as indicated by decreased ratio of free GFP/GFP-Atg8 (Supplementary Fig. 9g). Loss of Tpk1 and Tpk3 had no significant effect on GFP-Atg8 proteolysis (Supplementary Fig. 9h), consistent with their effect on autophagy gene expression (Supplementary Fig. 9e). Notably, the reduced free GFP/GFP-Atg8 in Jhd2-S321A, S340A and *tpk2Δ* mutants were increased to normal levels in *rpd3Δ* Jhd2-S321A, S340A and *rpd3Δ tpk2Δ* mutants (Fig. 7i). To strengthen these findings, we used a complementary assay by assessing the autophagy-dependent translocation of GFP-Atg8 to the vacuole by fluorescence microscopy. The percentage of autophagic cells that displayed clearly vacuolar localization of GFP was significantly reduced in Jhd2-S321A, S340A and *tpk2Δ* mutants but was significantly increased in *rpd3Δ* Jhd2-S321A, S340A and *rpd3Δ tpk2Δ* mutants (Fig. 7j), consistent with GFP liberation assay results (Fig. 7i). Upon autophagy induction condition such as glucose starvation (SD - C), the autophagy was significantly reduced in Jhd2-S321A, S340A and *tpk2Δ* mutants (Supplementary Fig. 10a). The transcription of autophagy genes was also significantly reduced in Jhd2-S321A, S340A and *tpk2Δ* mutants under glucose starvation condition (Supplementary Fig. 10b), suggesting that Tpk2-catalyzed Jhd2 phosphorylation promotes the transcription of autophagy genes under both basal and induction conditions.

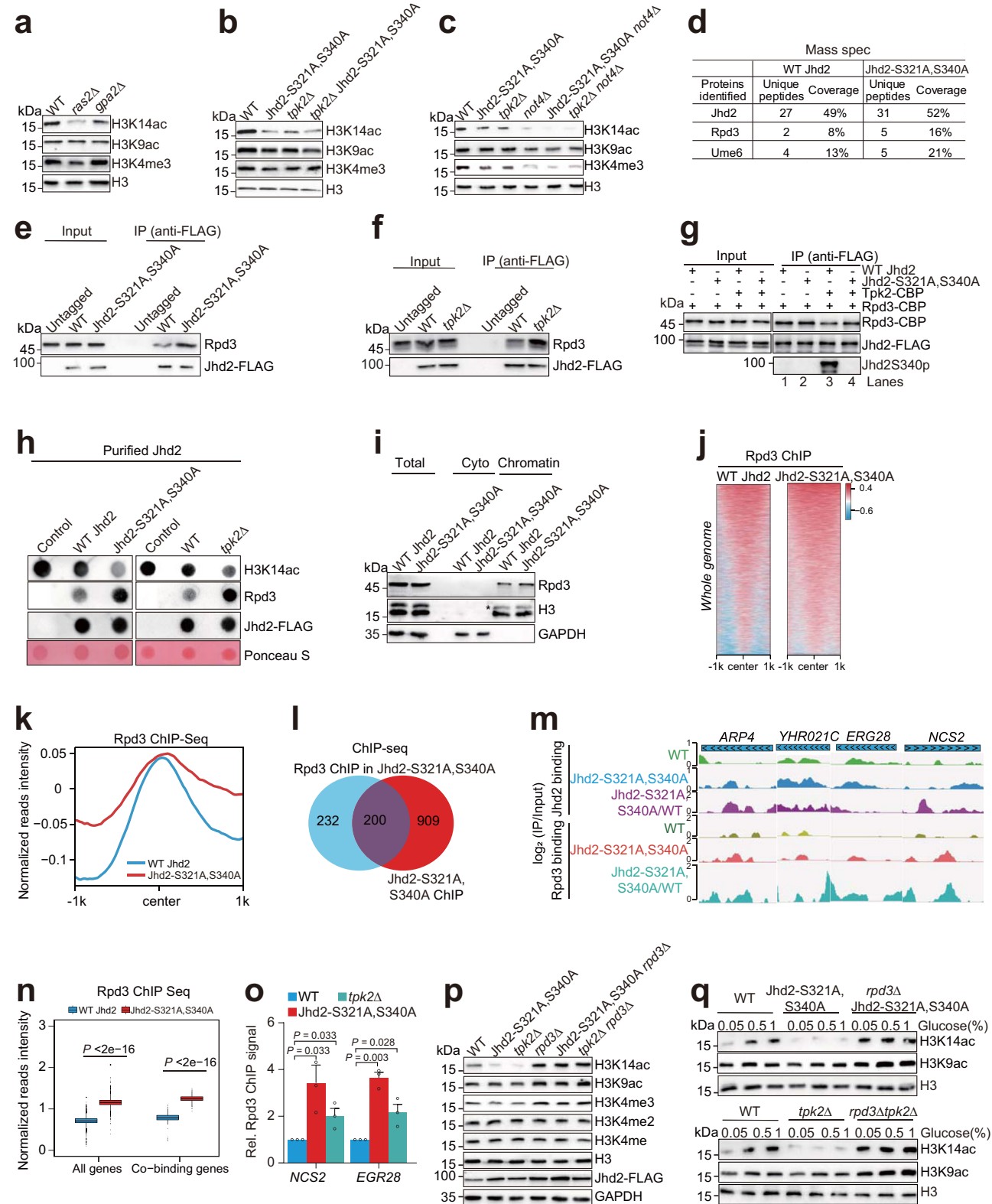

Collectively, these data suggest that Tpk2-catalyzed Jhd2 phosphorylation activates the transcription of autophagy genes and promotes the autophagy pathway, partly if not all, by inhibiting the binding of Rpd3 at autophagy genes (Fig. 7k).

## Discussion

Cells need to coordinate their transcriptional profile and metabolism to adapt to external nutritional changes through the cooperation of

multiple signal molecules including AMPK, mTOR, and cAMP-dependent PKA. However, the mechanism by which these signaling pathways mediate the biological responses to nutrient availability remains largely unknown. The current study demonstrates an intricate link between nutrient/energy sensing, chromatin modifications, longevity and transcriptional regulation of autophagy. We find that the Ras-cAMP-PKA(Tpk2) pathway is activated by glucose to antagonize Jhd2 to modulate histone modifications and gene expression. Tpk2-

**Fig. 6 | Tpk2-catalyzed Jhd2 phosphorylation reduces the interaction between Jhd2 and Rpd3. a** Western blot analysis of histone modifications in WT, *ras2Δ*, and *gpa2Δ* mutants. **b** Analysis of histone modifications in WT, Jhd2-S321A, S340A, *tpk2Δ*, and *tpk2Δ* Jhd2-S321A, S340A mutants. **c** Analysis of histone modifications in WT, Jhd2-S321A, S340A, *tpk2Δ*, *not4Δ*, Jhd2-S321A, S340A *not4Δ*, and *tpk2Δ not4Δ* mutants. **d** LC-MS analysis of proteins co-purified with WT Jhd2 (Jhd2-FLAG) and Jhd2-S321A, S340A (Jhd2-S321A, S340A-FLAG). **e, f** In vivo co-IP analysis of the interaction between Jhd2 and Rpd3 in WT, Jhd2-S321A, S340A and *tpk2Δ* mutants. **g** In vitro co-IP showing Tpk2-catalyzed Jhd2 phosphorylation reduced the interaction between Rpd3 and Jhd2. **h** In vitro histone deacetylase assay showing Jhd2 complex purified from Jhd2-S321A, S340A and *tpk2Δ* mutants interacted with more Rpd3 and had higher deacetylase activity. **i** Effects of Jhd2-S321A, S340A on total Rpd3, cytoplasmic Rpd3 and chromatin-bound Rpd3. **j** Heatmap showing the genome-wide occupancy of Rpd3 in WT Jhd2 and Jhd2-S321A, S340A mutant. **k** The genome-wide occupancy of Rpd3 was higher in

Jhd2-S321A, S340A mutant than WT Jhd2. **l** Venn diagram showing the overlap binding sites of Jhd2 and Rpd3 in Jhd2-S321A, S340A mutant. **m** ChIP-seq tracks of Jhd2 and Rpd3 occupancy in WT Jhd2 and Jhd2-S321A, S340A mutant. **n** Boxplots showing Jhd2-S321A, S340A mutation increased the genome-wide occupancy of Rpd3. **o** ChIP-qPCR analysis of Rpd3 binding at *NCS2* and *EGR28* in WT, Jhd2-S321A, S340A and *tpk2Δ* mutants. **p** Analysis of histone modifications in WT, Jhd2-S321A, S340A, *tpk2Δ*, *rpd3Δ*, Jhd2-S321A, S340A *rpd3Δ*, and *tpk2Δ rpd3Δ* mutants. **q** Analysis of histone modifications in WT, Jhd2-S321A, S340A, *tpk2Δ*, *rpd3Δ* Jhd2-S321A, S340A, and *rpd3Δ tpk2Δ* mutants when grown in 0.05%, 0.5 and 1% glucose. For **o**, data represent the mean ± SEM of three biological independent experiments. For **n**, centre lines denote medians; box limits denote 25th and 75th percentiles; whiskers denote maxima and minima. Two-sided Wilcoxon test in R (package ggpval) was used for statistical analysis. The center in ChIP-seq data of Jhd2 with Rpd3 is coincident with the center of Jhd2-S321A, S340A with Rpd3. Source data are provided as a Source data file.

catalyzed phosphorylation of Jhd2 prevents its nuclear localization, inhibiting its binding to chromatin, and promotes its polyubiquitination and degradation by the proteasome. Moreover, Tpk2-catalyzed phosphorylation of Jhd2 prevents the binding of Rpd3 to chromatin to increase H3K14ac. By repressing the activity of both Jhd2 and Rpd3, Tpk2 regulates gene expression, maintains normal chronological life span and promotes autophagy. Therefore, our study shed lights on how cells rewire their transcriptome by modulating histone modifying enzymes in response to nutrient availability.

Deregulation of histone demethylases may result in devastating consequences such as human cancers or developmental defects[51]. It is thus important to understand how histone demethylases are regulated. In this study, we find that Tpk2 phosphorylates Jhd2 and regulates its nuclear translocation, binding to chromatin and protein stability, which eventually reduce its demethylase activity. The phosphorylated Jhd2 is primarily localized in the cytoplasm and Tpk2-catalyzed Jhd2 phosphorylation hinders the nuclear localization of Jhd2, which could be due to the fact that its phosphorylation sites, S321 and S340 are located within its nucleus localization signal (NLS). Moreover, Jhd2 phosphorylation reduces its binding to chromatin. Notably, we observed Tpk2-catalyzed Jhd2 phosphorylation reduced Jhd2 stability. Jhd2 has been reported to be ubiquitinated by Not4 to maintain proper H3K4me3 and gene expression levels[28]. It is unknown what factor(s) trigger Jhd2 ubiquitination by Not4. Here, we find that Tpk2-catalyzed Jhd2 phosphorylation promotes Jhd2 ubiquitination by Not4. As Tpk2-catalyzed Jhd2 phosphorylation is regulated by glucose availability, our study provides a dynamic mechanism for regulation of Jhd2 activity and stability. The relatively low activity and expression of Jhd2 in rich medium could explain the long-standing question: why does loss of Jhd2 have a mild effect on global H3K4me3 when cells were grown in standard glucose medium? Jhd2 and Not4 are conserved from yeast to mammals and human Not4 can polyubiquitinate JARID1C, the human homolog of Jhd2[28]. It is tempting to hypothesize that JARID1C could be phosphorylated by PKA in mammals.

Glycolysis is required for H3K14ac, mono-ubiquitination of H2B at K123, H3pT11 and H3K4me3[15,16,18,50]. One plausible mechanism by which glycolysis regulates histone modifications is providing cofactors, i.e. acetyl-CoA and S-adenosylmethionine (SAM) for histone modifying enzymes. But this theory cannot explain why only a small subset of histone modifications is affected. It remains to be explored how cells convey the nutrient signal to histone modifications. Our work identified the first histone modifying enzyme that is a direct target for PKA(Tpk2). Blair et al. performed a mass spectrometry analysis of proteins co-purified with Jhd2 and Tpk2 was not detected in their study[52]. This could result from the use of different digestion methods and different mass spectrometers. By applying a combination of techniques, including Co-IP, in vitro kinase assay and a Shokat allele of Tpk2, we confirmed that Tpk2 can phosphorylate Jhd2. Tpk2-catalyzed Jhd2 phosphorylation impairs

its nuclear translocation, inhibits its binding to chromatin and promotes its degradation by the proteasome, which lead to reduced Jhd2 activity and increased H3K4 methylation. We also noted that loss of Tpk1 slightly reduced H3K4me3 and H3K14ac (Fig. 1e; Supplementary Fig. 7b). But compared with Tpk2, Tpk1 had a mild effect on H3K4me3 and H3K14ac. It is possible that Tpk1 may phosphorylate Jhd2 at other sites rather than S321 and S340, which then affects H3K4 demethylation activity of Jhd2. It is also possible that Tpk1 may affect the activity of Set1. By targeting Jhd2, Tpk2 directly connects glucose availability to histone modifications, which enables cells to respond to external nutritional changes in a rapid manner.

Although our mass spectrometry data detected Tpk1, Tpk2 and Tpk3 co-purified with Jhd2, only Tpk2 phosphorylates Jhd2 to inhibit its activity. It is possible that Tpk2 directly interacts with Jhd2, Tpk1 and Tpk3 indirectly interact with Jhd2. The different behavior of Tpk isoforms on Jhd2 is consistent with the study performed by Ptacek et al. that three Tpks have distinct substrate specificities and most substrates are recognized by only one of the Tpks[36]. Tpk1, but not Tpk2 and Tpk3 regulates non-homologous end joining double-stranded break repair by phosphorylating Nej1[37]. Tpk2 activates pseudohyphal growth, while Tpk3 inhibits filamentation and Tpk1 has no effect[38]. Tpk1 and Tpk2 have distinct functions in regulating iron uptake and respiration[53]. To understand the specific regulation of Jhd2 by Tpk2, we examined the subcellular localization of Tpk1, Tpk2 and Tpk3 within cells when grown in 2% glucose-containing medium. Tpk1, Tpk2 and Tpk3 are distributed over both the nuclear and the cytoplasmic compartments (Supplementary Fig. 11a), suggesting the specificity of Tpk2 to Jhd2 may not be related to its subcellular localization. It has been reported that PKA subunits are differentially expressed during fermentative growth[54]. We then examined the expression of PKA subunits (Tpk1, Tpk2, Tpk3) when cells were treated with different concentrations of glucose. Although the expression of PKA subunits was not significantly altered by glucose, the expression of Tpk1 and Tpk2 was significantly higher than Tpk3 (Supplementary Fig. 11b). We also examined the effect of glucose on the interaction between Bcy1 and Tpk1/2/3. Our data showed that Tpk2 dissociated with Bcy1 with a faster kinetics than Tpk1 and Tpk3 (Supplementary Fig. 11c). It is possible that the relatively high expression of Tpk2 and the faster dissociation kinetics of Tpk2 with Bcy1 in response to glucose determine the specific regulation of Jhd2 by Tpk2. In addition, Bcy1 has been reported to interact with other proteins, i.e., Eno2 (enolase II), Hsp60, and Ira2[55]. It is also possible that these Bcy1-interacting proteins may contribute to the specificity of cAMP-PKA pathway.

We identified an interaction between Jhd2 and Rpd3, which ensures the coordinate regulation of H3K4me3 and H3K14ac. The interaction between Jhd2 and Rpd3 could facilitate the binding of Rpd3 at chromatin. Tpk2-catalyzed Jhd2 phosphorylation reduced

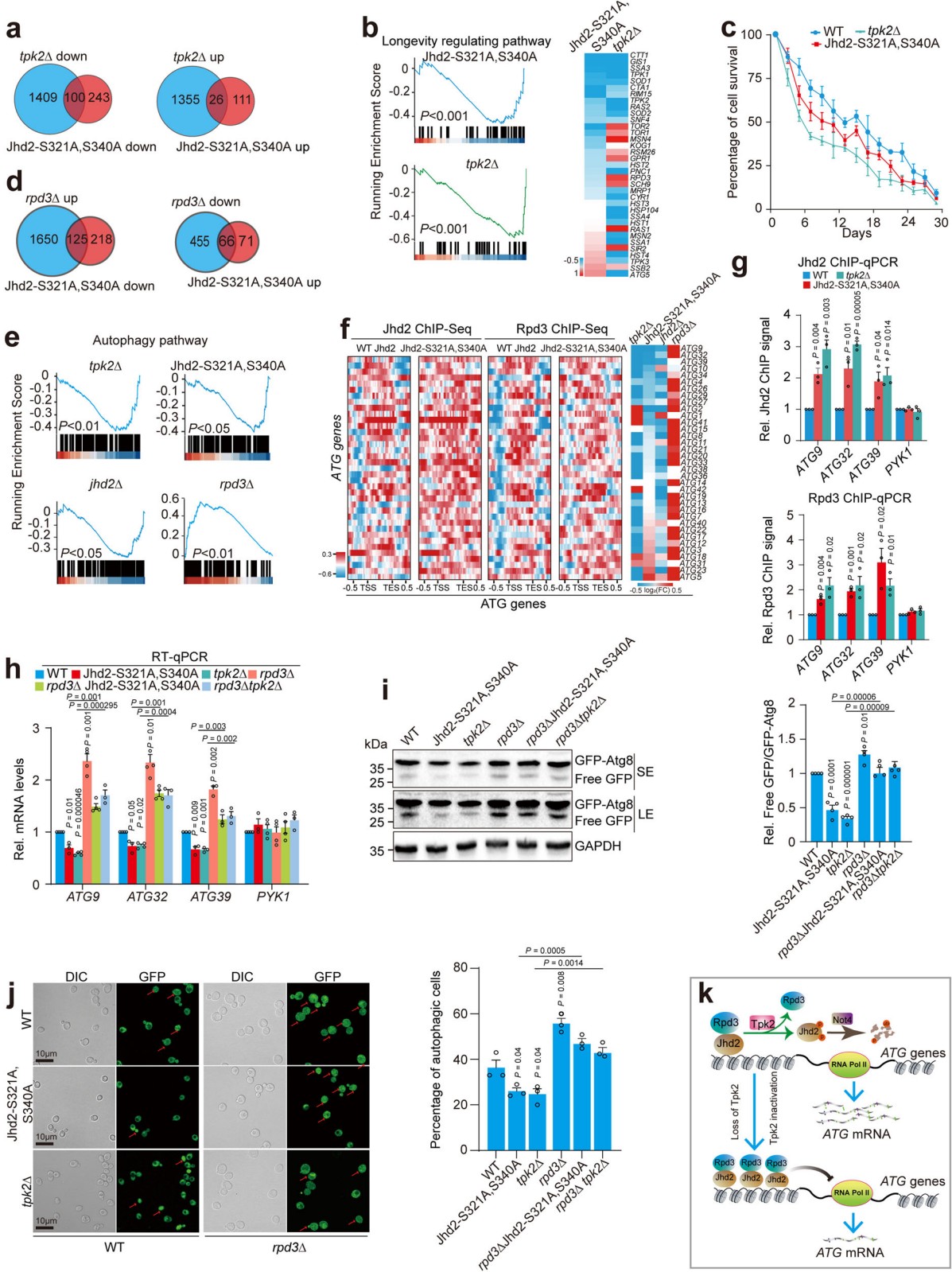

the binding of Jhd2 to chromatin as well as the interaction between Jhd2 and Rpd3, which results in decreased binding of Rpd3 at chromatin. In the presence of glucose, Jhd2 is phosphorylated by Tpk2 to reduce the interaction between Jhd2 and Rpd3, which increases both H3K4me3 and H3K14ac. Under glucose limitations, Jhd2 phosphorylation is reduced and the interaction between Rpd3 and Jhd2 is enhanced to reduce both H3K4me3 and H3K14ac. In

addition, our data reveal a reciprocal inhibitory loop between Jhd2 and H3K14ac in response to glucose availability. H3K14ac has been reported to inhibit Jhd2 binding at chromatin[56]. Here, we find that Jhd2 phosphorylation is required for H3K14ac to repel Jhd2 as the binding of Jhd2-S321A, S340A to H3 (1–23) peptide is not affected by H3K14ac (Extended Data Fig. 5b). Moreover, Tpk2-catalyzed Jhd2 phosphorylation in turn promotes H3K14ac by preventing the

**Fig. 7 | Tpk2-catalyzed Jhd2 phosphorylation maintains normal chronological life span and promotes autophagy. a** Venn diagram showing genes co-regulated by Jhd2-S321A, S340A and *tpk2Δ* mutants. **b** Left panel: GSEA analysis of longevity regulating genes in Jhd2-S321A, S340A and *tpk2Δ* mutants. Right panel: Heatmap showing the transcription of longevity regulating genes in Jhd2-S321A, S340A and *tpk2Δ* mutants. **c** Chronological life span analysis of WT, Jhd2-S321A, S340A and *tpk2Δ* mutants. **d** Venn diagram showing genes co-regulated by Jhd2-S321A, S340A and *rpd3Δ* mutants. **e** GSEA analysis of autophagy genes regulated by *tpk2Δ*, Jhd2-S321A, S340A, *jhd2Δ* and *rpd3Δ* mutants. **f** Heatmap showing the binding of Jhd2 and Rpd3 at autophagy genes in WT and Jhd2-S321A, S340A mutant. Heatmap for the transcription of autophagy genes in *jhd2Δ*, Jhd2-S321A, S340A, *tpk2Δ* and *rpd3Δ*

mutants was also provided. **g** ChIP-qPCR analysis of Jhd2 (top panel) and Rpd3 (bottom panel) occupancy at representative autophagy genes in WT, Jhd2-S321A, S340A and *tpk2Δ* mutants. *PYK1* was used as a negative control. **h** RT-qPCR analysis of representative autophagy genes in WT, Jhd2-S321A, S340A, *tpk2Δ*, *rpd3Δ*, *rpd3Δ* Jhd2-S321A, S340A, and *rpd3Δ tpk2Δ* mutants. **i, j** Analysis of autophagy activity in WT, Jhd2-S321A, S340A, *tpk2Δ*, *rpd3Δ*, *rpd3Δ* Jhd2-S321A, S340A, and *rpd3Δ tpk2Δ* mutants. **k** Proposed model for regulation of autophagy by Tpk2-catalyzed Jhd2 phosphorylation. **b, e**, The *P* value was calculated using a permutation test with 1,000 permutations. **c, g–j**, Data represent the mean ± SEM of three biological independent experiments. Two-sided *t*-tests were used for statistical analysis. Source data are provided as a Source data file.

binding of Rpd3 to chromatin. The increased H3K14ac can then inhibit Jhd2 binding at chromatin to further enhance H3K4me3. Rpd3 has been reported to be recruited by Pho23 and Rxt1 to a subset of genes such as *PHO5*, *REI1*, *TEA1* and *RRN11*[46,47]. H3K4me3 promotes the binding of Rpd3 via the PHD fingers of Pho23 and Rxt1 to repress the transcription of these genes[46,47]. However, loss of Pho23 and Rxt1 had no significant effect on Rpd3 occupancy at genes co-regulated by Jhd2-S321A, S340A and Rpd3, such as autophagy genes, *NCS2* and *EGR28*, suggesting that Rpd3 is recruited to different genes by different mechanisms. Thus, we identified a molecular pathway to coordinately regulate histone H3K4me3 and H3K14ac in response to glucose availability, which may help cells adjust their growth and proliferation to nutritional changes.

Autophagy is an important catabolic process that maintains cell homeostasis by adapting to external stress conditions[57]. Although the components of autophagy in the cytoplasm have been well studied, the mechanism for the transcriptional and epigenetic regulation of autophagy in response to nutrient changes remains poorly understood. Histone deacetylase Rpd3 has been reported to bind and repress the transcription of autophagy (*ATG*) genes[49]. But how Rpd3 is dissociated from *ATG* genes to derepress their transcription remains largely unknown. Here, our findings demonstrate that Tpk2-catalyzed Jhd2 phosphorylation is a crucial nuclear event to activate *ATG* genes transcription, which maintains basal autophagy and induces autophagy upon glucose starvation. Moreover, Tpk2 and phosphorylated Jhd2 promote autophagy by attenuating the inhibitory effect of Rpd3. Loss of Jhd2 phosphorylation or deletion of *TPK2* results in increased binding of Rpd3 at *ATG* genes, leading to their transcriptional repression. Therefore, we identified an upstream signaling pathway that antagonizes Rpd3, which may facilitate the induction of autophagy upon glucose starvation.

Simultaneous inactivation of PKA (Tpk1, Tpk2, Tpk3) has been shown to induce autophagy most likely by controlling the early stage of membrane trafficking[58,59]. The mTOR pathway inactivates autophagy in part by activating Tpk1[60], implying a negative role of Tpk1 in autophagy. However, we find that a positive role of Tpk2 in autophagy. Loss of Tpk2, but not Tpk1 and Tpk3 abolishes Jhd2 phosphorylation and increases the binding of Rpd3 at *ATG* genes, resulting in lower autophagy activity. In contrast to the role of Tpk2 in activating autophagy, our data showed that loss of Tpk1 and Tpk3 had no significant effect on the transcription of autophagy genes as well as autophagy activity. Thus, our data provide an example that PKA family kinases Tpk1, Tpk2 and Tpk3 performing different functions by phosphorylating distinct substrates.

Together, we identify a molecular signaling pathway that directly connects glycolysis to histone modifications and gene transcription, which may be conserved in mammalian cells. We also uncover a regulatory mechanism for histone demethylase Jhd2 and histone deacetylase Rpd3. Given the fact that yeast and cancer cells prefer aerobic glycolysis, so called the "Warburg effect" and H3K4me3 plays an important role in tumor growth[23,61], elucidating how glycolysis promotes H3K4me3 should shed lights on understanding the "Warburg effect".

## Methods

### Materials
All yeast strains used in this study are listed in Supplementary Table 1. The gene deletion mutants and genomic integration of C-terminal epitope tags were constructed by homologous recombination of PCR fragments. All yeast strains were verified by colony PCR, DNA sequencing, RT-qPCR, and/or Western blots. The sequence of oligomers used for qPCR is listed in Supplementary Table 2.

### Cell growth and treatment
To examine the effect of glucose on histone modifications, yeast cells were grown in 2% glucose-containing YPD (Yeast Extract Peptone Dextrose) medium until $OD_{600}$ of 0.7–1.0. Cells were then collected, washed and resuspended in YP medium for 3 hr followed by treatment with different concentrations of glucose for 0.5 hr. For 1NM-PP1 (MCE, HY-13804) treatment, cells were grown in 2% YPD until $OD_{600}$ of 0.7–1.0. Cells were then treated with 25 μM 1NM-PP1 for 0.5 hr. For FBP and pyruvate treatment, cells were pre-treated with Zymolase (Mpbio, 08320921) to increase the permeability. In brief, cells were grown in YPD medium until $OD_{600}$ of 0.7–1.0, centrifuged and resuspended in 1 ml SB buffer (100 mM Sorbitol, 20 mM Tris pH 7.4). 10 μl Zymolase was then added and incubated at 30 °C for 15 min. The spheroplasts were washed and cultured in YPD medium containing different concentrations of FBP and pyruvate for 3 hr.

### Immunoblot analysis
Cells were grown in 5 ml YPD or selective medium until $OD_{600}$ of 0.7–1.0. Cells were harvested and lysed in alkaline lysis buffer (2 M NaOH, 8% 2-mercaptoethanol). After centrifugation, the protein pellet was resuspended in 150 μl 2×SDS-sample buffer. Protein samples were separated by 8–15% SDS-PAGE and transferred to PVDF membrane. The blots were probed with primary antibodies followed by incubation with horseradish peroxidase-labelled IgG secondary antibodies. The protein bands were visualized using the ECL Chemiluminescence Detection Kit (Bio-Rad, 170–5061) and quantified with Image J software (v.1.8.0).

### Antibodies
Antibodies against anti-H3 (1: 5000; ab1791), histone H4 (1:5000; ab10158) and H3K14ac (1:2000; ab52946) were purchased from Abcam; antibodies against H3K4me3 (1:3000; A2357), H3K4me2 (1:5000; A2356), H3K4me1 (1:5000; A2355), H3K36me3 (1:5000; A2366), H3K79me3 (1:5000; A2369), Alexa Fluor 488-conjugated Goat Anti-Mouse IgG (H + L) (1:500; AS037), Alexa Fluor 594-conjugated Goat Anti-Rabbit IgG (H + L) (1:500; AS039) were purchased from Abclonal; antibodies against Rpd3 (1:500; sc-514160) and Set1 (1:1000; sc-101858) were purchased from Santa Cruz Biotechnology; antibodies against GAPDH (1:10000; 10494-1-AP), GFP (1:5000; 66002-1-1 g), Myc (1:5000; 60003-2-1 g), goat polyclonal anti-mouse IgG (1:5000; SA00001-1), and goat polyclonal anti-rabbit IgG (1:5000; SA00001-2) were obtained from proteintech; antibodies against histone H3 (1:3000; 9715 S), H3K9ac (1:5000; 9649 S), H2B (1:3000;12364 S), H2Bub (1:2000; 5546 S), phospho-PKA substrate (1:2000; 100G7E)

were purchased from Cell Signaling Technology; antibodies against FLAG M2 (1:3000; F1804-1MG), FK2 (1:1000; ST1200) were obtained from Sigma-Aldrich; antibody against CBP (1:2000; Abs130593) was purchased from Absin Bioscience Inc.; antibodies against unphosphorylated Jhd2 (1:500), Jhd2S321p (1:500) and Jhd2S340p (1:500) were custom-made in Abclonal; antibody against Sam1 was custom-made in Covance. The specificity of the custom-made antibodies was confirmed by Western blot analysis with either peptides or cell extract of corresponding mutants.

### Chromatin immunoprecipitation (ChIP) assay

Cells were grown in 200 ml YPD media at 30 °C until $OD_{600}$ of 1.0. The crosslinking was performed in 1% formaldehyde and quenched by adding 10 ml of 2.5 M glycine. Harvested cells were resuspended in FA-SDS lysis buffer (0.1% SDS, 40 mM HEPES-KOH, pH7.5, 1 mM EDTA pH 8.0, 1% Triton X-100, 0.1% Na deoxycholate, 1 mM PMSF, 2 μg/ml leupeptin, 1 μg/ml pepstatin A, protease inhibitor cocktail, phosphatase inhibitor cocktail) and lysed with glass beads vortexing. Chromatin was sonicated to an average size of ~500 bp and immunoprecipitated with anti-H3 (2 μl; ab1791, Abcam), anti-H3K4me3 (3 μl; ab5168, Abcam), anti-FLAG antibody (5 μl; F1804, Sigma), anti-Rpd3 (2 μl; sc-514160, Santa Cruz Biotechnology) pre-bound to Protein G Dynabeads (Invitrogen) at 4 °C overnight. The beads were washed with FA lysis buffer, FA buffer + 1 M NaCl, FA buffer + 0.5 M NaCl, TEL buffer (10 mM Tris pH 8.0, 1 mM EDTA, 0.25 M LiCl, 1% NP-40, 1% Na deoxycholate) and TE (10 mM Tris pH 7.4, 1 mM EDTA). The eluted DNA/protein complexes were treated with 20 μg Proteinase K (Roche) at 55 °C for 1 hr and the cross-link was reversed at 65 °C overnight. The DNA was digested by RNase (Roche), purified with ethanol precipitation and quantitated by qPCR with primers listed in Supplementary Table 2. For ChIP-qPCR, the percentage IP was calculated and then plotted with WT or untagged cells normalized to 1.

For ChIP-seq, the libraries were constructed and sequenced on an Illumina platform[19,62]. Reads were aligned to yeast genome sacCer3 from UCSC using bowtie2 version 2.1.0 with parameter -k 1. Data was put into R (3.1.0) for further analysis. Peaks were called using MACS2 (v.2.1.1, macs2 callpeak) with parameter -t -c -g 1.2e7 -n -B -q 0.01 --nomodel. Peak annotation was performed on a website service (https://manticore.niehs.nih.gov/pavis2/). Tracks were smoothed by deepTools2 (v.2.0) and visualized by IGV software (v.2.0) with a reference genome of *S. cerevisiae* (sacCer3).

### In vitro kinase assay

0–300 ng purified Jhd2 was mixed with or without 200 ng TAP purified Tpk2 in 2×kinase reaction buffer (100 mM Tris-HCl pH 7.4, 100 mM $MgCl_2$, 10 mM ATP, 0.2 μM PMSF, 1 mM $Na_3VO_4$, 5% glycerol) at 30 °C. Equal aliquots were taken at different time points, and the reaction was quenched by adding 2×SDS-PAGE loading buffer and incubated at 95 °C for 10 min. Boiled samples were resolved on 10% SDS-PAGE and subjected to Western blot analysis.

### In vitro demethylase assay

20 ng H3K4me3 (1–23) peptide was incubated with 2 ng purified Jhd2 in histone demethylation buffer (50 mM HEPES-KOH pH 8.0, 400 μM $FeSO_4$, 1 mM α-ketoglutarate, 2 mM ascorbate, 5% glycerol, 0.2 mM PMSF) at 37 °C for 3 hr. The reaction was quenched by adding 2×SDS-PAGE loading buffer and incubated at 95 °C for 10 min. Samples were spotted to PVDF membrane followed by dot blots with indicated antibodies.

### In vitro histone deacetylase (HDAC) assay

The purified WT Jhd2 and Jhd2-S320A, S340A proteins were incubated with 50 ng H3K14ac (1–23) peptide in 40 μl histone deacetylation buffer (10 mM Tris-HCl pH8.0, 150 mM NaCl, 1 mM MgOAc, 1 mM imidazole, 2 mM EGTA pH8.0, 10 mM 2-mercaptoethanol, 0.1% NP40,

10% glycerol) at 30°C for 1 hr. The reaction was quenched by 2×SDS-PAGE loading buffer and boiled at 95°C for 10 min. The reaction products were spotted to PVDF membrane followed by dot blots with indicated antibodies.

### In vitro histone methyltransferase (HMT) assay

The HMT reactions were carried out using purified COMPASS (Spp1-CBP) and 15 μg recombinant H3 in 50 μl reaction buffer (50 mM Tris pH 8.5, 20 mM KCl, 10 mM $MgCl_2$, 10 mM 2-mercaptoethanol, 100 μM SAM, 250 mM sucrose, protease inhibitor cocktail) at 30 °C for 0–2 hr. Reactions were quenched by adding 5 × SDS loading buffer and boiled at 95°C for 5 min. Protein was separated on a 10–20% SDS-PAGE gel followed by Western blots with the antibodies indicated.

### Quantitative reverse transcription PCR (RT-qPCR)

Total RNA was isolated from exponentially growing yeast cells by standard phenol-chloroform extraction procedures[17]. The extracted RNA was treated with RNase-free DNase I (Takara, 2270 A) and quantified by Nanodrop 2000 (Thermo scientific). The RNA integrity was determined by agarose gel electrophoresis. 500 ng total RNA was used for reverse transcription PCR (RT-PCR) in a 10 μl reaction volume with Reverse Transcriptase Kit (M-MLV) (ZOMANBIO). qPCR was carried out on a Bio-Rad real-time PCR machine with iTaq™ Universal SYBR® Green Supermix (Bio-Rad, 1725121). Primers used for RT-qPCR are described in Supplementary Table 2. $2^{(-\Delta\Delta Ct)}$ was used to calculate the quantity of relative transcription level.

### RNA sequencing (RNA-seq)

Total RNA was isolated from exponential growing yeast cells by standard phenol-chloroform extraction procedures and the quality of RNA was examined using Agilent Bioanalyzer. Library construction, sequencing and bioinformatics analysis were performed by Origingene company (Shanghai, China). There are three biological replicates for WT and Jhd2-S320A, S340A mutant. The differential expression levels of aligned sequences were calculated using significant thresholds (fold-change ≥1.5, fold-change ≤0.75; $P < 0.05$). The differentially expressed genes were further used for KEGG pathway analysis. One-side hypergeometric test was used for computing $P$ values.

### Microscopy analysis

Yeast cells were cultured in YPD or selective medium until $OD_{600}$ of 1.0. After washing with cold phosphate buffered saline (PBS), cells were fixed with 4% formaldehyde in PBS for 30 min and treated with DAPI (Solarbio, c0065) for 15 min at room temperature. Cells were then washed with cold PBS. The cell morphology was visualized using a ZEISS LSM710 microscope (Germany) with a 100× oil immersion objective by fluorescent microscopy. Images were acquired using ZEN Imaging Software ZEN 2.1 (ZEISS). DAPI was used to indicate the nucleus.

For immunofluorescence, cells were grown to $OD_{600}$ of 1.0 and fixed with formaldehyde for 30 min. After fixation, cells were resuspended in 0.5 ml of 1.2 M sorbitol phosphate citrate buffer (SPC) containing 0.01 % zymolyase 20 T for 1.5 hr at 30 °C. The spheroplasts were added on the poly-L-lysine-coated slides, blocked by 3% bovine serum albumin (BSA) in phosphate buffered saline (PBS) at 30 °C for 1 hr and then incubated with primary antibodies at 4 °C overnight. After washing with PBS for three times, cells were incubated with Alexa Fluor 594 goat anti-rabbit IgG (Invitrogen) at 30 °C for 1 hr. Cells were then washed with cold PBS and visualized by ZEISS LSM710 microscope using ZEN Imaging Software (ZEISS). The merged color images were generated by Fiji software.

### Sample preparation and mass spectrometry analysis

Yeast cells were grown in 2 L YPD media to an $OD_{600}$ of 1.0, washed and flash frozen in liquid nitrogen. Thawed cell pellets were resuspended in

TAP binding buffer (40 mM HEPES-KOH, pH 7.5, 350 mM NaCl, 1 mM EGTA, 10% glycerol, 0.1% Tween-20, 50 mM NaF, 1 mM $Na_3VO_4$) and lysed with glass beads using a Biospec bead beater [30 sec ON, 90 sec OFF] at 4 °C for 45 min. The lysate was centrifuged at 18,400 g at 4 °C for 1.5 hr and the supernatant was incubated with anti-FLAG IP Resin (GenScript, L00425) or IgG beads at 4 °C for 4 hr. The beads were washed extensively in TAP binding buffer for on-bead trypsin digestion. Briefly, immunoprecipitated proteins were disulphide-reduced by adding DTT to a final concentration of 25 mM and reacted at 37°C for 40 min. Cysteines were alkylated by adding iodoacetamide (Sigma, I1149) to a final concentration of 50 mM, followed by incubation in the dark for 30 min. Immunoprecipitated proteins were digested overnight with sequencing-grade trypsin (Promega) at 37°C and the supernatant peptides were then desalted using C18 columns (Thermo Fisher) and lyophilized. The dried peptides were reconstituted in 0.1% FA and loaded onto an Acclaim PepMap 100 C18 LC column (Thermo Fisher) utilizing a Thermo Easy nLC 1000 LC system (Thermo Fisher) connected to Q Exactive HF mass spectrometer (Thermo Fisher). The peptides were eluted with a 5–20% gradient of acetonitrile with 0.1% formic acid over 70 min with a flow rate of 300 nl min$^{-1}$. The MS1 scans were performed at a resolution of 60,000 over a mass range of 380–1,560 m/z, with a maximum injection time of 120 ms and an AGC target of $1 \times 10^6$. The MS2 scans were performed at a resolution of 15,000, with the normalized collision energy set to 24, a maximum injection time of 50 ms and an AGC target of $2 \times 10^5$. The raw mass spectrometry data were searched against the *Saccharomyces cerevisiae* proteome database from Uniport (https://www.uniprot.org/proteomes/UP000002311) using Sequest HT, MS Amanda and ptmRS algorithms in Proteome Discoverer 2.3 (Thermo Fisher). The precursor ion mass tolerance was set to 10 ppm and the fragment ion mass was 0.02 Da. Dynamic modifications were set for methionine oxidation and phosphorylation on serine, threonine and tyrosine. Only fully tryptic peptides with up to two mis-cleavages and a false-detection rate of 1% using the percolator validator algorithms were accepted.

The phosphorylation sites were determined using mass spectrometry as described[63]. Briefly, samples were analyzed by LC-MS/MS on a Thermo Q Exactive HF mass spectrometer (Thermo) using a top twenty method. MS/MS spectra were searched against a yeast protein database from Uniport and processed in default parameter by Proteome Discoverer 2.3 software.

## Molecular docking

The crystal structure of PKA was obtained from the Protein Data Bank (PDBID:1atp). To visualize the docked conformation, the PyMol molecular graphics system was used, which removed water molecules and peptide inhibitor PKI. The structure of Jhd2 peptides (318-RKLS-321, 337-RRSS-340) was generated using I-TASSER protein Structure and Function Prediction web server (http://zhanglab.ccmb.med.umich.edu/I-TASSER/). The molecular docking simulation of PKA and Jhd2 peptides was performed with the AutoDock Vina[64]. All residues within PKA binding site were included using the following spatial coordinates of the central cavity: x = 72, y = 74, and z = 90. The coordinates of the grid resolution were x = 9.662, y = 8.83, z = −2.971.

## Measurement of intracellular SAM levels

Metabolite samples were prepared according to the standard protocol[16]. Briefly, cells were grown in 6 ml YPD media at 28 °C until $OD_{600}$ of 0.7–1.0. Quenching was performed by adding 4 ml buffer 1 (60% methanol with 10 mM tricine) at −80 °C for 5 min. Cells were then centrifuged and resuspended in 400 µl buffer 2 (75% ethanol with 0.5 mM tricine). Cells were lysed by incubating at 80 °C for 3 min followed by cooling on ice bath for 5 min. Cell suspension was centrifuged at 20,000 × g for 10 min. Supernatant was collected and the SAM concentration was determined using the SAM quantification kit

with a standard curve (Comin Biotechnology Co., Ltd, Suzhou). At least three biological replicates for each sample were used for SAM quantification assay.

## Co-immunoprecipitation (Co-IP)

Yeast whole cell extract was prepared by vortexing with glass beads and then digested with Micrococcal Nuclease (MNase). Pre-cleared cell lysate was incubated with anti-FLAG M2 agarose for 1 hr at 4 °C. The beads were washed three times with IP washing buffer (40 mM HEPES-KOH, pH7.5, 0.1% NP-40, 10% glycerol, 1 mM PMSF, 350 mM NaCl, 2 µg/ml leupeptin, 1 µg/ml pepstatin A) and boiled in SDS-sample buffer for 5 min. Supernatants from the boiled beads were subjected to SDS-PAGE and Western blots.

For in vitro immunoprecipitation, Jhd2-FLAG was affinity purified by anti-FLAG M2 affinity gel from yeast cell lysate[18]. Purified Jhd2 was then incubated with 0.2 µg in vitro assembled recombinant nucleosomes. Jhd2 was immunoprecipitated with anti-FLAG M2 agarose and washed three times with 3 × 1 ml IP washing buffer. Supernatants from the boiled beads were subject to SDS-PAGE followed by Western blots.

## Peptide Pull-down Assay

The peptide pull-down assays were performed in accordance with the protocol described[65]. In brief, 10 µg biotinylated H3 (1–23) peptides (DGpeptides Co., Ltd, Wuhan, China) were incubated with 1 mg protein in binding buffer (50 mM Tris-HCl, pH 7.5, 250 mM NaCl, 0.1% NP-40, 1 mM PMSF, protease inhibitor cocktail) at 4 °C overnight. The peptides and associated proteins were pulled down by incubation with Streptavidin beads (Amersham) at 4 °C for 4 hr. The beads were washed with 3×1 ml binding buffer and boiled for 5 min. The supernatant from boiled beads was subject to Western blot analysis. The peptides used are the following:

Unmodified H3 (1-23): ARTKQTARKSTGGKAPRKQIASK;
H3K4me3 (1-23): ARTK(me3)QTARKSTGGKAPRKQIASK;
H3K14ac (1-23): ARTKQTARKSTGGK(ac)APRKQIASK.

## Statistics and Reproducibility

Representative results of at least two biological independent experiments were performed in all of the figure panels. For Figs. 1a, b, f; 2g–m; 6a–c, i; supplementary Figs. 2a, c–d; 3a, d–e; 4a–b; 6c; 9h; 11a, experiments were repeated three times independently with similar results. For Figs. 2a, b; 4a, c–d; 5e, g–h; 6d–g, p–q; supplementary Figs. 1d, g–h; 2e–f; 3f–h; 4c; 5a–b; 11c, experiments were repeated two times independently with similar results. The two-sided Student's *t*-test was used for comparison between two groups and *P* value < 0.05 was considered statistically significant. Microsoft Excel (professional Plus2013) was used for basic statistical analysis. For boxplots by ggplot2 package, two-sided Wilcoxon test in R (package ggpval) was used for statistical analysis. For all error bars, data are mean ± SEM.

## Reporting summary

Further information on research design is available in the Nature Research Reporting Summary linked to this article.

## Data availability

All data supporting the findings of this study are included in the manuscript and its supplementary files are available. The RNA-seq data for WT and Jhd2-S321A, S340A generated in this study have been deposited in the GEO database under accession number GSE175870. The ChIP-seq data for Jhd2, H3K4me3 and Rpd3 generated in this study have been deposited in the GEO database under accession number GSE175868. The RNA-seq data for *rpd3Δ*, *jhd2Δ*, and *tpk2Δ* are available in the GEO database under accession number GSE67149, GSE73407, and GSE28213, respectively. The mass spectrometry proteomics data have been deposited to the ProteomeXchange Consortium via the PRIDE partner repository with the dataset identifier PXD030815. The

protein structure data for molecular docking used in this study is available on PDB database with identifier of 1atp (https://www.rcsb.org/structure/1ATP). The data that support the findings of this study are available from the corresponding authors upon reasonable request. Source data are provided with this paper.

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

## Acknowledgements

We would like to thank Dr. Guohong Li (Institute of Biophysics, Chinese Academy of Sciences) for providing the nucleosomes. We also thank Professor Pingfang Yang and Ming Li (Hubei University) for suggestions on mass spectrometry. This project was supported by funding from National Natural Science Foundation of China (31970578 to S.L., 31872812 to X.Y.), Natural Science Foundation of Hubei Province (2021CFA013 to S.L., 2019CFA077 to X.Y.).

## Author contributions

Conceptualization: X.Y., S.L.; Experiments were performed by Q.Y., X.G., X.Y., Y.T., K.D., X.Z.; Mass spec analysis was performed by M.W., F.G.; Statistical analysis was performed by Q.Y., X.G., X.Y.; Writing, review and editing: X.Y., S.L.

## Competing interests

The authors declare no competing interests.
