## [Peer Review File · Nature Communications]

Phosphorylation of Jhd2 by the Ras-cAMP-PKA(Tpk2) pathway regulates histone modifications and autophagyREVIEWER COMMENTS

Reviewer #1 (Remarks to the Author):

In this manuscript, Yu et al report phosphorylation of Jhd2 H3K4 demethylase by the Ras-cAMP-PKA (Tpk2) pathway regulates localization, stability, and the functions of Jhd2 in regulating histone modifications. They further implicate a role for Jhd2 phosphorylation in maintaining chronological life span and autophagy in yeast cells. Employing standard molecular biology and genomics approaches, the authors have performed studies to provide a connection between nutrient signaling or metabolism, epigenetic histone modifications and gene expression related to autophagy. The efforts undertaken by the authors to perform the many experiments and present a high information content manuscript is highly commendable. However, absence of a clear logical plan of experiments, poor overall presentation and absence of key experiments make this manuscript unsuitable for publication. Some of the many major and minor concerns that led me to this conclusion are as stated below.

1) In figure 1, the authors describe their foundational data for the manuscript, that glucose/glycolysis promotes H3K4me3. *Saccharomyces cerevisiae* is a glucophilic yeast, and H3K4me3 and H3K14ac are well-established bona fide marks of gene activation. So, active gene expression in actively growing yeast and thus increase in the well-known transcriptional activation marks such as H3K4me3, H3K36me3 and H3K14ac, are all expected and thus as such their finding is not novel. The authors cite a study that showed that glucose/carbohydrates controls the upstream regulators of H3K4me3. The authors completely ignore and do not test the possibility that glucose controls the many well-known upstream regulators of H3K4me3, but instead go onto examining Jhd2 phosphorylation.

2) Throughout the manuscript, the authors cite some of the published studies on Jhd2 while ignoring many other relevant ones. Even with the cited studies, they completely disregard or not account for nor describe their results with the reported findings. Many references appear simply inserted for the sake of citing!

3) Decrease in H3K4me3 levels could be due to increased Jhd2 H3K4 demethylase activity or owing to decreased Set1 H3K4 methyltransferase activity. The authors show that Set1 levels remain unchanged upon loss of Tpk2 (Extended data Fig.2b). In yeast, Set1 is part of a multi-protein complex called COMPASS and many subunits of this complex such as Spp1/Cps40 regulate H3K4me3. However, the authors do not address whether absence of Tpk2 adversely affects the levels of COMPASS subunits to decrease H3K4me3.

4) In Fig.2, the authors report the results of their affinity purification and mass spectrometry (AP-MS) experiment to identify Jhd2 interacting proteins. A) The authors do not show relevant data, such as, silver stained gels of their affinity purification experiment (minor). B) Using the TAP tag purification strategy as reported in this manuscript, Jhd2 was previously reported to interact with the polyadenylation machinery by Blair et al. (*Sci. Adv.* 2016;2: e1501662), but Tpk2 was not reported as a Jhd2-interacting protein. How do authors reconcile with this discrepancy in their results with that in the published work?

5) One assumes that the logic behind the AP-MS and co-IP experiments is that because Jhd2 interacts with Tpk2, it must be a substrate of Tpk2 kinase. However, it is well known that kinase-substrate interactions are transient and as such, standard immunoprecipitation do not retain these interactions. Generally, a Shokat allele of Tpk2 and use of a bulky ATP analog can show that Jhd2 is a substrate of Tpk2. The authors can refer to a review by Sergio et al (Science Signaling 2016 Vol. 9, Issue 420, pp. re3). The discovery of a role for the Ras-cAMP-PKA (Tpk2) pathway in regulating H3K4me3 is the most interesting finding in this paper. Therefore, the role for phosphorylation of Jhd2 by Tpk2 can be further strengthened and highlighted in this manuscript.

6) To extend on the point above, from AP-MS, the authors report that Jhd2 also interacts with Tpk1 and Tpk3, but only loss of Tpk2 decreases Jhd2 phosphorylation. How do they reconcile with these findings?

7) Immunoblots using antibodies raised against S321 or S340 phosphorylated Jhd2 implicate Tpk2 as the effector kinase. However, the specificity of these antibodies appear to be very poor and recognize the backbone, as seen from the signal in the Jhd2-S321AS340A mutant (Fig.2i). In addition, the authors refer to the Jhd2-S321AS340A double mutant as Jhd2-S2A, which is completely misleading and confusing!

8) Why were the phosphomimetic Jhd2-S321D/S340D single or double mutants not included in the study? It seems logical to include these gain-of-function alleles along with the loss-of-function Jhd2-S321AS340A mutant.

9) The linker region containing the S321 and S340 residues was previously reported to be crucial for the stability of Jhd2. The authors do not refer to this published study!

10) In Fig 5e, the authors label high molecular weight bands for Jhd2 as Jhd2-(Ub)_n. In order to be labeled as polyubiquitinated forms, these high molecular weight Jhd2 bands need to cross-react with either an antibody recognizing ubiquitin or one recognizing mono- or poly-ubiquitinated proteins (clone FK2).

11) Using AP-MS, the authors report that Jhd2 interacts with Rpd3, and that this interaction is increased in the Jhd2-S321AS340A mutant. This observation is based on obtaining two and five peptides for Rpd3 following anti-Flag immunoprecipitation. How do we know this is significant? Their AP-MS experiments are not at all quantitative, as either SILAC or isobaric labeling approach was not used!

12) According to authors, loss of Jhd2 S321S340 phosphorylation results in decreased H3K4me3 as the Jhd2-S321AS340A mutant binds H3 peptide more tightly. They then report that Jhd2 interacts with Rpd3 deacetylase and because Jhd2-S321AS340A binds chromatin more so does Rpd3. In yeast, Rpd3 is also present as a component of multi-protein complexes (Rpd3S, Rpd3L, Rpd3μ, etc.). In yeast, Pho23 and Rxt1, both components of Rpd3L complex directly bind to H3K4me3 via their PHD finger. How do the authors reconcile their observations of increased Rpd3 binding upon decrease in H3K4me3?

13) In the introduction, the authors' state: "we and others noticed that loss of Jhd2 increases H3K4me3 only when cells were grown under low glucose conditions 21". However, many studies have clearly shown increased H3K4me3 in *jhd2Δ* strain even in standard glucose (2%) or glucose-replete media. In the reference cited #21, Figure 2d, increase in H3K4me3 is seen in *jhd2Δ* strain grown in glucose rich medium! So it is unclear what the authors mean by increase is see only when cells are grown under low glucose conditions!

Reviewer #2 (Remarks to the Author):

The manuscript submitted by Yu et al. aimed to identify the signaling pathway that regulates histone modifications in response to glucose availability. By screening glucose sensor mutants, they found that the PKA/Tpk2 is required for glycolysis to promote H3K4me3. Mass spec and in vitro kinase assay showed Tpk2 directly phosphorylates H3K4 demethylase Jhd2 at S321 and S340. Tpk2-catalyzed Jhd2 phosphorylation inhibits the demethylase activity of Jhd2 by reducing its binding at chromatin. Moreover, Tpk2-catalyzed Jhd2 phosphorylation impairs its nuclear localization and promotes its degradation. Interestingly, they showed that Tpk2-catalyzed Jhd2 phosphorylation reduced the binding of Rpd3 to chromatin, which coordinately regulates the crosstalk between H3K4me3 and H3K14ac. Finally, they showed that Tpk2-catalyzed Jhd2 phosphorylation regulates gene expression, chronological lifespan and autophagy.

The strengths of this paper include the use of multiple in vivo and in vitro approaches as well as the creation of unique reagents, such as Jhd2 S321 and S340 phosphorylation specific antibodies and Jhd2 phosphorylation site mutations to confirm Tpk2-mediated Jhd2 phosphorylation and explore its effect on protein-protein interaction and activity. These new findings reveal exciting connections between glucose metabolism and histone modifications that is of significant interest to both metabolism and chromatin community.

1. Since there are three protein kinase A (PKAs, Tpk1/2/3) in yeast, why only Tpk2 but not Tpk1 or Tpk3 affects histone modifications H3K4me3? What is the effect of Tpk1 and Tpk3 on autophagy?

2. The regulation of Rpd3 by Jhd2 phosphorylation is interesting. The binding of Rpd3 to chromatin is enhanced in Tpk2 and Jhd2 S2A mutant. But how Jhd2 phosphorylation reduced the binding of Rpd3 to chromatin is unknown. The author can provide plausible explanations in the discussion section. In addition, both H3K4me3 and H3K14ac are reduced in NOT4 deletion mutant. The reduced H3K4me3 is caused by increased Jhd2 stability and activity. As Jhd2 physically interacts with Rpd3, I am wondering whether Rpd3 can be degraded by Not4-related proteasome pathway.

3. The number of unique peptides for Tpk2 is low as a bait protein in purification and mass spectrometry in Fig. 2a. Although they used Co-IP and in vitro kinase assay to confirm their physical interaction, it is better to redo the purification and mass spec analysis.

4. For ChIP-qPCR experiments of Fig. 4o and 4p: a negative control should be added to underline the specificity of the three selected genes.

Minor points:

Fig. 4e and 6i: Non-specific band of H3 should be marked.

Fig. 7J: The labels of statistical significance are different from labels in Fig. 7i.

Reviewer #3 (Remarks to the Author):

The manuscript entitles “Phosphorylation of Jhd2 by the Ras-cAMP-PKA(Tpk2) pathway regulates histone modifications and autophagy” the authors describe and clearly demonstrate a mechanism by which Jhd2 together with Rpd3 regulate transcription of the genes involved in normal life span maintenance and in the induction of autophagy. The manuscript shows a lot of work performed by the authors and the most important is that it provides a link between nutrient availability, glucose, and the mechanism involved in the transcription regulation of life span regulation and autophagy. this work highlights a novel regulatory mechanism that involves the phosphorylation of Jhd2 by the Tpk2 subunit of PKA.

The results obtained from the experiments provide significant support for the conclusions. The conclusions are significant as they have taken a step forward in the field. The genomic data is well analysed but some concerns about the analysis and validations are detailed below.

The text is well written and very clear, and in general the results have been provided with the adequate context.

The following are points that should be addressed to help improve the paper

Introduction

Page 4- lines 61-63. “While AMPK is activated by low energy status and mediates cellular responses to energetic stress, mTOR couples nutrient abundance to cell growth in response to nutrients, growth factors and cellular energy”.

The writing is not clear and confusing. Please change the description of both pathways in a way that it indicates the real differences between them.

Pag 4 -lines 63-66. The cellular process regulated by the cAMP-PKA pathway are poorly described. There are many papers and specially reviews on the subject. Reference 6 should be replaced.

Pag 4- line 70. Histones are not the “structure unit” of chromatin, this phrase should be modified since the basic repeating structural (and functional) unit of chromatin is the nucleosome.

Pag 5-line 93. Although the cAMP-PKA pathway was mentioned previously, in the paragraph which begins in line 93, it is the first time that Tpk2 appears. It would be necessary to provide a complete description of *Saccharomyces cerevisiae* PKA and the corresponding references, before this line. PKA, specially the catalytic subunit Tpk2, has a key role in the mechanism described in this work.

On the other hand, Tpk2 is not activated. The holoenzyme PKA is activated when cAMP is bound to Bcy1 subunits. Released Tpk subunits (all isoforms) could then phosphorylate their substrates

Results

Page 6-lines 106-107. In these lines it is the first time that the authors mention that the yeast cells were grown in medium containing different glucose concentrations. In the section material and methods, they do not describe clearly what the culture conditions were, or at least I could not find this information.

How were the cells grown? to early log phase? Which was the OD600 chosen? When the cells were in the OD600 the cells were washed once, and resuspended in the medium lacking glucose and then the glucose was added to different final concentrations? Or were cells grown to early log phase overnight in different glucose concentrations? Did the authors use pre-cultures? Were the cultures really grown with 0% glucose as indicated in the figures?

It is important that each time that the authors write “cells were grown”, they detail the growth phase (early log phase?, OD?)

It is necessary to clarify how the yeast cells were grown.

Page 6- line 121. The authors wrote that they deleted PDE1 and PDE2, however in the strains table, is described that the source of these strains is Open Biosystems. Related to this concern, in figure 1 panel I, change *ped2Δ* by *pde2Δ*.

Fig 1 panel h. This panel is not necessary, the decreased level of cAMP is well described previously. There are several works (at least since 1983).

Page 7- line 128. "While loss of Ras1 slightly reduce H3K4me3...". The difference described by the authors between Ras1 and Ras2 is not clear since both mutants show differences with WT strain with the same significance (extended data Fig. 1 c).

Pag 7- line 129-130. "Ras1/2 can be activated by glycolysis130 derived fructose 1,6-biphosphate (FBP) 25 (Fig. 1g)".

FBP activates Ras1/2 trough cdc25 not directly. I consider that cdc25 should be added in the text and figure

Pag 7- line 130-131. The treatment with FBP and Pyruvate is not described in Material and methods, Have the authors used yeast spheroplasts to assure the cells permeability?

Pag 7- line 131. Misspelling: lines instead line.

Pag 7- line 136. Reference 26 should be replaced by a review.

Fig 2a. The strain Jhd2-TAP is not described in Strains table

Fig 2 e. The sequence RXXS/T is not the canonical consensus sequence of PKA.

The real sequence consensus is RRXS/T.

Pag 9-Line 190. Sequence NLS: 313-RLLPAEKLSIDELEEMFWSLVTKNR RSS-340

Is the E really an E or an R? Please clarify this mistake.

Pag 10-Lines 194-195. There is not mention to the granular distribution of Jhd2 (Extended Data Fig.3 b).

Pag 11-Line 237. Peptides H3 (1-23). Detail in material and methods the peptides used. Sometimes the peptide is mentioned as Peptides H3 (1-23) and others as Peptides H3. Please, make uniform.

Pag. 12- Lines 253-256. Figure 4 e, the difference of distribution of Jhd2WT and Jhd2 S2A in subcellular fractions, is not well visualized. The % of total protein in each fraction should be calculated.

Extended Figure 4a. It is Pull-down not IP

Fig 4 h. The figure legend should describe what is defined as center.

Fig 4 o and p. In the legend, it should be detailed where the primers were designed. (ORF?). The genes chosen to confirm the ChiP-seq data, are from the 614 co-binding genes identified?

Pag 14 Lines 285-286. "The protein level of Jhd2 was significantly higher in Jhd2 S2A and *tpk2Δ* mutants than is not caused by differences in the transcription. WT Jhd2 and this difference was not caused by altered Jhd2 transcription (Fig. 5a; Extended Data Fig. 5a), suggesting that Jhd2 phosphorylation might affect Jhd2 protein stability".

The methodological approach performed to obtain the data of

Extended Data Fig 5 a is a qRT-PCR (I suppose it because it is not indicated in the legend). With this approach is not right to conclude that the difference observed between WT, Jhd2SA and *tpk2Δ* is not caused by differences in transcription. The mRNA levels measured are those from steady state, and therefore, the detectable levels are a result of degradation and transcription. The same with the protein levels, which could be the result of differences in protein translation. This paragraph should be changed.

Pag 14 line 292-293. The Jhd2 stability is regulated by ubiquitination and proteosome. This concept has just been published previously. The new result is that this process is regulated by Tpk2 phosphorylation. This should be clarified and reference 29 should be cited here.

Extended Data Fig 5b. There is a mistake, Jhd2SA instead of *cim3-1*.

Figure 6 c- In this figure it could be observed that Tpk1 also affects H3K4m3. The effect is observed in *tpk1Δ* and *tpk1Δ tpk2Δ*. This effect is neither mentioned in the results nor discussed in discussion section.

-The center in ChiP seq data of Jhd2 with Rpd3 and Jhd2SA with Rpd3 are coincident?

Fig 6L. The 200 genes in which overlapping binding sites of Jhd2 and Rpd3 (in Jhd2 S2A mutant) were detected correspond to Sites thar are unique for Jhd2? Or do they include Co-occupied sites with wt?

In the discussion section, there is not an integration of the results of ChiP-seq and RNA-seq. What is the relation between the genes occupied only by Jhd2 Wt, genes occupied only by Jhd2 S2A and the genes resulting from the overlapping and mRNA levels?

There is no discussion about the genes that are overlapped between Jhd2SA and Rpd3 summarized in the diagram of Figure 7 d. It would be interesting to hypothesize a mechanism, could other methylases be involved?

The role of autophagy in the regulation of yeast life span should also be discussed, there is bibliography about this concept, and it may be important to support the results. Here, the authors show that the longevity regulating genes are downregulated, and the ATG genes are also downregulated in mutant Jhd2SA. What is happening then in WT strain? It is described in different reports that a role for autophagy or autophagy factors in the extension of yeast life span has been identified. Please, relate this concept with the results obtained.

Page 25-lines 538-539. The concept wrote in this sentence is not clear or it is wrong (reference 38). The authors in this work show that Ras/cAMP pathway partially repress rapamycin induced autophagy maybe by controlling subcellular localization and activity of Tpk1.

Page 25-lines 542-543. Reference 39. Bibliography in which the concept of specificity of PKA is demonstrated should be added. There are a lot of recently works and reviews that address this concept. The result that only Tpk2 phosphorylates Jhd2 is very important to support cAMP-PKA specificity.

Page 25-lines 543-545. Again, the concept of cAMP-PKA specificity is very important, the possible mechanisms that allow the differential activity of each isoform of Tpk are more than those listed by the authors, it is incomplete. Reference 40 does not correspond to illustrate the concept. It is only about autophagy and in MAMMALS! And the authors are talking of yeasts. Replace this reference by OTHER more suitable ones.

The last paragraph is unnecessary from my point of view. It is not necessary to force very good results that explain a molecular mechanism in regulation of transcription in yeast, in order to extrapolate them to explain a cancer cell process as "Warburg effect".

Point-by-point response to the editors' and referees' comments

Response to editors' comments

Our responses are in blue.

Please provide complete protocols of their AP/IP-MS and phosphosite mapping experiment and deposit the mass spec raw data.

We provided complete protocols for AP/IP-MS and phosphosite mapping experiment in the Methods section of the revised manuscript (Page 34, line 760). We also deposited mass spec raw data to the ProteomeXchange Consortium via the PRIDE partner repository with the dataset identifier PXD030815.

Response to the referees' comments:

Reviewer #1 (Remarks to the Author):

In this manuscript, Yu et al report phosphorylation of Jhd2 H3K4 demethylase by the Ras-cAMP-PKA (Tpk2) pathway regulates localization, stability, and the functions of Jhd2 in regulating histone modifications. They further implicate a role for Jhd2 phosphorylation in maintaining chronological life span and autophagy in yeast cells. Employing standard molecular biology and genomics approaches, the authors have performed studies to provide a connection between nutrient signaling or metabolism, epigenetic histone modifications and gene expression related to autophagy. The efforts undertaken by the authors to perform the many experiments and present a high information content manuscript is highly commendable. However, absence of a clear logical plan of experiments, poor overall presentation and absence of key experiments make this manuscript unsuitable for publication. Some of the many major and minor concerns that led me to this conclusion are as stated below.

We appreciate the reviewer's recognition the efforts and high information content of our work. In the revised manuscript, we performed a large number of experiments, per the reviewer's comments, to support our conclusions and addressed all the concerns raised by this reviewer. Moreover, we improved the logic and overall presentation. We also cited and compared our work to those relevant references.

1) In figure 1, the authors describe their foundational data for the manuscript, that glucose/glycolysis promotes H3K4me3. *Saccharomyces cerevisiae* is a glucophilic yeast, and H3K4me3 and H3K14ac are well-established bona fide marks of gene activation. So, active gene expression in actively growing yeast and thus increase in the well-known transcriptional activation marks such as H3K4me3, H3K36me3 and H3K14ac, are all expected and thus as such their finding is not novel. The authors cite

a study that showed that glucose/carbohydrates controls the upstream regulators of H3K4me3. The authors completely ignore and do not test the possibility that glucose controls the many well-known upstream regulators of H3K4me3, but instead go onto examining Jhd2 phosphorylation.

We thank the reviewer for this comment. We agree with the reviewer that *Saccharomyces cerevisiae* has a preference for glucose. In actively growing yeast, it is not surprising to see active gene transcription and observe an increase in transcription activation marks. However, we think the finding that glucose-induced H3K4me3 is interesting for the following reasons:

1. Not all active histone markers are induced by glucose. We examined the changes of known histone modifications in response to glucose, including H2BK123 monoubiquitination (H2Bub), H3R2 asymmetric dimethylation (H3R2me2a), H3K4me3, H3K36me3, H3K79me1, H3K79me2 and H3K79me3. Our data showed that H2Bub and H3K4me3 are significantly induced by glucose in a dose-dependent manner (**Supplementary Fig. 1a**). However, glucose has no effect on other active transcription histone markers such as H3K36me3 and H3K79me3 (**Supplementary Fig. 1a**), suggesting that there is a specific mechanism for glucose to promote H3K4me3.
2. H3K4me3 is an important histone modification that regulates cell death, DNA damage, histone gene expression, chronological ageing, and telomere silencing. Most importantly, H3K4me3 is linked to transcription responsiveness, such as transcription memory of galactose-inducible gene *GAL1* (*Mol Cell* 2003, 11:709-19). As our purpose is to address how cells transduce nutrient availability to appropriate gene expression programs, it is necessary to study the regulation of H3K4me3 by glucose. By studying how glucose induces H3K4me3, we identified a novel epigenetic mechanism to transduce glucose availability to gene transcription via the PKA(Tpk2)-mediated phosphorylation of Jhd2.

To explore the mechanism for how glucose promotes H3K4me3, we examined the effect of glucose on upstream regulators of H3K4me3 from the following aspects:

1. We performed additional experiments to show that there is a H2BK123 monoubiquitination (H2Bub)-independent mechanism for glucose to induce H3K4me3. We studied the effect of glucose on histone modifications that regulate H3K4me3. H3K4me3 is regulated by H2BK123 monoubiquitination (H2Bub) and H3R2 asymmetric dimethylation (H3R2me2a) (*Cell* 2007, 131:1084-1096; *Nature* 2007, 449:928-32). H2Bub but not H3R2me2a is induced by glucose (**Supplementary Fig. 1a**). We then determine whether glucose increases H3K4me3 by inducing H2Bub. As there is no H3K4me3 in the deletion mutants of *RAD6* and *BRE1*, which catalyze H2Bub (*J Biol Chem* 2002, 277: 28368-28371; *Mol Cell* 2003, 11: 267-274), it is impossible to determine the effect of glucose on H3K4me3 in *rad6Δ* and *bre1Δ* mutants. We thus examined the effect of glucose on H3K4me3 in the deletion mutant of *UBP8* and *UBP10*, which encode H2B ubiquitin proteases. WT and *ubp8Δ ubp10Δ* mutant were treated with different concentrations of

glucose. Although H2Bub is induced by glucose in a dose-dependent manner in WT cells, H2Bub levels remain relatively high and constant in *ubp8Δ ubp10Δ* mutant (**Supplementary Fig. 1b**), suggesting that glucose has no inducing effect on H2Bub in *ubp8Δ ubp10Δ* mutant. However, we still observed a gradual increase of H3K4me3 in *ubp8Δ ubp10Δ* mutant by glucose (**Supplementary Fig. 1b**). Moreover, as H2Bub is positively regulated by Chd1 (*Genes Dev* 2012, 26:914-9), we also examined H3K4me3 in *chd1Δ* mutant when grown in different glucose concentrations. We still observed glucose-induced H3K4me3 in *chd1Δ* mutant (**Rebuttal letter Fig. 1a**). All these data indicate that there is a H2Bub-independent mechanism for glucose to induce H3K4me3 (e.g. enhancing H3K4 methyltransferase activity or inhibiting the H3K4 demethylase activity).

2. We performed additional experiments to show that glucose induces H3K4me3 not by enhancing the activity of Set1 complex, COMPASS. We purified COMPASS from cells grown in 0.05% glucose and 4% glucose, respectively and then performed the *in vitro* histone methyltransferase (HMT) assay. Our data showed that the HMT activity of COMPASS was not regulated by glucose (**Supplementary Fig. 1c**), suggesting that glucose induces H3K4me3 not by enhancing the activity of COMPASS.
3. The Ras-cAMP-PKA(Tpk2) pathway is required for glucose to induce H3K4me3. By analyzing H3K4me3 in the mutants of potential nutrient sensitive signaling molecules, including AMPK (Snf1), mTOR, Sch9 and PKA, we found that Tpk2 is required for glucose to induce H3K4me3. Further study showed that the Ras-cAMP-PKA(Tpk2) pathway is required for glucose to induce H3K4me3.
4. Tpk2 maintains H3K4me3 by phosphorylating Jhd2. To identify the downstream target(s) of Tpk2, we examined histone modifications in WT and the *tpk2Δ* mutant. However, loss of Tpk2 had no effect on H2Bub and H3R2me2a (**Supplementary Fig. 2a**). In addition, we purified COMPASS from WT and the *tpk2Δ* mutant and our *in vitro* HMT assay showed that loss of Tpk2 did not influence the HMT activity of COMPASS (**Supplementary Fig. 2f**). We then analyzed the proteins that co-purified with Tpk2 by mass spec. We detected the presence of Jhd2 but not Set1 (**Fig. 2a**). We also performed an *in vitro* kinase assay and found that Tpk2 directly phosphorylates Jhd2 but not Set1 (**Rebuttal letter Fig. 1b**). Tpk2 phosphorylates Jhd2 to reduce its activity to demethylate H3K4 (**Fig. 3d, 3g**). That is why we focused on studying H3K4 demethylation by Jhd2.

All these data prompted us to investigate whether glucose induces H3K4me3 by promoting Tpk2-mediated phosphorylation and inactivation of Jhd2. To improve the logic of this manuscript and overall presentation, we added these new experiments about the effects of glucose and Tpk2 on upstream regulators (histone modifications, COMPASS) of H3K4me3 (**Supplementary Fig. 1a, 1b, 1c, 2a, 2d, 2e, 2f**) in the revised manuscript.

Rebuttal letter Fig. 1. a Western blot analysis the effect of glucose on histone modifications in WT and *chd1Δ* mutant. WT and *chd1Δ* mutant were grown in YPD medium until OD₆₀₀ of 0.7-1.0. Cells were harvested, washed with YP medium and treated with YP medium supplemented with different concentrations of glucose for 0.5 hr. Cells were then harvested and the extracted histones were analyzed by Western blots with indicated antibodies. **b** Set1 was not phosphorylated by Tpk2 as determined by *in vitro* kinase assay. Purified Set1-FLAG was incubated with purified Tpk2-CBP at 30 °C for 1 hr. Purified Jhd2-FLAG was used as a positive control.

Page 5, line 109: Changed “To examine the effect of glucose on histone modifications, we grew yeast cells in medium containing different concentrations of glucose and found that glucose can induce H3K4me3 but not H3K4me1 and H3K4me2 in a dose-dependent manner (Fig. 1a).” to “To examine the effect of glucose on histone modifications, we grew yeast cells in 2% glucose-containing medium (YPD) until log phase. Cells were collected and then grown in medium containing different concentrations of glucose for 0.5 hr. By Western blot analysis of known histone modifications, we found that glucose induced H3K4me3 but not H3K4me1 and H3K4me2 in a dose-dependent manner (Fig. 1a). Other active histone markers, such as H3K36me3 and H3K79me3 were not significantly increased by glucose (Supplementary Fig. 1a)”.

Page 6, line 117: Added “we first examined the effect of glucose on histone modifications that regulate Set1-catalyzed H3K4me3, including H2BK123 monoubiquitination (H2Bub) and H3R2 asymmetric dimethylation (H3R2me2a)^{26,27,28}. Glucose induced H2Bub but had no significant effect on H3R2me2a (Supplementary Fig. 1a). We then determine whether glucose increases H3K4me3 by inducing H2Bub. As there is no H3K4me3 in the deletion mutants of *RAD6* and *BRE1*, which catalyze H2Bub^{26,27}, it is impossible to determine the effect of glucose on H3K4me3 in *rad6Δ* and *bre1Δ* mutants. We thus examined the effect of glucose on H3K4me3 in the deletion mutant of *UBP8* and *UBP10*, which encode H2B ubiquitin proteases^{29,30}. Although H2Bub remained relatively high and constant in *ubp8Δ ubp10Δ* mutant, H3K4me3 was still increased by glucose (Supplementary Fig. 1b), suggesting there is a H2Bub-independent signaling pathway for glucose to induce H3K4me3. We also examined the effect of glucose on the activity of Set1-containing complex (COMPASS). COMPASS

(Spp1-CBP) was purified from cells grown in 0.05% glucose and 2% glucose, respectively and the *in vitro* histone methyltransferase (HMT) assay showed that glucose did not affect the HMT activity of COMPASS (Supplementary Fig. 1c).”

Page 8, line 166: Added “we first examined the effect of Tpk2 on H2Bub and H3R2me2a that regulate Set1-catalyzed H3K4me3. Loss of Tpk2 did not affect these two modifications (Supplementary Fig. 2a)”.

Page 8, line 172: Added “We then examined the effect of Tpk2 on the expression and activity of Set1-containing complex, COMPASS. Loss of Tpk2 had no significant effect on the protein levels of Set1 and COMPASS subunits (Supplementary Fig. 2c, d). We also purified COMPASS from WT and the *tpk2Δ* mutant. Loss of Tpk2 did not affect the subunit composition of COMPASS or its histone methyltransferase activity (Supplementary Fig. 2e, f).”

2) Throughout the manuscript, the authors cite some of the published studies on Jhd2 while ignoring many other relevant ones. Even with the cited studies, they completely disregard or not account for nor describe their results with the reported findings. Many references appear simply inserted for the sake of citing!

In the revised manuscript, we cited the references of Jhd2 (*Genes Dev* 2009, 23:951-62; *Sci Rep* 2016, 6:37942) to show that loss of Jhd2 only mildly increases the global H3K4me3 when grown in 2% glucose media. Loss of Jhd2 resulted in prominent increased H3K4me3 when cells were grown in media containing acetate instead of glucose as a sole carbon source (*Sci Rep* 2016, 6:37942). We also cited and compared our results to the study by Huang and colleagues, which showed that deletion of the region between PHD and JmjC domains leads to Jhd2 protein instability (*J Biol Chem* 2010, 285:24548-61). In addition, we compared our Jhd2 mass spec data to that performed by Blair et al. (*Sci Adv* 2016, 2: e1501662) and discussed why they did not report Tpk2 as a Jhd2-interacting protein.

Page 4, line 91: Changed “we and others noticed that loss of Jhd2 increases H3K4me3 only when cells were grown under low glucose conditions ²¹” to “Although H3K4 is demethylated by Jhd2, loss of Jhd2 only mildly increases H3K4me3 when cells were grown under standard 2% glucose conditions ^{24,25}. Loss of Jhd2 strongly increases H3K4me3 when cells were grown in glucose-depletion media with acetate as the sole carbon source (YPA) ²⁵”.

Page 15, line 333: Added “As Jhd2 phosphorylation sites (S321, S340) are located between PHD and JmjC domains (Fig. 2e), our data is hence consistent with the study by Huang and colleagues that deletion of the region between PHD and JmjC domains leads to Jhd2 protein instability ⁴².”

Page 25, line 563: Added “Blair et al. performed a mass spectrometry analysis of proteins co-purified with Jhd2 and Tpk2 was not detected in their study⁴⁹. This could result from the use of different digestion methods and different mass spectrometers used. By using a combination of techniques, including Co-IP, *in vitro* kinase assay and a Shokat allele of Tpk2, we confirmed that Tpk2 can phosphorylate Jhd2.”

3) Decrease in H3K4me3 levels could be due to increased Jhd2 H3K4 demethylase activity or owing to decreased Set1 H3K4 methyltransferase activity. The authors show that Set1 levels remain unchanged upon loss of Tpk2 (Extended data Fig.2b). In yeast, Set1 is part of a multi-protein complex called COMPASS and many subunits of this complex such as Spp1/Cps40 regulate H3K4me3. However, the authors do not address whether absence of Tpk2 adversely affects the levels of COMPASS subunits to decrease H3K4me3.

We thank the reviewer for this important suggestion. We performed the following experiments to show that Tpk2 has no effect on the expression, subunit composition and/or activity of COMPASS:

1. We examined the effect of *TPK2* deletion on the expression of COMPASS subunits, including Set1, Swd1, Swd2, Swd3, Sdc1, Spp1 and Bre2. Our data showed that their protein levels were not changed by loss of Tpk2 (**Supplementary Fig. 2c, 2d**).
2. We purified COMPASS from WT and the *tpk2Δ* mutant. By silver staining of purified COMPASS, we did not observe any difference in the subunit composition (**Supplementary Fig. 2e**).
3. We performed *in vitro* histone methyltransferase (HMT) assay with COMPASS purified from WT and the *tpk2Δ* mutant. Our data showed that the HMT activity of COMPASS was not regulated by Tpk2 (**Supplementary Fig. 2f**).
4. We purified COMPASS from cells grown in YP media containing 0.05% glucose and 2% glucose, respectively. Our data showed that the enzymatic activity of COMPASS was not regulated by glucose (**Supplementary Fig. 1c**).

Collectively, the above data indicate that loss of Tpk2 does not adversely affect the expression, composition and activity of COMPASS to reduce H3K4me3. In the revised manuscript, we added the above information in the results section.

Page 8, line 172: Added “We then examined the effect of Tpk2 on the expression and activity of Set1-containing complex, COMPASS. Loss of Tpk2 had no significant effect on the protein levels of Set1 and COMPASS subunits (Supplementary Fig. 2c, d). We also purified COMPASS from WT and the *tpk2Δ* mutant. Loss of Tpk2 did not affect the subunit composition of COMPASS or its histone methyltransferase activity (Supplementary Fig. 2e, f).”

4) In Fig.2, the authors report the results of their affinity purification and mass

spectrometry (AP-MS) experiment to identify Jhd2 interacting proteins. A) The authors do not show relevant data, such as, silver stained gels of their affinity purification experiment (minor). B) Using the TAP tag purification strategy as reported in this manuscript, Jhd2 was previously reported to interact with the polyadenylation machinery by Blair et al. (Sci. Adv. 2016; 2: e1501662), but Tpk2 was not reported as a Jhd2-interacting protein. How do authors reconcile with this discrepancy in their results with that in the published work?

We thank this reviewer for this comment. In the revised manuscript, we provided the silver staining gels for purified Tpk2-TAP and Jhd2-TAP in Fig. 2a.

By comparing our IP-MS to the method by Blair et al. (Sci Adv 2016, 2: e1501662), we noticed the following differences, which may explain why Tpk2 was not reported as a Jhd2-interacting protein in their study:

1. They excised gel into 30 bands and then digested gel bands using trypsin, which may lead to loss of proteins. In our IP-MS protocols, we did not digest gel bands but instead we digested the immunoprecipitated proteins.
2. We used a different mass spectrometer. Compared with Thermo LTQ-XL Linear Ion Trap MS used by Blair et al., we used Thermo Q Exactive HF MS, which is a more recent model and has a relatively fast scan rate.
3. Blair et al. did not provide the full list of proteins interacting with Jhd2 in their paper. As we have no access to their raw data, it is impossible to tell whether Tpk2 is not a Jhd2-interacting protein.
4. Using a combination of techniques, including Co-IP, *in vitro* kinase assay and a Shokat allele of Tpk2, we confirmed that Tpk2 can interact with and phosphorylate Jhd2.

Taken together, we do not think our data contradict with the study by Blair et al. In the revised manuscript, we added the above possible explanation in the discussion section.

Page 25, line 563: Added “Blair et al. performed a mass spectrometry analysis of proteins co-purified with Jhd2 and Tpk2 was not detected⁴⁹. This could result from the use of different digestion methods and different mass spectrometers used. By using a combination of techniques, including Co-IP, *in vitro* kinase assay and a Shokat allele of Tpk2, we confirmed that Tpk2 can phosphorylate Jhd2.”

5) One assumes that the logic behind the AP-MS and co-IP experiments is that because Jhd2 interacts with Tpk2, it must be a substrate of Tpk2 kinase. However, it is well known that kinase-substrate interactions are transient and as such, standard immunoprecipitation do not retain these interactions. Generally, a Shokat allele of Tpk2 and use of a bulky ATP analog can show that Jhd2 is a substrate of Tpk2. The authors can refer to a review by Sergio et al (Science Signaling 2016 Vol. 9, Issue 420, pp. re3). The discovery of a role for the Ras-cAMP-PKA (Tpk2) pathway in regulating

H3K4me3 is the most interesting finding in this paper. Therefore, the role for phosphorylation of Jhd2 by Tpk2 can be further strengthened and highlighted in this manuscript.

We appreciate the reviewer for recognition the role of the Ras-cAMP-PKA (Tpk2) pathway in regulating H3K4me3. We thank the reviewer for suggesting the Shokat allele of Tpk2 to strength the conclusion that Tpk2 can phosphorylate Jhd2. We agree with the reviewer that the kinase-substrate interactions may be transient and standard immunoprecipitation may not be able to retain these interactions.

To strength the role of Tpk2 on Jhd2 phosphorylation, we constructed the Shokat allele of Tpk2 (Tpk2-as) by mutating M147A within its kinase active site as suggested. Using purified Wt Tpk2 and Tpk2-as mutant, we performed *in vitro* kinase assay. Our data showed that Jhd2 can be phosphorylated by Wt Tpk2 and Tpk2-as. By adding the bulky ATP analog, 1NM-PP1, we observed that the ability of Tpk2-as to phosphorylate Jhd2 was remarkably reduced (**Fig. 2j**). Moreover, 1NM-PP1 reduced the intracellular Jhd2 phosphorylation and H3K4me3 in Tpk2-as mutant but not WT cells (**Supplementary Fig. 3h, 4c**). As a control, we also made a Shokat allele of Tpk1 (Tpk1-as) and Tpk3 (Tpk3-as) by mutating M164A and M165A within their kinase active sites, respectively. 1NM-PP1 had no effect on intracellular H3K4me3 in both Tpk1-as and Tpk3-as mutants (**Supplementary Fig. 4c**).

Taken together, we are confident that Tpk2 can act as a protein kinase to phosphorylate Jhd2 at S321 and S340. In the revised manuscript, we added the data for the Shokat allele of Tpk1, Tpk2 and Tpk3 (**Fig. 2j; Supplementary Fig. 3h, 4c**).

Page 10, line 209: Added “To further verify that Tpk2 phosphorylates Jhd2, we constructed a Shokat allele of Tpk2 (Tpk2-as) by mutating M147A within its kinase active site, whose kinase activity can be inhibited by adding the bulky ATP analog, 1NM-PP1^{39,40}. The intracellular Jhd2 phosphorylation was significantly reduced in Tpk2-as mutant but not WT cells upon treatment with 1NM-PP1 (Supplementary Fig. 3h). The *in vitro* kinase assay also showed that 1NM-PP1 directly inhibited Tpk2-catalyzed Jhd2 phosphorylation (Fig. 2j, lanes 4-5).”

Page 11, line 243: Added “The intracellular H3K4me3 was significantly reduced in Tpk2-as mutant but not Tpk1-as and Tpk3-as mutants (Supplementary Fig. 4c), suggesting that the kinase activity of Tpk2 is required to promote H3K4me3.”

6) To extend on the point above, from AP-MS, the authors report that Jhd2 also interacts with Tpk1 and Tpk3, but only loss of Tpk2 decreases Jhd2 phosphorylation. How do they reconcile with these findings?

We thank this reviewer for this comment. It is true that we detected Tpk1 and Tpk3 co-

purified with Jhd2 in our IP-MS data (**Fig. 2a**). To test whether Tpk1 and Tpk3 phosphorylate Jhd2, we examined the phosphorylation of Jhd2 in WT, *tpk1Δ* and *tpk3Δ* mutants. Our data showed that Jhd2 phosphorylation was unaffected by loss of Tpk1 and Tpk3 (**Supplementary Fig. 3f**). By using the Shokat allele of Tpk1, Tpk2 and Tpk3 treated with or without the inhibitor, 1NMPP1, we observed that inhibition of Tpk2 but not Tpk1 and Tpk3 reduced the intracellular H3K4me3 (**Supplementary Fig. 4c**). All these data confirm that only Tpk2 phosphorylates Jhd2 and inhibits its activity. Nonetheless, we do not think these data are inconsistent for the following reasons:

1. It is possible that only Tpk2 directly interacts with Jhd2. We performed the Co-IP to examine the interaction between Jhd2 and Tpk2 in WT, *tpk1Δ* and *tpk3Δ* mutants. Loss of Tpk1 and Tpk3 had no effect on the interaction between Tpk2 and Jhd2 (**Rebuttal letter Fig. 2**). It is likely that Tpk2 directly interact with Jhd2, while Tpk1 and Tpk3 indirectly interact with Jhd2 via Tpk2.
2. Although the PKA family kinases Tpk1, Tpk2 and Tpk3 interact with each other, they tend to have distinct substrate specificities. The Michael Snyder's group performed a global analysis of protein phosphorylation in yeast (*Nature* 2005, 438:679-84). Their data showed that these three kinases have only 8 common substrates. The vast majority of substrates are recognized by only one of the Tpk's. Moreover, loss of these three PKA have distinct phenotypes. For example, Tpk2 stimulates pseudohyphal morphogenesis, whereas Tpk3 inhibits filamentation and Tpk1 has no effect (*Proc Natl Acad Sci USA* 1998, 95: 13783-13787). Our data showed that loss of Tpk2, but not Tpk1 and Tpk3 reduced the transcription of autophagy genes as well as autophagy activity (**Fig. 7h-j; Supplementary Fig. 9e, 9g, 9h**). Thus, our study provides another example for PKA family kinases Tpk1, Tpk2 and Tpk3 performing different functions by phosphorylating distinct substrates.

Taken together, we identified a novel role of Tpk2 in promoting glucose-induced H3K4me3. Moreover, our data provide a new example for PKA family kinases Tpk1, Tpk2 and Tpk3 performing different functions by phosphorylating distinct substrates.

Rebuttal letter Fig. 2. Co-IP assay showing that the interaction between Jhd2 and Tpk2 was unaffected by loss of Tpk1 and Tpk3. Tpk2-FLAG was immunoprecipitated from

untagged (control), Tpk2-FLAG, Tpk2-FLAG *tpk1Δ*, and Tpk2-FLAG *tpk3Δ* cells when grown in YPD.

Page 27, line 612: Added “Although our mass spectrometry data detected Tpk1, Tpk2 and Tpk3 co-purified with Jhd2, only Tpk2 phosphorylates Jhd2 to inhibit its activity. It is possible that Tpk2 directly interacts with Jhd2, Tpk1 and Tpk3 indirectly interact with Jhd2. The different behavior of Tpk isoforms on Jhd2 is consistent with the study performed by Ptacek et al. that three Tpk3s have distinct substrate specificities and most substrates are recognized by only one of the Tpk3s³³. Tpk1, but not Tpk2 and Tpk3 regulates non-homologous end joining double-stranded break repair by phosphorylating Nej1³⁴. Tpk2 activates pseudohyphal growth, while Tpk3 inhibits filamentation and Tpk1 has no effect³⁵. Tpk1 and Tpk2 have distinct functions in regulating iron uptake and respiration⁵⁵. In contrast to the role of Tpk2 in activating autophagy, our data showed that loss of Tpk1 and Tpk3 had no significant effect on the transcription of autophagy genes as well as autophagy activity. Thus, our data provide an example that PKA family kinases Tpk1, Tpk2 and Tpk3 performing different functions by phosphorylating distinct substrates.”

7) Immunoblots using antibodies raised against S321 or S340 phosphorylated Jhd2 implicate Tpk2 as the effector kinase. However, the specificity of these antibodies appear to be very poor and recognize the backbone, as seen from the signal in the Jhd2-S321AS340A mutant (Fig.2i). In addition, the authors refer to the Jhd2-S321AS340A double mutant as Jhd2-S2A, which is completely misleading and confusing!

We thank the reviewer for this comment. To improve the specificity of antibodies, we re-purified these two antibodies with Jhd2-S321 phosphorylated and Jhd2-S340 phosphorylated peptides and then used in Western blots. Using these re-purified antibodies, we obtained data with little to no background (**Fig. 2i**).

We changed “Jhd2-S2A” to “Jhd2-2SA” to stand for Jhd2-S321AS340A in the revised manuscript to prevent any confusion. We feel sorry for this error.

8) Why were the phosphomimetic Jhd2-S321D/S340D single or double mutants not included in the study? It seems logical to include these gain-of-function alleles along with the loss-of-function Jhd2-S321AS340A mutant.

We thank this reviewer for this good comment. We constructed the phosphomimetic Jhd2-S321D, Jhd2-S340D and Jhd2-S321DS340D (Jhd2-2SD) mutants. By Western blots, we found that while the intracellular H3K4me3 was significantly lower in Jhd2-2SA than WT, H3K4me3 was significantly higher in Jhd2-2SD mutant than WT (**Fig. 3a**). Meanwhile, we mutated Jhd2-S321DS340D in *tpk2Δ* mutant to get Jhd2-2SD

tpk2Δ mutant. We clearly see that the intracellular H3K4me3 was significantly reduced in *tpk2Δ* mutant but significantly increased in Jhd2-2SD and *tpk2Δ* Jhd2-2SD mutant (Fig. 3b). Therefore, Jhd2-2SA could function as a loss-of-function mutant and Jhd2-2SD could act as a gain-of-function mutant. Both mutants indicate that Tpk2 can inhibit the activity of Jhd2 by phosphorylating Jhd2 at S321 and S340.

Page 11, line 240: Added “We also constructed Jhd2 phosphomimic mutants, Jhd2-S321D, Jhd2-S340D, Jhd2-S321D S340D (Jhd2-2SD) and found that the global levels of H3K4me3 were increased in these Jhd2 phosphomimic mutants (Fig. 3a).”

Page 11, line 245: Changed “Similarly, loss of Tpk2 reduced the global levels of H3K4me3 (Fig. 3b). Deletion of *TPK2* in Jhd2 S2A mutant did not further reduce H3K4me3 (Fig. 3b)” to “Loss of Tpk2 significantly reduced the global levels of H3K4me3; deletion of *TPK2* in Jhd2-2SA mutant did not further reduce H3K4me3 and deletion of *TPK2* in Jhd2-2SD mutant did not further increase H3K4me3 (Fig. 3b; Supplementary Fig. 4d).”

9) The linker region containing the S321 and S340 residues was previously reported to be crucial for the stability of Jhd2. The authors do not refer to this published study!

We thank this reviewer for this point. In the revised manuscript, we cited the published study (*J Biol Chem* 2010, 285: 24548-24561) and added the following sentence: “As Jhd2 phosphorylation sites (S321, S340) are located between PHD and JmjC domains (Fig. 2e), our data is hence consistent with the study by Huang and colleagues that deletion of the region between PHD and JmjC domains leads to Jhd2 protein instability⁴².”

Page 9, line 188: Changed “which are located within the consensus PKA phosphorylation motif in Jhd2 (Fig. 2e)” to “which are located within the consensus PKA phosphorylation motif between the PHD and JmjC domains of Jhd2 (Fig. 2e)”.

Page 15, line 333: Added “As Jhd2 phosphorylation sites (S321, S340) are located between PHD and JmjC domains (Fig. 2e), our data is hence consistent with the study by Huang and colleagues that deletion of the region between PHD and JmjC domains leads to Jhd2 protein instability⁴².”

10) In Fig 5e, the authors label high molecular weight bands for Jhd2 as Jhd2-(Ub)_n. In order to be labeled as polyubiquitinated forms, these high molecular weight Jhd2 bands need to cross-react with either an antibody recognizing ubiquitin or one recognizing mono- or poly-ubiquitinated proteins (clone FK2).

In our original manuscript, the high molecular weight Jhd2 bands were detected with

an antibody recognizing ubiquitin. We apologize that we did not label it clearly. In the revised manuscript, we also detected the high molecular weight Jhd2 bands with the antibody recognizing poly-ubiquitinated proteins (clone FK2) as recommended. Our data confirmed that these high molecular weight bands are the poly-ubiquitinated form of Jhd2 (**Fig. 5e**).

11) Using AP-MS, the authors report that Jhd2 interacts with Rpd3, and that this interaction is increased in the Jhd2-S321AS340A mutant. This observation is based on obtaining two and five peptides for Rpd3 following anti-Flag immunoprecipitation. How do we know this is significant? Their AP-MS experiments are not at all quantitative, as either SILAC or isobaric labeling approach was not used!

We thank this reviewer for this comment. It is true that we did not use quantitative mass spec in this work but as a tool of discovery. However, our Co-IP assay showed that the interaction between Rpd3 and Jhd2 was increased in Jhd2-2SA mutant (**Fig. 6e**), which confirmed the conclusion. To prevent any confusion, we changed the sentence about mass spec data from “We then immunoprecipitated WT Jhd2 and Jhd2 S2A and LC-MS analysis revealed that Rpd3 was a Jhd2-interacting protein and more Rpd3 was detected in Jhd2 S2A purification (Fig. 6d)” to “We then immunoprecipitated WT Jhd2 and Jhd2-2SA and LC-MS analysis revealed that Rpd3 was a Jhd2-interacting protein (Fig. 6d)”.

Page 18, line 394: Changed “We then immunoprecipitated WT Jhd2 and Jhd2 S2A and LC-MS analysis revealed that Rpd3 was a Jhd2-interacting protein and more Rpd3 was detected in Jhd2 S2A purification (Fig. 6d)” to “We then immunoprecipitated WT Jhd2 and Jhd2-2SA and LC-MS analysis revealed that Rpd3 was a Jhd2-interacting protein (Fig. 6d)”.

12) According to authors, loss of Jhd2 S321S340 phosphorylation results in decreased H3K4me3 as the Jhd2-S321AS340A mutant binds H3 peptide more tightly. They then report that Jhd2 interacts with Rpd3 deacetylase and because Jhd2-S321AS340A binds chromatin more so does Rpd3. In yeast, Rpd3 is also present as a component of multi-protein complexes (Rpd3S, Rpd3L, Rpd3 μ , etc.). In yeast, Pho23 and Rxt1, both components of Rpd3L complex directly bind to H3K4me3 via their PHD finger. How do the authors reconcile their observations of increased Rpd3 binding upon decrease in H3K4me3?

We thank this reviewer for this comment. The TaeSoo Kim group reported that H3K4me3 can recruit Rpd3L complex to gene promoters to repress gene transcription when grown in galactose-containing media, so called transcription repression memory (*Nucleic Acids Res* 2018, 46:8261-8274). The Jin-Qiu Zhou group reported that the PHD fingers of Pho23 and Rxt1 recognize H3K4me3 and recruit Rpd3 to *PHO5* to

repress its transcription when grown in high phosphate media (*Mol Cell Biol* 2011, 31:3171-81). However, we do not think our results conflict with their observations for the following reasons:

1. Cells were grown in different media in these studies. In the transcription memory study by the TaeSoo Kim group, cells were initially grown in SC medium containing raffinose, shifted to a media containing galactose (120 min), then shifted to glucose-containing media for 120 min, and then shifted back in galactose-containing media for 30 min (**Rebuttal letter Fig. 3a**). Under these conditions, some galactose-repressed genes, i.e., *REI1*, *TEA1* and *RRN11* are repressed faster and stronger by galactose (Gal^{2nd}) if they were initially repressed by galactose (**Rebuttal letter Fig. 3b**). This repression memory phenomenon is mediated by Rpd3L complex, where Pho23 recognizes H3K4me3 and recruit Rpd3 to genes to repress their transcription. Deletion of *PHO23* and *RPD3* significantly increased the expression of these genes when cells were grown in galactose-containing media but not when cells were grown in glucose-containing media (**Rebuttal letter Fig. 3b, 3c**). The repression of *PHO5* by Rpd3 was performed when cells were grown in high phosphate conditions (*Mol Cell Biol* 2011, 31:3171-81). In our study, we focused on the effect of glucose on gene transcription when cells were grown in glucose-containing media.
2. Different genes were studied in these studies. In the transcription memory study by the TaeSoo Kim group, they focused on galactose-repressed genes, i.e., *REI1*, *TEA1* and *RRN11*. These genes were repressed by Rpd3 and Pho23 but not Jhd2-2SA in galactose-containing media (**Rebuttal letter Fig. 3b**). In the study by the Jin-Qiu Zhou group, they focused on *PHO5*. In our study, we focused on genes co-regulated by Rpd3 and Jhd2-2SA, such as autophagy genes and other genes (*EGR28*, *NCS2*). These genes were not regulated by Pho23 (**Rebuttal letter Fig. 3b, c; Supplementary Fig. 9f**). We also performed ChIP-qPCR to examine the binding of Rpd3 at *EGR28*, *NCSS*, and *ATG* genes in WT, Jhd2-2SA, *pho23Δ*, *rxt1Δ*, Jhd2-2SA *pho23Δ* and Jhd2-2SA *rxt1Δ* mutants. Our data showed that Pho23 and Rxt1 are not required for binding of Rpd3 at these genes in both WT and Jhd2-2SA mutant (**Rebuttal letter Fig. 3d; Supplementary Fig. 7i, 9d**). As a positive control, Pho23 and Rxt1 are required for binding of Rpd3 at *REI1* and *TEA1* (**Rebuttal letter Fig. 3d**).
3. The interaction between Jhd2 and Rpd3 is independent on Pho23 and Rxt1. We examined the interaction between Jhd2 and Rpd3 in WT, *pho23Δ* and *rxt1Δ* mutants by Co-IP assay. Our data showed that Pho23 and Rxt1 did not affect the interaction between Jhd2 and Rpd3 (**Rebuttal letter Fig. 3e**).

Collectively, these three studies suggest that Rpd3 is recruited to different subset of genes by different mechanisms under different growth conditions. In the revised manuscript, we added the data for the effect of Pho23 and Rxt1 on Rpd3 occupancy at *NCS2*, *EGR28* and *ATG* genes (**Supplementary Fig. 7i, 9d**). We also added the RT-qPCR data for the transcription of *ATG* genes in WT and *pho23Δ* mutant (**Supplementary Fig. 9f**). In addition, we added the following sentences in the discussion section: “Rpd3 has been reported to be recruited by Pho23 and Rxt1 to a

subset of genes such as *PHO5*, *REI1*, *TEA1* and *RRN11*^{43,44}. H3K4me3 promotes the binding of Rpd3 via the PHD fingers of Pho23 and Rxt1 to repress the transcription of these genes^{43,44}. However, loss of Pho23 and Rxt1 had no significant effect on Rpd3 occupancy at genes co-regulated by Jhd2-2SA and Rpd3, such as autophagy genes, *NCS2* and *EGR28*, suggesting that Rpd3 is recruited to different genes by different mechanisms.”

Rebuttal letter Fig. 3. **a** Schematic representation of the experiments to study transcription repression during carbon-source shifts. **b** RT-qPCR analysis the transcription of *REI1*, *ATG39* and *ATG9* in WT, Jhd2-2SA, *pho23Δ* and *rpd3Δ* upon carbon-source shifts. **c** RT-qPCR analysis the transcription of *ATG9*, *ATG39*, *REI1* and

TEA1 in WT, Jhd2-2SA, *pho23Δ* and *rpd3Δ* mutants when cells were grown in glucose-containing media. **d** ChIP-qPCR analysis of Rpd3 binding at *ATG9* and *ATG39* in WT, Jhd2-2SA, *pho23Δ*, *rxt1Δ*, Jhd2-2SA *pho23Δ* and Jhd2-2SA *rxt1Δ* mutants when cells were grown in glucose-containing media. ChIP-qPCR analysis of Rpd3 binding at *REI1* and *TEA1* in WT, Jhd2-2SA, *pho23Δ*, *rxt1Δ*, Jhd2-2SA *pho23Δ* and Jhd2-2SA *rxt1Δ* mutants when cells were grown in galactose-containing media. **e** Co-IP assay showing the interaction between Rpd3 and Jhd2 was unaffected by loss of Pho23 and Rxt1. Jhd2-FLAG was immunoprecipitated from untagged, Jhd2-FLAG (WT), Jhd2-FLAG *pho23Δ* and Jhd2-FLAG *rxt1Δ* mutants when grown in YPD media.

Page 19, line 425: Added “Rpd3 has been reported to be recruited by Pho23 and Rxt1 to repress the expression of *PHO5* under high phosphate conditions ⁴³ and repress the transcription of galactose-repressed genes, such as *REI1*, *TEA1* and *RRN11* ⁴⁴. However, loss of Pho23 and Rxt1 had no significant effect on Rpd3 occupancy at *NCS2* and *EGR28* (Supplementary Fig. 7i), suggesting that Rpd3 is recruited to different subsets of genes by distinct mechanisms.”

Page 22, line 481: Added “Loss of Pho23 and Rxt1 had no significant effect on Rpd3 occupancy at *ATG* genes (Supplementary Fig. 9d)”.

Page 22, line 491: Added “Loss of Pho23 had no significant effect on the transcription of *ATG* genes (Supplementary Fig. 9f)”.

Page 26, line 586: Added “Rpd3 has been reported to be recruited by Pho23 and Rxt1 to a subset of genes such as *PHO5*, *REI1*, *TEA1* and *RRN11* ^{43,44}. H3K4me3 promotes the binding of Rpd3 via the PHD fingers of Pho23 and Rxt1 to repress the transcription of these genes ^{43,44}. However, loss of Pho23 and Rxt1 had no significant effect on Rpd3 occupancy at genes co-regulated by Jhd2-2SA and Rpd3, such as autophagy genes, *NCS2* and *EGR28*, suggesting that Rpd3 is recruited to different genes by different mechanisms.”

13) In the introduction, the authors' state: “we and others noticed that loss of Jhd2 increases H3K4me3 only when cells were grown under low glucose conditions 21”. However, many studies have clearly shown increased H3K4me3 in *jhd2Δ* strain even in standard glucose (2%) or glucose-replete media. In the reference cited #21, Figure 2d, increase in H3K4me3 is seen in *jhd2Δ* strain grown in glucose rich medium! So it is unclear what the authors mean by increase is see only when cells are grown under low glucose conditions!

We feel sorry for the confusion caused by our writing. We agree with the reviewer that loss of Jhd2 increases H3K4me3 even when cells were grown in standard glucose (2%) media. However, loss of Jhd2 only mildly increases the global H3K4me3 when grown

in 2% glucose media (*Genes Dev* 1999, 23:951-962; *Sci Rep* 2016, 6:37942). When cells were grown in media containing acetate instead of glucose as a sole carbon source, loss of Jhd2 resulted in prominent increased H3K4me3 (*Sci Rep* 2016, 6:37942). We also compared H3K4me3 in WT and *jhd2Δ* mutant when grown in different glucose conditions. Our data showed that the difference of H3K4me3 between WT and *jhd2Δ* mutant is more prominent when cells were grown in low glucose medium (0.05%) than grown in standard glucose medium (2%) (**Rebuttal letter Fig. 4**). These data suggest that Jhd2-mediated H3K4 demethylation is partly, if not completely repressed in standard glucose-containing media.

In the revised manuscript, we changed the sentence from “we and others noticed that loss of Jhd2 increases H3K4me3 only when cells were grown under low glucose conditions” to “Although H3K4 is demethylated by Jhd2, loss of Jhd2 only mildly increases H3K4me3 when cells were grown under standard 2% glucose (YPD) conditions^{24,25}. Loss of Jhd2 strongly increases H3K4me3 when cells were grown in glucose-depletion media with acetate as the sole carbon source (YPA)²⁵, implying that Jhd2 may be regulated by an unknown mechanism in response to glucose availability.” We also replaced reference 21 with references 24 and 25.

Rebuttal letter Fig. 4. Western blot analysis the effect of glucose on H3K4me3 in WT and *jhd2Δ* mutant. WT and *jhd2Δ* mutant were grown in YPD medium until OD₆₀₀ of 0.7-1.0. Cells were then treated in YP medium supplemented with different concentrations of glucose for 0.5 hr. Cells were harvested and the extracted histones were analyzed by Western blots with indicated antibodies.

Page 4, line 91: Changed “we and others noticed that loss of Jhd2 increases H3K4me3 only when cells were grown under low glucose conditions²¹, implying that Jhd2 may be regulated by an unknown mechanism in response to glucose availability.” to “Although H3K4 is demethylated by Jhd2, loss of Jhd2 only mildly increases H3K4me3 when cells were grown under standard 2% glucose (YPD) conditions^{24,25}. Loss of Jhd2 strongly increases H3K4me3 when cells were grown in glucose-depletion media with acetate as the sole carbon source (YPA)²⁵, implying that Jhd2 may be regulated by an unknown mechanism in response to glucose availability.”

Page 25, line 551: Changed “The relatively low activity and expression of Jhd2 in rich

media could explain the long-standing question: why does loss of Jhd2 have no significant effect on global H3K4me3?” to “The relatively low activity and expression of Jhd2 in rich media could explain the long-standing question: why does loss of Jhd2 have mild effect on global H3K4me3 when cells were grown in standard glucose media?”.

Reviewer #2 (Remarks to the Author):

The manuscript submitted by Yu et al. aimed to identify the signaling pathway that regulates histone modifications in response to glucose availability. By screening glucose sensor mutants, they found that the PKA/Tpk2 is required for glycolysis to promote H3K4me3. Mass spec and in vitro kinase assay showed Tpk2 directly phosphorylates H3K4 demethylase Jhd2 at S321 and S340. Tpk2-catalyzed Jhd2 phosphorylation inhibits the demethylase activity of Jhd2 by reducing its binding at chromatin. Moreover, Tpk2-catalyzed Jhd2 phosphorylation impairs its nuclear localization and promotes its degradation. Interestingly, they showed that Tpk2-catalyzed Jhd2 phosphorylation reduced the binding of Rpd3 to chromatin, which coordinately regulates the crosstalk between H3K4me3 and H3K14ac. Finally, they showed that Tpk2-catalyzed Jhd2 phosphorylation regulates gene expression, chronological lifespan and autophagy.

The strengths of this paper include the use of multiple in vivo and in vitro approaches as well as the creation of unique reagents, such as Jhd2 S321 and S340 phosphorylation specific antibodies and Jhd2 phosphorylation site mutations to confirm Tpk2-mediated Jhd2 phosphorylation and explore its effect on protein-protein interaction and activity. These new findings reveal exciting connections between glucose metabolism and histone modifications that is of significant interest to both metabolism and chromatin community.

We thank the reviewer for recognition the novelty of our work. In the revised manuscript, we addressed the role of three PKAs on H3K4me3 and autophagy activity. We also examined the role of Not4 on Rpd3 protein level. In addition, we re-purified Tpk2 and re-did the mass spec analysis. We hope that this reviewer would find our manuscript was significantly improved.

1. Since there are three protein kinase A (PKAs, Tpk1/2/3) in yeast, why only Tpk2 but not Tpk1 or Tpk3 affects histone modifications H3K4me3? What is the effect of Tpk1 and Tpk3 on autophagy?

We thank this reviewer for this comment. It has been reported that the PKA family kinases Tpk1, Tpk2 and Tpk3 tend to have distinct substrate specificities. The Michael Snyder's group performed a global analysis of protein phosphorylation in yeast (*Nature*

2005, 438:679-84). Their data showed that only 8 substrates are recognized by three kinases. The vast majority of substrates are recognized by only one of the Tpk. In addition, loss of these three PKA have distinct phenotypes. For example, Tpk2p stimulates pseudohyphal morphogenesis, whereas Tpk1p and Tpk3p have a repressing effect (*Proc Natl Acad Sci U S A* 1998, 95(23): 13783-13787). Tpk1, but not Tpk2 and Tpk3 regulates non-homologous end joining double-stranded break repair by phosphorylating Nej1 (*Nucleic Acids Res* 2021, 49: 8145-8160).

We performed Co-IP to examine the interaction between Jhd2 and Tpk2 in WT, *tpk1Δ* and *tpk3Δ* mutants. We can clearly see that the Tpk2-Jhd2 interaction was unaffected by Tpk1 and Tpk3. It is possible that only Tpk2 directly interacts with Jhd2, Tpk1 and Tpk3 indirectly interact with Jhd2 via Tpk2 (**Rebuttal letter Fig. 2**).

Meanwhile, we examined the effect of Tpk1 and Tpk3 on autophagy. In contrast to the role of Tpk2 in activating autophagy, our data showed that loss of Tpk1 and Tpk3 had no significant effect on the transcription of autophagy genes (**Supplementary Fig. 9e**). Tpk1 and Tpk3 also had no effect on autophagy activity (**Supplementary Fig. 9h**). Thus, our data provide another example for PKA family kinases Tpk1, Tpk2 and Tpk3 performing different functions by phosphorylating distinct substrates.

Rebuttal letter Fig. 2. Co-IP assay showing that the interaction between Jhd2 and Tpk2 was unaffected by loss of Tpk1 and Tpk3. Tpk2-FLAG was immunoprecipitated from untagged (control), Tpk2-FLAG, Tpk2-FLAG *tpk1Δ*, and Tpk2-FLAG *tpk3Δ* cells when grown in YPD.

Page 27, line 613: Added “It is possible that Tpk2 directly interacts with Jhd2, Tpk1 and Tpk3 indirectly interact with Jhd2. The different behavior of Tpk isoforms on Jhd2 is consistent with the study performed by Ptacek et al. that three Tpk have distinct substrate specificities and most substrates are recognized by only one of the Tpk³³. Tpk1, but not Tpk2 and Tpk3 regulates non-homologous end joining double-stranded break repair by phosphorylating Nej1³⁴. Tpk2 activates pseudohyphal growth, while Tpk3 inhibits filamentation and Tpk1 has no effect³⁵. Tpk1 and Tpk2 have distinct

functions in regulating iron uptake and respiration ⁵⁵. In contrast to the role of Tpk2 in activating autophagy, our data showed that loss of Tpk1 and Tpk3 had no significant effect on the transcription of autophagy genes as well as autophagy activity. Thus, our data provide an example that PKA family kinases Tpk1, Tpk2 and Tpk3 performing different functions by phosphorylating distinct substrates.”

2. The regulation of Rpd3 by Jhd2 phosphorylation is interesting. The binding of Rpd3 to chromatin is enhanced in *tpk2Δ* and Jhd2 S2A mutant. But how Jhd2 phosphorylation reduced the binding of Rpd3 to chromatin is unknown. The author can provide plausible explanations in the discussion section. In addition, both H3K4me3 and H3K14ac are reduced in NOT4 deletion mutant. The reduced H3K4me3 is caused by increased Jhd2 stability and activity. As Jhd2 physically interacts with Rpd3, I am wondering whether Rpd3 can be degraded by Not4-related proteasome pathway.

We thank this reviewer for this comment. The enhanced Rpd3 binding to chromatin in *tpk2Δ* and Jhd2-2SA mutants could be caused by increased binding of Jhd2 at chromatin and enhanced interaction between Jhd2 and Rpd3 (**Fig. 4e, 4f, 4o, 6e, 6f**), which led to increased Rpd3 at chromatin (**Fig. 6i, 6j**). Tpk2-catalyzed Jhd2 phosphorylation reduced the binding of Jhd2 to chromatin as well as the interaction between Jhd2 and Rpd3, which results in decreased binding of Rpd3 at chromatin. In the revised manuscript, we added this description in the discussion section.

We examined whether Rpd3 can be degraded by Not4-related proteasome pathway. In the absence of Not4, there is no significant change of Rpd3 (**Fig. 5f**), indicating that Rpd3 is not degraded by Not4-mediated proteasome pathway.

Page 26, line 573: Added “The interaction between Jhd2 and Rpd3 could facilitate the binding of Rpd3 at chromatin. Tpk2-catalyzed Jhd2 phosphorylation reduced the binding of Jhd2 to chromatin as well as the interaction between Jhd2 and Rpd3, which results in decreased binding of Rpd3 at chromatin.”

3. The number of unique peptides for Tpk2 is low as a bait protein in purification and mass spectrometry in Fig. 2a. Although they used Co-IP and in vitro kinase assay to confirm their physical interaction, it is better to redo the purification and mass spec analysis.

In the revised manuscript, we re-did Tpk2 purification with a better yield. By mass spectrometry analysis, we detected more peptides and coverage for Tpk2 as a bait protein (**Fig. 2a**).

4. For ChIP-qPCR experiments of Fig. 4o and 4p: a negative control should be added

to underline the specificity of the three selected genes.

We thank this reviewer for this important suggestion. We added *PYK1* as a negative control to show the specificity of the three selected genes.

Minor points:

5. Fig. 4e and 6i: Non-specific band of H3 should be marked.

We added asterisk to indicate the non-specific band of H3.

6. Fig. 7J: The labels of statistical significance are different from labels in Fig. 7i.

We re-labeled the statistical significance in Fig. 7j to make it consistent with Fig. 7i.

Reviewer #3 (Remarks to the Author):

The manuscript entitles “Phosphorylation of Jhd2 by the Ras-cAMP-PKA(Tpk2) pathway regulates histone modifications and autophagy” the authors describe and clearly demonstrate a mechanism by which Jhd2 together with Rpd3 regulate transcription of the genes involved in normal life span maintenance and in the induction of autophagy. The manuscript shows a lot of work performed by the authors and the most important is that it provides a link between nutrient availability, glucose, and the mechanism involved in the transcription regulation of life span regulation and autophagy. this work highlights a novel regulatory mechanism that involves the phosphorylation of Jdh2 by the Tpk2 subunit of PKA.

The results obtained from the experiments provide significant support for the conclusions. The conclusions are significant as they have taken a step forward in the field. The genomic data is well analysed but some concerns about the analysis and validations are detailed below.

The text is well written and very clear, and in general the results have been provided with the adequate context.

The following are points that should be addressed to help improve the paper.

We appreciate the reviewer’s recognition the significance of our work. In the revised manuscript, we addressed all points raised by this reviewer, which improved the manuscript.

Introduction

1. Page 4- lines 61-63. “While AMPK is activated by low energy status and mediates

cellular responses to energetic stress, mTOR couples nutrient abundance to cell growth in response to nutrients, growth factors and cellular energy”.

The writing is not clear and confusing. Please change the description of both pathways in a way that it indicates the real differences between them.

We feel sorry for any confusion by our writing. In the revised manuscript, we changed “While AMPK is activated by low energy status and mediates cellular responses to energetic stress, mTOR couples nutrient abundance to cell growth in response to nutrients, growth factors and cellular energy” to “While AMPK is activated by low energy or lack of nutrients to inhibit cell growth, mTOR is activated by nutrient availability to promote cell growth ^{4,5}”.

Page 3, line 55: Changed “While AMPK is activated by low energy status and mediates cellular responses to energetic stress, mTOR couples nutrient abundance to cell growth in response to nutrients, growth factors and cellular energy” to “While AMPK is activated by low energy or lack of nutrients to inhibit cell growth, mTOR is activated by nutrient availability to promote cell growth ^{4,5}.”

2. Pag 4 -lines 63-66. The cellular process regulated by the cAMP-PKA pathway are poorly described. There are many papers and specially reviews on the subject. Reference 6 should be replaced.

We thank this reviewer for this important suggestion. We described the cAMP-PKA pathway in detail in the revised manuscript. In addition, we replaced reference 6 with a review paper as requested.

Page 3, line 57: Changed “The Ras/cAMP/PKA pathway is a nutrient-sensitive signaling pathway that regulates vegetative growth, carbohydrate metabolism, and entry into meiosis. Activated PKA exerts its cellular functions by phosphorylating its target proteins ⁶” to “PKA is a tetrameric holoenzyme, which is composed of a regulatory subunit Bcy1 dimer and two catalytic subunits encoded by *TPK1*, *TPK2* and *TPK3*. The glycolytic metabolite, fructose-58 1,6-biphosphate (FBP) activates small GTP-binding proteins, Ras1 and Ras2 through the guanine exchange factor, Cdc25 ⁶. Ras stimulates adenylate cyclase (Cyr1) to synthesize cAMP, which then activates PKA by binding to Bcy1 regulatory subunit to release the catalytic subunits ⁷. Activated PKA then exerts its cellular functions by phosphorylating its target proteins ⁷, which constitutes the Ras/cAMP/PKA pathway that regulates vegetative growth, carbohydrate metabolism, and entry into meiosis in response to glucose ⁷. In addition to the Ras/cAMP/PKA pathway, Cyr1 can also be stimulated by glucose-sensing G-protein-coupled receptor (GPCR) system, which is composed of G-protein-coupled receptor Gpr1, G protein α subunit Gpa2, and its GTPase activating protein Rgs2, so called the Gpr1/Gpa2 branch ³.”

3. Pag 4- line 70. Histones are not the “structure unit” of chromatin, this phrase should be modified since the basic repeating structural (and functional) unit of chromatin is the nucleosome.

We thank this reviewer for this comment. We changed “As the structure unit of chromatin, histones undergo posttranslational modifications, i.e., acetylation, methylation, phosphorylation, ubiquitination, which play important roles in regulating chromatin structure and gene expression” to “The well-known epigenetic modifications are histone post-translational modifications, including acetylation, methylation, phosphorylation, ubiquitination, which play important roles in regulating chromatin structure and gene expression” (Page 4, line 71).

4. Pag 5-line 93. Although the cAMP-PKA pathway was mentioned previously, in the paragraph which begins in line 93, it is the first time that Tpk2 appears. It would be necessary to provide a complete description of *Saccharomyces cerevisiae* PKA and the corresponding references, before this line. PKA, especially the catalytic subunit Tpk2, has a key role in the mechanism described in this work.

On the other hand, Tpk2 is not activated. The holoenzyme PKA is activated when cAMP is bound to Bcy1 subunits. Released Tpk subunits (all isoforms) could then phosphorylate their substrates

We thank this reviewer for this comment. In the revised manuscript, we added a detailed description of PKA in the introduction section (Page 3, line 57). We also described “three Tpk2s have distinct substrates and different functions in cellular processes” before Tpk2 in the result section (Page 6, line 137). In addition, we removed statement about “activated Tpk2” in the abstract.

Page 2, line 28: Changed “Herein, we report that glycolysis promotes H3K4me3 by activating Tpk2, a catalytic subunit of protein kinase A (PKA) via the Ras-cyclic AMP pathway. Activated Tpk2 antagonizes Jhd2-catalyzed H3K4 demethylation by phosphorylating Jhd2 at Ser321 and Ser340 in response to glucose availability.” to “Herein, we report that glycolysis promotes H3K4me3 by activating protein kinase A (PKA) via the Ras-cyclic AMP pathway. The catalytic subunit of PKA, Tpk2 antagonizes Jhd2-catalyzed H3K4 demethylation by phosphorylating Jhd2 at Ser321 and Ser340 in response to glucose availability”.

Page 3, line 57: Changed “The Ras/cAMP/PKA pathway is a nutrient-sensitive signaling pathway that regulates vegetative growth, carbohydrate metabolism, and entry into meiosis. Activated PKA exerts its cellular functions by phosphorylating its target proteins⁶” to “PKA is a tetrameric holoenzyme, which is composed of a regulatory subunit Bcy1 dimer and two catalytic subunits encoded by *TPK1*, *TPK2* and *TPK3*. The glycolytic metabolite, fructose-58 1,6-biphosphate (FBP) activates small

GTP-binding proteins, Ras1 and Ras2 through the guanine exchange factor, Cdc25⁶. Ras stimulates adenylate cyclase (Cyr1) to synthesize cAMP, which then activates PKA by binding to Bcy1 regulatory subunit to release the catalytic subunits⁷. Activated PKA then exerts its cellular functions by phosphorylating its target proteins⁷, which constitutes the Ras/cAMP/PKA pathway that regulates vegetative growth, carbohydrate metabolism, and entry into meiosis in response to glucose⁷. In addition to the Ras/cAMP/PKA pathway, Cyr1 can also be stimulated by glucose-sensing G-protein-coupled receptor (GPCR) system, which is composed of G-protein-coupled receptor Gpr1, G protein α subunit Gpa2, and its GTPase activating protein Rgs2, so called the Gpr1/Gpa2 branch³.”

Page 6, line 137: Added “These three Tpk3 have distinct substrates and different functions in cellular processes³³. For example, Tpk1, but not Tpk2 and Tpk3 regulates non-homologous end joining double-stranded break repair by phosphorylating Nej1³⁴. While Tpk2 activates pseudohyphal growth, Tpk3 inhibits filamentation and Tpk1 has no effect³⁵.”

Results

5. Page 6-lines 106-107. In these lines it is the first time that the authors mention that the yeast cells were grown in medium containing different glucose concentrations. In the section material and methods, they do not describe clearly what the culture conditions were, or at least I could not find this information. How were the cells grown? to early log phase? Which was the OD600 chosen? When the cells were in the OD600 the cells were washed once, and resuspended in the medium lacking glucose and then the glucose was added to different final concentrations? Or were cells grown to early log phase overnight in different glucose concentrations? Did the authors use pre-cultures? Were the cultures really grown with 0% glucose as indicated in the figures? It is important that each time that the authors write “cells were grown”, they detail the growth phase (early log phase?, OD?). It is necessary to clarify how the yeast cells were grown.

We thank this reviewer for this comment. We added “**Cell growth and treatment**” in the Methods section (Page 28, line 639) to describe how cells were grown and treated in detail. In the result section (Page 5, line 109), we also added the detail about how cells were treated. For example, we changed “To examine the effect of glucose on histone modifications, we grew yeast cells in medium containing different concentrations of glucose and found that glucose can induce H3K4me3 but not H3K4me1 and H3K4me2 in a dose-dependent manner (Fig. 1a)” to “To examine the effect of glucose on histone modifications, we grew yeast cells in 2% glucose-containing medium (YPD) until log phase. Cells were collected and then grown in medium containing different concentrations of glucose for 0.5 hr. By Western blot analysis of known histone modifications, we found that glucose can induce H3K4me3

but not H3K4me1 and H3K4me2 in a dose-dependent manner (Fig. 1a).”

Page 5, line 109: Changed “To examine the effect of glucose on histone modifications, we grew yeast cells in medium containing different concentrations of glucose and found that glucose can induce H3K4me3 but not H3K4me1 and H3K4me2 in a dose-dependent manner (Fig. 1a).” to “To examine the effect of glucose on histone modifications, we grew yeast cells in 2% glucose-containing medium (YPD) until log phase. Cells were collected and then grown in medium containing different concentrations of glucose for 0.5 hr. By Western blot analysis of known histone modifications, we found that glucose can induce H3K4me3 but not H3K4me1 and H3K4me2 in a dose-dependent manner (Fig. 1a).”

Page 28, line 639: Added “**Cell growth and treatment.** To examine the effect of glucose on histone modifications, yeast cells were grown in 2% glucose-containing YPD (Yeast Extract Peptide Dextrose) medium until OD₆₀₀ of 0.7-1.0. Cells were then collected, washed and resuspended in medium containing different concentrations of glucose for 3 hr. For 1NM-PP1 treatment, cells were grown in 2% YPD until OD₆₀₀ of 0.7. Cells were then treated with 25 μM 1NM-PP1 for 0.5 hr.”

6. Page 6- line 121. The authors wrote that they deleted PDE1 and PDE2, however in the strains table, is described that the source of these strains is Open Biosystems. Related to this concern, in figure 1 panel I, change *ped2Δ* by *pde2Δ*.

We apologized for this confusion. In the revised manuscript, we changed “We thus deleted cAMP phosphodiesterases *PDE1* and *PDE2*” to “We thus examined H3K4me3 in deletion mutants of cAMP phosphodiesterases *PDE1* and *PDE2*”. We also changed *ped2Δ* to *pde2Δ* in Fig. 1h as requested.

Page 7, line 145: Changed “We thus deleted cAMP phosphodiesterases *PDE1* and *PDE2*” to “We thus examined H3K4me3 in deletion mutants of cAMP phosphodiesterases *PDE1* and *PDE2*”.

7. Fig 1 panel h. This panel is not necessary, the decreased level of cAMP is well described previously. There are several works (at least since 1983).

We thank this reviewer for this comment. We removed Fig. 1h from Fig. 1 and cited relevant references as suggested.

Page 7, line 145: Changed “We thus deleted cAMP phosphodiesterases *PDE1* and *PDE2*, which encode enzymes to hydrolyze cAMP and inactivate PKA. As expected, the cAMP levels were significantly increased in *pde1Δ* and *pde2Δ* mutants (Fig. 1h).” to “We thus examined H3K4me3 in deletion mutants of cAMP phosphodiesterases

PDE1 and *PDE2*, which encode enzymes to hydrolyze cAMP and inactivate PKA^{36,37}. In accordance with increased cAMP levels in *pde1Δ* and *pde2Δ* mutants^{36,37}, H3K4me3 was significantly elevated (Fig. 1h).”

8. Page 7- line 128. “While loss of Ras1 slightly reduce H3K4me3...”. The difference described by the authors between Ras1 and Ras2 is not clear since both mutants show differences with WT strain with the same significance (extended data Fig. 1 c).

We thank this reviewer for this comment. We agree with the reviewer that loss of Ras1 and Ras2 reduced H3K4me3 with the same significance. In the revised manuscript, we changed “While loss of Ras1 slightly reduced H3K4me3, loss of Ras2 remarkably reduced H3K4me3 (Extended Data Fig. 1c)” to “Loss of Ras1 and Ras2 significantly reduced H3K4me3 (Supplementary Fig. 1f)”.

Page 7, line 151: Changed “While loss of Ras1 slightly reduced H3K4me3, loss of Ras2 remarkably reduced H3K4me3 (Extended Data Fig. 1c)” to “Loss of Ras1 and Ras2 significantly reduced H3K4me3 (Supplementary Fig. 1f)”.

9. Pag 7- line 129-130. “Ras1/2 can be activated by glycolysis derived fructose 1,6-biphosphate (FBP) 25 (Fig. 1g)”.

FBP activates Ras1/2 trough *cdc25* not directly. I consider that *cdc25* should be added in the text and figure.

We thank this reviewer for this important suggestion. It is true that FBP activates Ras1/2 through *Cdc25*. In the revised manuscript, we changed “Ras1/2 can be activated by glycolysis-derived fructose 1,6-biphosphate (FBP) ⁶ (Fig. 1g)” to “Ras1/2 can be activated by glycolysis-derived fructose-1,6-biphosphate (FBP) via *Cdc25* ⁶ (Fig. 1g)”. In addition, we added *Cdc25* in Fig. 1g (**Fig. 1g**).

Page 3, line 58: Added “The glycolytic metabolite, fructose-1,6-biphosphate (FBP) activates small GTP-binding proteins, Ras1 and Ras2 through the guanine exchange factor, *Cdc25* ⁶”.

Page 7, line 152: Changed “Ras1/2 can be activated by glycolysis-derived fructose 1,6-biphosphate (FBP) ¹ (Fig. 1g)” to “Ras1/2 can be activated by glycolysis-derived fructose-1,6-biphosphate (FBP) via *Cdc25* ⁶ (Fig. 1g)”.

10. Pag 7- line 130-131. The treatment with FBP and Pyruvate is not described in Material and methods, Have the authors used yeast spheroplasts to assure the cells permeability?

We thank the reviewer for this comment. We pre-treated cells with zymolase to get yeast spheroplasts to increase cell permeability. In the revised manuscript, we described how we treated cells with FBP and pyruvate in the Methods section.

Page 29, line 643: Added “For FBP and pyruvate treatment, cells were pretreated with Zymolase to increase the permeability. In brief, cells were grown in YPD medium until OD₆₀₀ of 0.7-1.0, centrifuged and resuspended in 1 ml SB buffer (100 mM Sorbitol, 20 mM Tris pH 7.4). 10 µl Zymolase 20T was then added and incubated at 30°C for 15 min. The spheroplasts were washed and cultured in YPD medium containing different concentrations of FBP and pyruvate for 3 hr.”

11. Pag 7- line 131. Misspelling: lines instead line.

We changed “lines” to “line” as suggested.

12. Pag 7- line 136. Reference 26 should be replaced by a review.

We replaced reference 26 with a review paper (*Annu Rev Genet* 2008, 42:27-81).

13. Fig 2a. The strain Jhd2-TAP is not described in Strains table

We added Jhd2-TAP in Supplementary table 1.

14. Fig 2 e. The sequence RXXS/T is not the canonical consensus sequence of PKA. The real sequence consensus is RRXS/T.

We changed “RXXS/T” to “RRXS/T” in Fig. 2e.

15. Pag 9-Line 190. Sequence NLS: 313-RLLPAEKLSIDELEEMFWSLVTKNR RSS-340

Is the E really an E or an R? Please clarify this mistake.

We thank this reviewer for this comment. It is “R” instead of “E”. We feel sorry for this typo error.

Page 10, line 226: Changed “₃₁₃RLLPAEKLSIDELEEMFWSLVTKNRRSS₃₄₀.” to “₃₁₃RLLPARKLSIDELEEMFWSLVTKNRRSS₃₄₀.”

16. Pag 10-Lines 194-195. There is not mention to the granular distribution of Jhd2 (Extended Data Fig. 3b).

We changed “Consistently, we also observed more Jhd2 in the nucleus in Jhd2 S2A and *tpk2Δ* mutants by immunofluorescence (Extended Data Fig. 3b)” to “Consistently, we also observed more Jhd2 in the nucleus with granular distribution in Jhd2-2SA and *tpk2Δ* mutants by immunofluorescence (Supplementary Fig. 4b)”.

Page 11, line 230: Changed “Consistently, we also observed more Jhd2 in the nucleus in Jhd2 S2A and *tpk2Δ* mutants by immunofluorescence (Extended Data Fig. 3b)” to “Consistently, we also observed more Jhd2 in the nucleus with granular distribution in Jhd2-2SA and *tpk2Δ* mutants by immunofluorescence (Supplementary Fig. 4b)”.

17. Pag 11-Line 237. Peptides H3 (1-23). Detail in material and methods the peptides used. Sometimes the peptide is mentioned as Peptides H3 (1-23) and others as Peptides H3. Please, make uniform.

We thank the reviewer for this comment. In the Methods section, we added the description about how the peptide pull-down assay was performed as well as the information for peptide sequences. In addition, we made the expression of “Peptides H3 (1-23)” uniform throughout the manuscript.

Page 13, line 277: Changed “We performed the peptide pull-down assay by incubating purified WT Jhd2 and Jhd2-2SA with the biotinylated unmodified or H3K4 trimethylated (H3K4me3) (1-23) peptides.” to “We performed the peptide pull-down assay by incubating purified WT Jhd2 and Jhd2-2SA with the biotinylated unmodified H3 (1-23), or H3K4 trimethylated (H3K4me3) (1-23) peptides.”

Page 13, line 279: Changed “Compared with WT Jhd2, more Jhd2-2SA was pulled down by both unmodified and H3K4me3 H3 peptides.” to “Compared with WT Jhd2, more Jhd2-2SA was pulled down by both unmodified H3 (1-23) and H3K4me3 H3 (1-23) peptides.”

Page 13, line 283: Changed “we confirmed that Jhd2-2SA has a higher binding affinity to H3 peptide compared to WT Jhd2 (Fig. 4b).” to “we confirmed that Jhd2-2SA has a higher binding affinity to H3 (1-23) peptide compared to WT Jhd2 (Fig. 4b).”

Page 13, line 285: Changed “H3K14ac indeed reduced the binding of WT Jhd2 to H3 peptide; however, H3K14ac had little effect on the binding of Jhd2-2SA to H3 peptide” to “H3K14ac indeed reduced the binding of WT Jhd2 to H3 (1-23) peptide; however, H3K14ac had little effect on the binding of Jhd2-2SA to H3 (1-23) peptide”.

Page 37, line 826: Added “Peptide Pull-down Assay. The peptide pull-down assays

were performed in accordance with the protocol described previously⁵⁸. In brief, 10 µg biotinylated H3 (1-23) peptides were incubated with 1 mg protein in binding buffer (50 mM Tris-HCl, pH 7.5, 250 mM NaCl, 0.1% NP-40, 1 mM PMSF, protease inhibitor cocktail) at 4 °C overnight. The peptides and associated proteins were pulled down by incubation with Streptavidin beads (Amersham) at 4 °C for 4 hr. The beads were washed with 3×1ml binding buffer and boiled for 5 min. The supernatant from boiled beads was subject to Western blot analysis. The peptides used are the following:
Unmodified H3 (1-23): ARTKQTARKSTGGKAPRKQIASK;
H3K4me3 (1-23): ARTK(me3)QTARKSTGGKAPRKQIASK;
H3K14ac (1-23): ARTKQTARKSTGGK(ac)APRKQIASK.”

18. Pag. 12- Lines 253-256. Figure 4 e, the difference of distribution of Jhd2WT and Jhd2 S2A in subcellular fractions, is not well visualized. The % of total protein in each fraction should be calculated.

We performed two additional replicates for the subcellular fractionation experiments in Fig. 4e and did the quantification about the percentage of total protein in each fraction. Our data showed that the total Jhd2 and chromatin-bound Jhd2 were significantly higher in Jhd2-2SA mutant than WT (Fig. 4e).

19. Extended Figure 4a. It is Pull-down not IP

We changed “IP” to “Pull-down” (now Supplementary Fig. 5a).

20. Fig 4 h. The figure legend should describe what is defined as center.

We added “Center was defined as the binding peaks for Jhd2” in the figure legend of Fig. 4h.

Page 46, line 1137: Added “Center was defined as the binding peaks for Jhd2.”

REVIEWER COMMENTS

Reviewer #1 (Remarks to the Author):

In this resubmission, Yu and colleagues have diligently undertaken the experiments suggested by the reviewers. Their earnest effort at completing the many experiments to address the reviewers' concerns is highly commendable. Their results in general match with their conclusions. The authors have done a series of experiments to lay the mechanistic pathway towards the regulation of autophagy during nutrient signaling via alterations in the epigenetic landscape. At some places, their interpretations and data presentation seem a bit forced or biased to fit their narrative and should be avoided. Some concerns still exist and are listed below, and they need to be addressed prior to publication.

1) Lines 63-64, the authors state "...the Ras/cAMP/PKA pathway that regulates vegetative growth, carbohydrate metabolism, and entry into meiosis in response to glucose 7". The reference cited does not describe meiosis but describes pseudohyphal growth in yeast in response to LOW glucose (not 'in response to glucose' as stated). Given that this manuscript deals with replicative life span and growth of yeast, where the authors intending to say "mitosis"? The authors should take care in citing references correctly. A role for Jhd2 in gametogenesis/sporulation is known from the work of Meneghini and Madhani groups (Xu et al 2012) but is not cited.

2) It is well-established that H3K79me3 is strictly dependent on H2B monoubiquitination. Reported cryo-EM structures also show how the contacts of Dot1 H3K79 methyltransferase with ubiquitin conjugated to histone H2B can stimulate its activity. Therefore, it is surprising that the authors do not see an increase in H3K79me3 when they see a dose-dependent increase in H2BK123ub1 and H3K4me3. It appears that the authors are attempting to showcase or interpret their findings to fit their overall narrative that Ras-cAMP-PKC regulates H3K4me3 via controlling Jhd2 H3K4 demethylase activity.

3) H3K36me3 is also reported to be regulated by H2Bub1 (Bilokapi and Halic (2019) Nat Communications 10:3795). So, again it is surprising that the authors do not see an increase in H3K36me3 levels. Throughout the manuscript, the immunoblots presented show a single concentration or amount of their cell extracts is used, it is therefore possible that they are not in the linear range of detection in their immunoblotting to detect the changes in H3K79me3 and H3K36me3.

4) The authors have relabeled their phospho-site mutant from Jhd2-S2A originally to Jhd2-2SA in this revised version, but it would be appropriate to label them as Jhd2-S321A,S340A.

5) In Supplementary Figure 9h, Change Atg8-GFP (i.e., GFP as a C-terminal tag) label to GFP-Atg8!

Reviewer #2 (Remarks to the Author):

All my concerns have been addressed appropriately

Reviewer #3 (Remarks to the Author):

Although the new version of this manuscript has improved compared to the previous one, the authors have not answered all my concerns.

Below there are listed the comments which have not been answered.

In the other hand, the concept of cAMP-PKA signal transduction specificity has not been sufficiently explored. I recommend that authors include topics such as BCY1 interactors, subunits localization, and regulation of PKA subunit expression to better understand why Tpk2-specific phosphorylation occurs.

Concerns without answer

Fig 4 o and p. In the legend, it should be detailed where the primers were designed. (ORF?). The genes chosen to confirm the ChIP-seq data, are from the 614 co-binding genes identified?

Pag 14 Lines 285-286. "The protein level of Jhd2 was significantly higher in Jhd2 S2A and *tpk2Δ* mutants than is not caused by differences in the transcription. WT Jhd2 and this difference was not caused by altered Jhd2 transcription (Fig. 5a; Extended Data Fig. 5a), suggesting that Jhd2 phosphorylation might affect Jhd2 protein stability".

The methodological approach performed to obtain the data of Extended Data Fig 5 a is a qRT-PCR (I suppose it because it is not indicated in the legend). With this approach is not right to conclude that the difference observed between WT, Jhd2SA and *tpk2Δ* is not caused by differences in transcription. The mRNA levels measured are those from steady state, and therefore, the detectable levels are a result of degradation and transcription. The same with the protein levels, which could be the result of differences in protein translation. This paragraph should be changed.

Pag 14 line 292-293. The Jhd2 stability is regulated by ubiquitination and proteasome. This concept has just been published previously. The new result is that this process is regulated by Tpk2 phosphorylation. This should be clarified and reference 29 should be cited here.

Extended Data Fig 5b. There is a mistake, Jhd2SA instead of *cim3-1*.

Figure 6 c- In this figure it could be observed that Tpk1 also affects H3K4m3. The effect is observed in *tpk1Δ* and *tpk1Δ tpk2Δ*. This effect is neither mentioned in the results nor discussed in discussion section.

-The center in ChIP seq data of Jhd2 with Rpd3 and Jhd2SA with Rpd3 are coincident?

- Fig 6L. The 200 genes in which overlapping binding sites of Jhd2 and Rpd3 (in Jhd2 S2A mutant) were detected correspond to Sites thar are unique for Jhd2? Or do they include Co-occupied sites with wt?

Point-by-point response to the referees' comments

Reviewers' Comments:

Our responses are in blue.

Reviewer #1 (Remarks to the Author):

In this resubmission, Yu and colleagues have diligently undertaken the experiments suggested by the reviewers. Their earnest effort at completing the many experiments to address the reviewers' concerns is highly commendable. Their results in general match with their conclusions. The authors have done a series of experiments to lay the mechanistic pathway towards the regulation of autophagy during nutrient signaling via alterations in the epigenetic landscape. At some places, their interpretations and data presentation seem a bit forced or biased to fit their narrative and should be avoided. Some concerns still exist and are listed below, and they need to be addressed prior to publication.

We thank this reviewer for these positive comments on our revision. In the revised manuscript, we performed additional experiments to address the reviewer's concerns. In addition, we presented our data in a more unbiased way.

1) Lines 63-64, the authors state "...the Ras/cAMP/PKA pathway that regulates vegetative growth, carbohydrate metabolism, and entry into meiosis in response to glucose ⁷". The reference cited does not describe meiosis but describes pseudohyphal growth in yeast in response to LOW glucose (not 'in response to glucose' as stated). Given that this manuscript deals with replicative life span and growth of yeast, where the authors intending to say "mitosis"? The authors should take care in citing references correctly. A role for Jhd2 in gametogenesis/sporulation is known from the work of Meneghini and Madhani groups (Xu et al 2012) but is not cited.

We apologize for not citing references properly. In the introduction section of the revised manuscript, we changed "regulates vegetative growth, carbohydrate metabolism, and entry into meiosis in response to glucose ⁷" to "regulates vegetative growth, carbohydrate metabolism, morphogenesis, stress resistance, cell cycle progression and meiosis ^{7, 8, 9, 10}". In addition, we added the reference (Xu et al 2012) about the role of Jhd2 in gametogenesis/sporulation.

Page 3, paragraph 1: Changed "regulates vegetative growth, carbohydrate metabolism, and entry into meiosis in response to glucose ⁷" to "regulates vegetative growth, carbohydrate metabolism, morphogenesis, stress resistance, cell cycle progression and meiosis ^{7, 8, 9, 10}".

Page 4, paragraph 2: Added "Jhd2 has been reported to demethylate H3K4 to regulate

the transcription of postmeiotic genes that are critical for the production of healthy meiotic progeny²⁷.”

2) It is well-established that H3K79me3 is strictly dependent on H2B monoubiquitination. Reported cryo-EM structures also show how the contacts of Dot1 H3K79 methyltransferase with ubiquitin conjugated to histone H2B can stimulate its activity. Therefore, it is surprising that the authors do not see an increase in H3K79me3 when they see a dose-dependent increase in H2BK123ub1 and H3K4me3. It appears that the authors are attempting to showcase or interpret their findings to fit their overall narrative that Ras-cAMP-PKC regulates H3K4me3 via controlling Jhd2 H3K4 demethylase activity.

We thank this reviewer for this comment. We agree with the reviewer that H3K79me3 depends on H2B ubiquitination. As the reviewer suggested, we loaded serial-diluted cell extracts to make sure the Western blots were detected within the linear range. We indeed observed significantly increased H3K79me3 induced by glucose (Supplementary Fig. 1a). We also performed ChIP-qPCR to examine the effect of glucose on H3K79me3 at genes (*PAU8*, *DRS2*) controlled by H2Bub. We observed glucose significantly increased H3K79me3 at H2Bub-dependent genes (*PAU8*, *DRS2*) but not at H2Bub-independent gene (*ATG5*) (See rebuttal letter fig. 1).

Rebuttal letter fig. 1. Glucose increases the occupancy of H3K79me3 at H2Bub-dependent genes. **a** RT-qPCR analysis the transcription of genes in WT and *bre1Δ* mutant. Bre1 is the E3 ligase for H2B monoubiquitination (H2Bub). The transcription of *PAU8* and *DRS2* was dependent on H2Bub. **b** ChIP-qPCR analysis the enrichment of H3K79me3/H3 at indicated genes when cells were grown in YP + 0.05% glucose and YP + 2% glucose.

Page 5, paragraph 3: Changed “By Western blot analysis of known histone modifications, we found that glucose induced H3K4me3 but not H3K4me1 and H3K4me2 in a dose-dependent manner (Fig. 1a). Other active histone markers, such as

H3K36me3 and H3K79me3 were not significantly increased by glucose (Supplementary Fig. 1a).” to “By Western blot analysis of known histone modifications, we found that glucose induced H3K4me3, and to a less extent induced H3K36me3 and H3K79me3 but had no effect on H3K4me1 and H3K4me2 in a dose-dependent manner (Fig. 1a; Supplementary Fig. 1a).”

3) H3K36me3 is also reported to be regulated by H2Bub1 (Bilokapi and Halic (2019) Nat Communications 10:3795). So, again it is surprising that the authors do not see an increase in H3K36me3 levels. Throughout the manuscript, the immunoblots presented show a single concentration or amount of their cell extracts is used, it is therefore possible that they are not in the linear range of detection in their immunoblotting to detect the changes in H3K79me3 and H3K36me3.

We thank this reviewer for this good suggestion. We agree with the reviewer that H3K36me3 depends on H2B ubiquitination. As per suggested by the reviewer, we loaded serial-diluted cell extracts to make sure the Western blots were detected within the linear range. We indeed observed significantly increased H3K36me3 by glucose (Supplementary Fig. 1a).

Page 5, paragraph 3: Changed “By Western blot analysis of known histone modifications, we found that glucose induced H3K4me3 but not H3K4me1 and H3K4me2 in a dose-dependent manner (Fig. 1a). Other active histone markers, such as H3K36me3 and H3K79me3 were not significantly increased by glucose (Supplementary Fig. 1a).” to “By Western blot analysis of known histone modifications, we found that glucose induced H3K4me3, and to a less extent induced H3K36me3 and H3K79me3 but had no effect on H3K4me1 and H3K4me2 in a dose-dependent manner (Fig. 1a; Supplementary Fig. 1a).”

4) The authors have relabeled their phospho-site mutant from Jhd2-S2A originally to Jhd2-2SA in this revised version, but it would be appropriate to label them as Jhd2-S321A,S340A.

We changed all Jhd2-2SA labels to Jhd2-S321A, S340A as suggested in the revised manuscript.

5) In Supplementary Figure 9h, Change Atg8-GFP (i.e., GFP as a C-terminal tag) label to GFP-Atg8!

We feel sorry for this mistake. We changed Atg8-GFP label to GFP-Atg8 in the revised manuscript.

Reviewer #2 (Remarks to the Author):

All my concerns have been addressed appropriately

Reviewer #3 (Remarks to the Author):

Although the new version of this manuscript has improved compared to the previous one, the authors have not answered all my concerns.

We sincerely thank this reviewer for these useful comments and suggestions. We apologize for not answering all your concerns in the first revision. In this revised manuscript, we tried our best to address all your concerns, which significantly improved the manuscript.

Below there are listed the comments which have not been answered.

1) In the other hand, the concept of cAMP-PKA signal transduction specificity has not been sufficiently explored. I recommend that authors include topics such as BCY1 interactors, subunits localization, and regulation of PKA subunit expression to better understand why Tpk2-specific phosphorylation occurs.

We thank the reviewer for this comment. We explored the reason why Tpk2 specifically phosphorylates Jhd2 from the following aspects as suggested:

- (i) We examined the localization of Tpk1, Tpk2 and Tpk3 within cells when grown in 2% glucose-containing medium. Tpk1, Tpk2 and Tpk3 are distributed over the nuclear and the cytoplasmic compartments (Supplementary Fig. 11a), suggesting the specificity of Tpk2 to Jhd2 may not be related to its subcellular localization.
- (ii) We examined the expression of PKA subunits (Tpk1, Tpk2, Tpk3) when cells were treated with different concentrations of glucose. Although the expression of PKA subunits was not significantly altered by glucose treatment, the expression of Tpk1 and Tpk2 was significantly higher than Tpk3 (Supplementary Fig. 11b).
- (iii) We examined the effect of glucose on the interaction between Bcy1 and Tpk1/2/3. Our data showed that when cells were treated with 2% glucose, Tpk2 dissociated with Bcy1 with a faster kinetics than Tpk1 and Tpk3 (Supplementary Fig. 11c).

Taken together, the specificity of Tpk2 towards Jhd2 could be due to the relatively high expression of Tpk2 and faster dissociation kinetics of Tpk2 with Bcy1 in response to glucose availability. In addition, Bcy1 has been reported to interact with other proteins, i.e., Eno2 (enolase II), Hsp60, and Ira2 (Galello et al., J Proteomics 2014, 109: 261-75). It is possible that these Bcy1-interacting proteins may also contribute to the specificity of cAMP-PKA pathway. We have added these results in the discussion section.

Page 27, paragraph 2: Added “To understand the specific regulation of Jhd2 by Tpk2, we examined the subcellular localization of Tpk1, Tpk2 and Tpk3 within cells when grown in 2% glucose-containing medium (Supplementary Fig. 11a). Tpk1, Tpk2 and Tpk3 are distributed over both the nuclear and the cytoplasmic compartments, suggesting the specificity of Tpk2 to Jhd2 may not be related to its subcellular localization. It has been reported that PKA subunits are differentially expressed during fermentative growth⁵⁴. We then examined the expression of PKA subunits (Tpk1, Tpk2, Tpk3) when cells were treated with different concentrations of glucose. Although the expression of PKA subunits was not significantly altered by glucose, the expression of Tpk1 and Tpk2 was significantly higher than Tpk3 (Supplementary Fig. 11b). We also examined the effect of glucose on the interaction between Bcy1 and Tpk1/2/3. Our data showed that Tpk2 dissociated with Bcy1 with a faster kinetics than Tpk1 and Tpk3 (Supplementary Fig. 11c). It is possible that the relatively high expression of Tpk2 and the faster dissociation kinetics of Tpk2 with Bcy1 in response to glucose determine the specific regulation of Jhd2 by Tpk2. In addition, Bcy1 has been reported to interact with other proteins, i.e., Eno2 (enolase II), Hsp60, and Ira2⁵⁵. It is also possible that these Bcy1-interacting proteins may contribute to the specificity of cAMP-PKA pathway.”

Concerns without answer

2) Fig 4 o and p. In the legend, it should be detailed where the primers were designed. (ORF?). The genes chosen to confirm the ChIP-seq data, are from the 614 co-binding genes identified?

We thank the reviewer for this comment. The primers were designed for the ORF regions of indicated genes. *PMAI* and *YEF3* were chosen from 614 co-binding genes identified. *MPOI* has been reported to be regulated by Jhd2 (Soloveychik et al., 2016). We have added this information in the Fig. 4o-p figure legends.

Page 49, paragraph 1: Added: “*PMAI* and *YEF3* were chosen from 614 co-binding genes identified. *MPOI* was reported to be regulated by Jhd2²⁹. The primers were designed for the ORF regions of indicated genes.”

3) Pag 14 Lines 285-286. “The protein level of Jhd2 was significantly higher in Jhd2 S2A and *tpk2Δ* mutants than is not caused by differences in the transcription. WT Jhd2 and this difference was not caused by altered Jhd2 transcription (Fig. 5a; Extended Data Fig. 5a), suggesting that Jhd2 phosphorylation might affect Jhd2 protein stability”.

The methodological approach performed to obtain the data of Extended Data Fig 5 a is a qRT-PCR (I suppose it because it is not indicated in the legend). With this approach is not right to conclude that the difference observed between WT, Jhd2SA and *tpk2Δ* is not caused by differences in transcription. The mRNA levels measured are those from steady state, and therefore, the detectable levels are a result of degradation and

transcription. The same with the protein levels, which could be the result of differences in protein translation. This paragraph should be changed.

We thank the reviewer for this comment. We added qRT-PCR information in the figure legend of Supplementary Fig. 5a (Now Supplementary Fig. 6a). We agree with the reviewer that the data from qRT-PCR cannot conclude that the difference between WT, Jhd2SA and *tpk2Δ* is caused by differences in transcription. In the revised manuscript, we changed “The protein level of Jhd2 was significantly higher in Jhd2-2SA and *tpk2Δ* mutants than WT Jhd2 and this difference was not caused by altered *JHD2* transcription (Fig. 5a; Supplementary Fig. 6a), suggesting that Jhd2 phosphorylation might affect Jhd2 protein stability. To confirm that, we pretreated WT Jhd2 and Jhd2-2SA mutant with cycloheximide (CHX) to block protein synthesis.” to “The protein level of Jhd2 was significantly higher in Jhd2-S321A, S340A and *tpk2Δ* mutants than WT Jhd2 and the steady state mRNA levels of *JHD2* were similar in WT, Jhd2-S321A, S340A and *tpk2Δ* mutants (Fig. 5a; Supplementary Fig. 6a). To examine whether Jhd2 phosphorylation affects Jhd2 protein stability, we pretreated WT Jhd2 and Jhd2-S321A, S340A mutant with cycloheximide (CHX) to block protein synthesis.”

Page 15, paragraph 3: Changed “The protein level of Jhd2 was significantly higher in Jhd2-2SA and *tpk2Δ* mutants than WT Jhd2 and this difference was not caused by altered *JHD2* transcription (Fig. 5a; Supplementary Fig. 6a), suggesting that Jhd2 phosphorylation might affect Jhd2 protein stability. To confirm that, we pretreated WT Jhd2 and Jhd2-2SA mutant with cycloheximide (CHX) to block protein synthesis.” to “The protein level of Jhd2 was significantly higher in Jhd2-S321A, S340A and *tpk2Δ* mutants than WT Jhd2 and the steady state mRNA levels of *JHD2* were similar in WT, Jhd2-S321A, S340A and *tpk2Δ* mutants (Fig. 5a; Supplementary Fig. 6a). To examine whether Jhd2 phosphorylation affects Jhd2 protein stability, we pretreated WT Jhd2 and Jhd2-S321A, S340A mutant with cycloheximide (CHX) to block protein synthesis.”

4) Pag 14 line 292-293. The Jhd2 stability is regulated by ubiquitination and proteasome. This concept has just been published previously. The new result is that this process is regulated by Tpk2 phosphorylation. This should be clarified and reference 29 should be cited here.

We thank the reviewer for this comment. We agree with the reviewer that it has been reported that the Jhd2 stability is regulated by ubiquitination and proteasome. To clarify that point, we added “Inactivation of Cim3 in *cim3-1* mutant at 30°C but not 26°C significantly increased Jhd2 protein level (Supplementary Fig. 6b), which is consistent with the report that Jhd2 is ubiquitinated and degraded by the proteasome pathway²⁸”.

Page 16, paragraph 2: Added “which is consistent with the report that Jhd2 is ubiquitinated and degraded by the proteasome pathway²⁸”.

5) Extended Data Fig 5b. There is a mistake, Jhd2SA instead of *cim3-1*.

We apologize we did not make this clear. In Supplementary Fig. 5b (now Supplementary Fig. 6b), the mutant used is *cim3-1*. To make this clearer, we changed “Inactivation of Cim3 at 30°C but not 26°C significantly increased Jhd2 protein level in *cim3-1* mutant” to “Inactivation of Cim3 in *cim3-1* mutant at 30°C but not 26°C significantly increased Jhd2 protein level”.

Page 16, paragraph 2: Changed “Inactivation of Cim3 at 30°C but not 26°C significantly increased Jhd2 protein level in *cim3-1* mutant” to “Inactivation of Cim3 in *cim3-1* mutant at 30°C but not 26°C significantly increased Jhd2 protein level”.

6) Figure 6 c- In this figure it could be observed that Tpk1 also affects H3K4m3. The effect is observed in *tpk1Δ* and *tpk1Δ tpk2Δ*. This effect is neither mentioned in the results nor discussed in discussion section.

We thank this reviewer for this comment. It is true that loss of Tpk1 also led to reduced H3K4me3 and H3K14ac. But compared with Tpk2, Tpk1 had a mild effect on H3K4me3 and H3K14ac. It is possible that Tpk1 may phosphorylate Jhd2 at other sites rather than S321 and S340, which then affects H3K4 demethylation activity of Jhd2. It is also possible that Tpk1 may affect the activity of Set1. In the results section, we added “Loss of Tpk1 slightly reduced H3K4me3 in WT but not in *tpk2Δ* mutant (Fig. 1e).” and “Loss of Tpk1 also slightly reduced H3K14ac in WT but not in *tpk2Δ* mutant (Supplementary Fig. 7b).” We also discussed the possibility in the discussion section.

Page 7, paragraph 1: Added “Loss of Tpk1 slightly reduced H3K4me3 in WT but not in *tpk2Δ* mutant (Fig. 1e).”

Page 18, paragraph 2: Added “Loss of Tpk1 also slightly reduced H3K14ac in WT but not in *tpk2Δ* mutant (Supplementary Fig. 7b).”

Page 26, paragraph 2: Added “We also noted that loss of Tpk1 slightly reduced H3K4me3 and H3K14ac (Fig. 1e; Supplementary Fig. 7b). But compared with Tpk2, Tpk1 had a mild effect on H3K4me3 and H3K14ac. It is possible that Tpk1 may phosphorylate Jhd2 at other sites rather than S321 and S340, which then affects H3K4 demethylation activity of Jhd2. It is also possible that Tpk1 may affect the activity of Set1.”

7) -The center in ChIP seq data of Jhd2 with Rpd3 and Jhd2SA with Rpd3 are coincident?

We thank the reviewer for this comment. When we analyzed the ChIP-seq data for Jhd2

and Rpd3 in WT and Jhd2-S2A mutant, we make sure that the center in ChIP-seq data of Jhd2 with Rpd3 is coincident with the center of Jhd2-S2A with Rpd3. We have added this information in the figure legends of Fig. 6. We also would like to change if there is any point we misunderstood.

Page 50, paragraph 1: Added “The center in ChIP-seq data of Jhd2 with Rpd3 is coincident with the center of Jhd2-S321A, S340A with Rpd3.”

8) - Fig 6L. The 200 genes in which overlapping binding sites of Jhd2 and Rpd3 (in Jhd2 S2A mutant) were detected correspond to sites that are unique for Jhd2? Or do they include Co-occupied sites with wt?

We thank the reviewer for this comment. Among the 200 genes co-occupied by Jhd2 and Rpd3 in Jhd2-S2A mutant, 55 genes are unique for Jhd2 in WT cells, 16 genes are unique for Rpd3 in WT cells, 100 genes are co-occupied by Jhd2 and Rpd3 in WT cells. We have added this information in the result section.

Page 19, paragraph 3: Added “Among these 200 genes, 55 genes are unique for Jhd2 in WT cells, 16 genes are unique for Rpd3 in WT cells, 100 genes are co-occupied by Jhd2 and Rpd3 in WT cells.”

REVIEWERS' COMMENTS

Reviewer #3 (Remarks to the Author):

All my concerns have been addressed appropriately